# Aspartyl proteases target host actin nucleator complex protein to limit epithelial innate immunity

Sandip Patra[1,2] & Rupinder Kaur [iD][1][✉]

## Abstract

Epithelial-immune cell communication is pivotal to control microbial infections. We show that glycosylphosphatidylinositol-linked aspartyl proteases (Yapsins) of the human opportunistic pathogenic yeast *Candida glabrata* (*Cg*) thwart epithelial cell (EC)-neutrophil signalling by targeting the EC protein, Arpc1B (actin nucleator Arp2/3 complex subunit), which leads to actin disassembly and impeded IL-8 secretion by ECs. Further, the diminished IL-8 secretion inhibits neutrophil migration, and protects *Cg* from the neutrophil-mediated killing. CgYapsin-dependent Arpc1B degradation requires Arginine-142 in Arpc1B, and leads to reduced Arpc1B-p38 MAPK interaction and downregulated p38 signalling. Consistently, Arpc1B or p38 deletion promotes survival of the *Cg* aspartyl protease-deficient mutant in ECs. Importantly, kidneys of the protease-deficient mutant-infected mice display elevated immune cell infiltration and cytokine secretion, implicating CgYapsins in immune response suppression in vivo. Besides delineating *Cg*-EC interplay, our results uncover a novel target, Arpc1B, that pathogens attack to constrain the host signalling networks, and link Arpc1B mechanistically with p38 activation.

Keywords *Candida glabrata*; Arp2/3 Complex; Epithelial Cells; Epa1 Adhesin; p38 MAPK
Subject Categories Microbiology, Virology & Host Pathogen Interaction; Signal Transduction

## Introduction

*Candida* (*Nakaseomyces*) *glabrata* is a drug-resistant opportunistic fungal pathogen of high-priority which resides on mucosal surfaces of the oral cavity, genitourinary and gastrointestinal tracts in the healthy human host (Gabaldón and Carreté, 2016; Pappas et al, 2018; Kumar et al, 2019). It causes superficial mucosal and fatal invasive infections in individuals with a debilitated immune system (Pappas et al, 2018; Kumar et al, 2019). *C. glabrata* (*Cg*) deftly survives a high degree of diverse stresses, adheres to host tissues and abiotic surfaces, and impedes phagolysosome acidification, suppresses proinflammatory response and proliferates in macrophages (Kumar et al, 2019). Two key virulence factors, adhesins

and aspartyl proteases, encoded by multigene families, are pivotal to *Cg* pathogenesis (Timmermans et al, 2018; Kumar et al, 2019).

*Cg* possesses 81 putative glycosylphosphatidylinositol (GPI)-linked cell wall adhesins and 11 aspartyl proteases (Kaur et al, 2007; Xu et al, 2020; Kumar et al, 2019). The latter, also called CgYapsins, are encoded by *CgYPS1-11* genes, and aid *Cg* in maintaining cell wall, pH and vacuole homeostasis, and survive in macrophages, and fly and mouse hosts (Kaur et al, 2007; Kumar et al, 2019). CgYapsins are also involved in cleaving the major adhesin Epa1 off the cell wall, with *CgYPS1-11* deletion resulting in impeded Epa1 processing and increased adherence of *Cg* to Lec2 epithelial cells (ECs) (Kaur et al, 2007). Epa1 accounts for >90% adherence of *Cg* to ECs in vitro (Cormack et al, 1999). Epa1 binds to glycan-containing ligands including NKp46/NCR1 on natural killer (NK) cells, and, along with Epa6 and Epa7 adhesins, is important for NK cell-mediated *Cg* clearance (Vitenshtein et al, 2016).

*Cg* exists solely in the yeast form, however, it successfully invades multiple organs during invasive infections (Gabaldón and Carreté, 2016; Kumar et al, 2019). *Cg* invasive infections can originate from endogenous reservoirs including the gastrointestinal tract (Gouba and Drancourt, 2015; Richard and Sokol, 2019). Many host cell types including macrophages, neutrophils, natural killer cells and ECs constitute the host arsenal against *Cg* infections (Rasheed et al, 2020a; Kumar et al, 2019). However, *Cg*-host cell interactions are yet to be deciphered fully. Particularly, very little is known about how *Cg* surmounts the host epithelial barrier during disseminated infections. Here, we uncover the EC protein Arpc1B (actin-related protein 2/3 complex subunit 1B) to be a novel interactor of two key CgYapsins, CgYps1 and CgYps7. We show that CgYapsins suppress inflammatory signalling by interacting with Arpc1B, which result in Arpc1B downregulation, release of p38 from the Arpc1B-p38 complex, suppression of p38 MAPK activation and decreased cytokine secretion. Since Arpc1B mutations are associated with an increased propensity to develop infections (Kahr et al, 2017; Brigida et al, 2018; Tur-Gracia and Martinez-Quiles, 2021), our findings open up a new therapeutic avenue, involving Arp2/3 complex augmentation, for infectious diseases.

## Results

### CgYapsins modulate *Cg* adherence and survival in host ECs

CgYapsin loss is associated with elevated adherence to Lec2 ECs (Kaur et al, 2007). Therefore, to delineate the functions of

[1]Laboratory of Fungal Pathogenesis, BRIC-Centre for DNA Fingerprinting and Diagnostics, Hyderabad-500039, Telangana, India. [2]Graduate Studies, Regional Centre for Biotechnology, Faridabad-121001, Haryana, India. [✉]E-mail: rkaur@cdfd.org.in

CgYapsins in host adhesion, we examined the adherence of *Cg wild-type* (*wt*) and *Cgyps1-11Δ* (lacks eleven CgYapsins, CgYps1-11) strains to human gastric (AGS) and kidney (A-498) epithelial cell lines. Compared to ~30% adherent *wt*, 60% *Cgyps1-11Δ* cells adhered to live A-498 cells (Fig. 1A). Importantly, the *Cg* adherence pattern was similar for formaldehyde-fixed A-498 cells (Fig. EV1A), suggesting that formaldehyde fixation largely preserves the cell surface architecture in A-498. Further, the mutants lacking *CgYPS1* or *CgYPS7* genes exhibited elevated adherence to A-498, while the *Cgyps2ΔCΔ* mutant {lacks 9 *CgYPS* [*CgYPS2*, and eight cluster (*CgYPS3-6,8-11*)] genes} (Kaur et al, 2007) exhibited *wt*-like adherence (Fig. 1B). The *Cgyps1-11Δ* mutant also adhered 1.9-fold better to AGS than *wt*, however, *Cg* adherence to AGS was not affected upon *CgYPS1* or *CgYPS7* deletion (Fig. EV1B). While underscoring the host cell type specificity and functional redundancy among CgYapsins, these data corroborate our earlier finding (Kaur et al, 2007) that CgYapsins negatively regulate *Cg* adherence to ECs.

Notably, *CgYPS1* or *CgYPS7* expression reduced the hyperadherence of the *Cgyps1-11Δ* mutant, with both catalytic aspartate residues, D-91 and D-378, of CgYps1 (Rasheed et al, 2018) being essential for this rescue (Fig. 1B). Since CgYps7 catalytic residues are not known, their effect on *Cg* adhesion to ECs could not be examined. The diminished *albeit* the same adherence of *epa1Δ* and *epa1Δ6Δ7Δ* (lacks three key adhesins, Epa1, 6 and 7) mutants (Fig. EV1A,B), and the *epa1Δ*-like adherence of the *Cgyps1-11Δepa1Δ* mutant (Fig. EV1A) suggest that Epa1 majorly contributes to the observed adhesion. Altogether, these data underscore that *CgYPS1-11* loss rendered *Cg* hyperadherent to kidney and stomach EC types, and this elevated adhesion is mediated by Epa1.

Next, super-resolution imaging analysis confirmed the increased adherence of *Cgyps1-11Δ* to ECs (Fig. EV1C). Intriguingly, we observed few *wt* cells inside A-498 ECs during the imaging analysis. Thus, to probe this further, we performed inside/outside staining to differentiate A-498-internalized and A-498-associated *Cg*, wherein the mCherry-expressing intracellular and extracellular *Cg* were unstained and stained with concanavalin A-FITC, respectively. We found 2.5- and 2-fold higher and lower number of intracellular *wt* and *Cgyps1-11Δ* cells, respectively, at 24 h, compared to 4 h post-infection (Fig. 1C). This was further confirmed by colony-forming unit (CFU)-based analysis, wherein a 2.5- and 2-fold increase and decrease in *wt* and *Cgyps1-11Δ* CFUs, respectively, was observed at 24 h post-infection (Fig. 1D). These data suggest that while *CgYPS1-11* loss increases the adherence potential of *Cg*, it impairs the survival of *Cg* in ECs.

## CgYapsins suppress cytokine secretion

Since *Cg* lacks the invasive hyphal form (Gabaldón and Carreté, 2016; Kumar et al, 2019), we reasoned that *Cg*, for its uptake, could induce CgYapsin-dependent endocytosis in A-498 cells. To investigate this, we treated A-498 with a cell-permeable dynamin inhibitor dynasore which blocks clathrin-coated vesicle formation during dynamin-dependent endocytosis (Macia et al, 2006). We found ~8-fold decrease in *wt* and *Cgyps1-11Δ* CFUs in dynasore-treated A-498 at 4 h post-infection (Fig. 1E), which was also apparent in A-498 cells treated with both dynasore and lactose (inhibits Epa1-mediated adherence) (Cormack et al, 1999) (Fig. 1F),

thereby precluding any appreciable effect of dynasore on *Cg* adherence. Consistently, dynasore treatment altered neither *wt* nor *Cgyps1-11Δ* adherence (Fig. 1G).

Next, we asked if *Cg* internalization impacts cellular signalling in A-498. For this, we examined the EC response to *wt* and *Cgyps1-11Δ* infection by measuring cytokine secretion and signalling cascade activation. While we detected no change in GM-CSF secretion (Fig. EV1D), proinflammatory IL-6 (Fig. 2A) and IL-8 (Fig. 2B) cytokines were secreted at lower and higher levels in *wt*- and *Cgyps1-11Δ*-infected A-498, respectively, with heat-killing or UV-killing of *Cg* abrogating the effect. Importantly, this differential cytokine secretion pattern between *wt*- and *Cgyps1-11Δ*-infected A-498 (Fig. EV1E,F) was also observed at a tenfold higher MoI (multiplicity of infection), suggesting that *Cg* cell load differences have no significant impact on decreased and increased cytokine secretion in *wt*- and *Cgyps1-11Δ*-infected A-498, respectively. Of note, *Cg*-infected A-498 probably did not secrete TNF-α and IL-1β, as these were not detected in the culture medium. Further, dynasore treatment resulted in similar levels of IL-6 (Fig. EV1G) and IL-8 (Fig. EV1H) secretion in uninfected, *wt*-infected and *Cgyps1-11Δ*-infected A-498, underlining the effect of *Cg* endocytosis on cytokine secretion. Importantly, both CgYps7 and catalytically-active CgYps1 abolished the elevated IL-6 (Fig. 2C) and IL-8 (Fig. 2D) secretion in *Cgyps1-11Δ*-infected A-498, implicating CgYps1's proteolytic activity in modulating the EC response, and underscoring the functional redundancy between CgYps1 and CgYps7.

Next, to determine the in vivo relevance of the observed differential cytokine release upon infection of A-498 with *wt* and *Cgyps1-11Δ*, we checked the levels of IL-6 and KC and MIP-2 (murine homologs of IL-8) cytokines in kidney homogenates of uninfected, and *wt* and *Cgyps1-11Δ*-infected mice, with kidneys being the major target organ in the mouse systemic candidiasis model (Kaur et al, 2007; Rasheed et al, 2018). We found IL-6 (Fig. 3A), KC (Fig. 3B) and MIP-2 (Fig. 3C) levels to be ~1.5 and 3-fold higher in kidney homogenates of the *Cgyps1-11Δ*-infected mice, compared to the uninfected and *wt*-infected mice. These data suggest that CgYapsins suppress inflammatory response in vivo as well. Hematoxylin-eosin staining and anti-CD45 antibody (CD45 is the surface marker of immune cells) staining-based histological analysis further validated this inference, as the immune cell infiltration was highly increased in kidneys of the *Cgyps1-11Δ*-infected mice, compared to the uninfected and *wt*-infected mice (Fig. EV2A,B). Similarly, staining with anti-Ly6G antibody (Ly6G is the surface protein of mouse neutrophils) revealed 1.4- and 2-fold higher number of Ly6G-positive neutrophils in kidneys of the *Cgyps1-11Δ*-infected mice, compared to the uninfected and *wt*-infected mice, respectively (Fig. 3D). This increased immune cell migration may contribute to *Cgyps1-11Δ* clearance in mice. In this regard, it is worth noting that *Cgyps1-11Δ* is severely attenuated for survival in the murine model of systemic candidiasis (Kaur et al, 2007; Rasheed et al, 2018).

## CgYapsins suppress EC activation

To investigate the signalling pathways that could contribute to the differential cytokine response of A-498 to *wt* and *Cgyps1-11Δ* infection, we checked the activation status of three serine-threonine MAP kinases, extracellular signal-regulated kinase (ERK), stress-

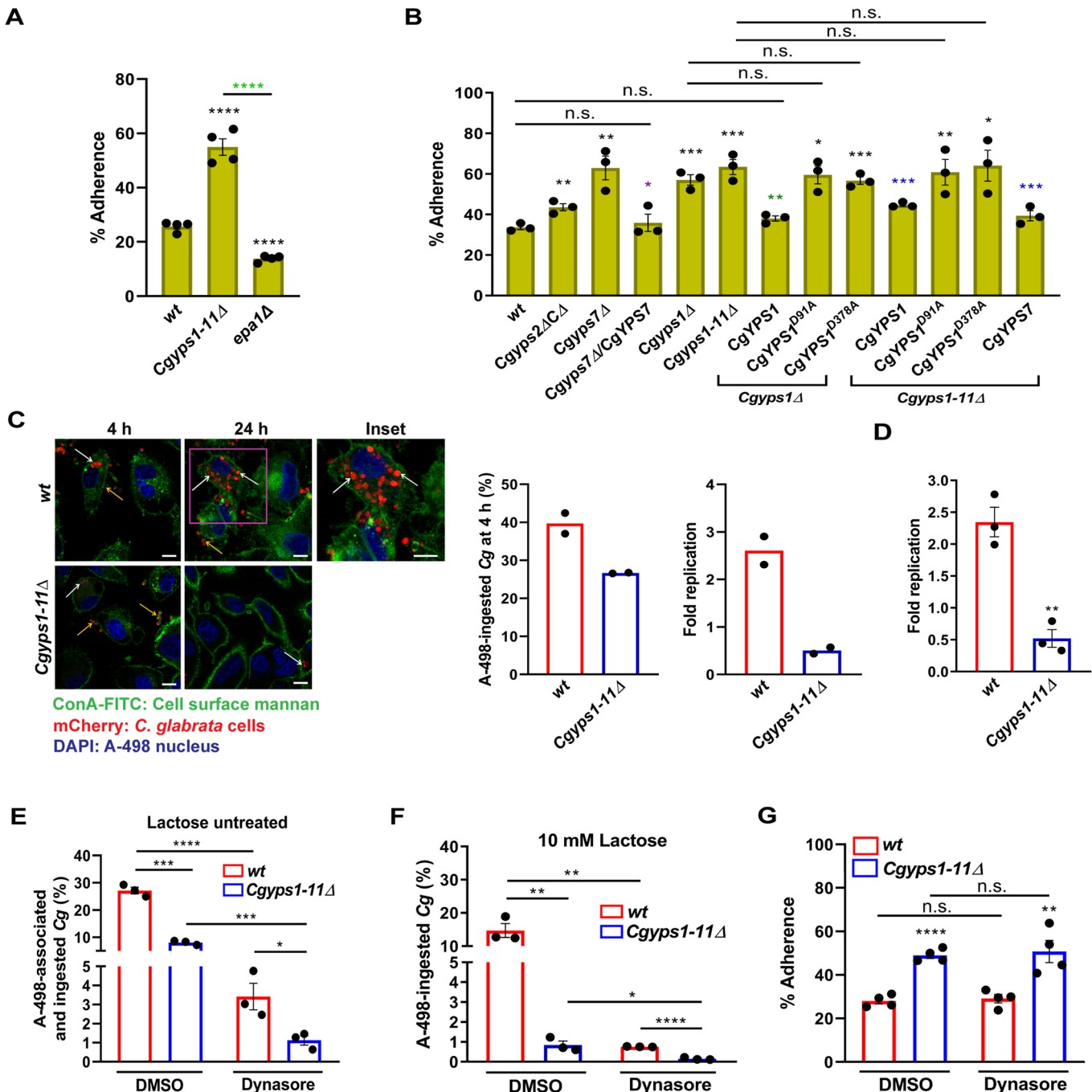

activated protein kinases c-Jun N-terminal kinase (JNK) and p38 kinase. In addition, we also examined the activation status of nuclear factor-κB (NF-κB) signalling, with NF-κB consisting of two subunits, p50 and p65 (Liu et al, 2017). Compared to uninfected A-498, phospho-ERK1/2 (Appendix Fig. S1A) and phospho-JNK1/2 (Appendix Fig. S1B) levels in *Cgyps1-11Δ*-infected A-498 were unaltered. However, while *wt* infection of A-498 led to ~35% reduction in phospho-ERK1/2 (Appendix Fig. S1A), phospho-JNK1/2 levels were similar in *wt*-infected and uninfected A-498 (Appendix Fig. S1B), indicating that ERK signalling is downregulated upon *Cg* infection. Further, the levels of phosphorylated-p38 MAPK (Fig. 4A) and

phospho-p65/RelA (reflects activated NF-κB signalling) (Fig. 4B) were ~2.5- to 6-fold higher in *Cgyps1-11Δ*-infected A-498, compared to uninfected and *wt*-infected A-498. Notably, *wt*-infected A-498 displayed lower phosphorylated-p38 (Fig. 4A) and higher phosphorylated-p65 levels (Fig. 4B), compared to uninfected cells. Consistent with the Western data, p65 nuclear localization was elevated in *wt*-infected and *Cgyps1-11Δ*-infected A-498 cells (Appendix Fig. S1C). Of note, the secreted aspartyl protease Sap6 in *C. albicans* has recently been reported to invoke IL-1β secretion in human oral epithelial cells which was mediated by p38 MAPK (Kumar et al, 2022). However, despite a hyperphosphorylated p38, we did not

Figure 1. CgYapsins govern *Cg*-epithelial cell interaction.

(A) Adherence of indicated, S[35]-labelled *Cg* strains to live A-498 (human kidney epithelial) cells after 2 h co-incubation, as determined by CFU (colony-forming unit)-based assay. Black asterisks denote adherence differences, as compared to *wt*-infected A-498 cells. *n* = 4 biological replicates. (B) Adherence of indicated, S[35]-labelled *Cg* strains to fixed A-498 (cells after 2 h co-incubation. Black, green, purple and blue asterisks indicate A-498 adherence differences in indicated strains, as compared to *wt*, *Cgyps1Δ*, *Cgyps7Δ* and *Cgyps1-11Δ* strains, respectively. *n* = 3 biological replicates. (C) Representative confocal micrographs showing association/internalization of mCherry-expressing *wt* and *Cgyps1-11Δ* by A-498 at 4 and 24 h post-infection (hpi). Infected A-498 cells were stained with concanavalin A (ConA)-FITC to distinguish intracellular and extracellular *Cg*. Representative z-stack images with a step size of 1 μm per slice were captured using Leica SP8 with 63X/1.44 NA objective lens. Yellow and white arrows mark A-498-associated (ConA-FITC-stained) and A-498-internalized (ConA-FITC-unstained) *Cg*, respectively. A minimum of 200 *Cg* cells were counted for each strain at 4 and 24 hpi. Data (mean; *n* = 2 biological replicates) represent the number of A-498-ingested (ConA-FITC-unstained) *Cg* cells. Fold-replication for each strain was determined by dividing the number of intracellular *Cg* at 24 h by that at 4 h. DAPI stains the cell nuclei. (D) CFU-based intracellular survival analysis. At 4 and 24 hpi, *wt*- and *Cgyps1-11Δ*-infected A-498 cells were lysed in water, followed by lysate plating on YPD medium. Fold-replication was determined by dividing *Cg* CFUs at 24 h by that at 4 h. *n* = 3 biological replicates. (E, F) *Cg* adherence and internalization analysis. A-498 cells, pre-treated with DMSO (solvent control) or dynasore (50 μM) were infected with *Cg* that were either left untreated (E) or blocked with 10 mM lactose (F), prior to A-498 infection. At 4 hpi, A-498 cells were washed, and A-498-associated/ingested *Cg* cell numbers were determined by plating A-498 lysates on YPD medium. % association and/or ingestion was calculated by dividing 4 h CFUs by 0 h CFUs (Number of *Cg* cells infected to A-498), and multiplying the number by 100. *n* = 3 biological replicates in (E, F). (G) Adherence of *wt* and *Cgyps1-11Δ* to DMSO or dynasore (50 μM)-pre-treated, fixed A-498 cells, after 2 h co-incubation, as determined by CFU-based assay. *n* = 4 biological replicates. Data information: In (A, B, D, E, F, G), data are presented as mean ± SEM. *P < 0.05; **P < 0.01; ***P < 0.001; ****P < 0.0001; n.s. not significant. Unpaired two-tailed Student's *t* test in (A, B, D, E, F, G). *P* = 0.00009466 (*Cgyps1-11Δ* vs. *wt*), *P* = 0.00003650 (*epa1Δ* vs. *wt*), *P* = 0.0000113 (*Cgyps1-11Δ* vs. *epa1Δ*) in (A). *P* = 0.0089 (*Cgyps2CΔ* vs. *wt*), *P* = 0.0077 (*Cgyps7Δ* vs. *wt*), *P* = 0.0005 (*Cgyps1Δ* vs. *wt*), *P* = 0.0016 (*Cgyps1-11Δ* vs. *wt*), *P* = 0.0133 (*Cgyps1Δ/CgYPS1^{D91A}* vs. *wt*), *P* = 0.0004 (*Cgyps1Δ/CgYPS1^{D378A}* vs. *wt*), *P* = 0.0047 (*Cgyps1-11Δ/CgYPS1^{D91A}* vs. *wt*), *P* = 0.0089 (*Cgyps1-11Δ/CgYPS1^{D378A}* vs. *wt*), *P* = 0.0014 (*Cgyps1Δ/CgYPS1* vs. *Cgyps1Δ*), *P* = 0.0197 (*Cgyps7Δ/CgYPS7* vs. *Cgyps7Δ*), *P* = 0.0008 (*Cgyps1-11Δ/CgYPS1* vs. *Cgyps1-11Δ*), *P* = 0.0006 (*Cgyps1-11Δ/CgYPS7* vs. *Cgyps1-11Δ*) in (B). *P* = 0.0025 (*Cgyps1-11Δ* vs. *wt*) in (D). *P* = 0.0001 (*Cgyps1-11Δ*-DMSO vs. *wt*-DMSO), *P* = 0.0357 (*Cgyps1-11Δ*-Dynasore vs. *wt*-Dynasore), *P* = 0.000074 (*wt*-Dynasore vs. *wt*-DMSO), *P* = 0.0002 (*Cgyps1-11Δ*- Dynasore vs. *Cgyps1-11Δ*- DMSO) in (E). *P* = 0.0027 (*Cgyps1-11Δ*-DMSO vs. *wt*-DMSO), *P* = 0.0000760 (*Cgyps1-11Δ*-Dynasore vs. *wt*-Dynasore), *P* = 0.0026 (*wt*-Dynasore vs. *wt*-DMSO), *P* = 0.0243 (*Cgyps1-11Δ*-Dynasore vs. *Cgyps1-11Δ*- DMSO) in (F). *P* = 0.00004431 (*Cgyps1-11Δ*-DMSO vs. *wt*-DMSO), *P* = 0.0076 (*Cgyps1-11Δ*-Dynasore vs. *wt*-Dynasore) in (G). Scale bar = 10 μm in (C). Source data are available online for this figure.

observe IL-1β secretion in *Cgyps1-11Δ*-infected A-498. Similarly, although *Cg* is known to activate IL-1β gene expression weakly in oral epithelial cells at 12 h post-infection (Schaller et al, 2002), IL-1β secretion in *wt*-infected A-498 cells was not observed. These discrepancies could either be due to a modest IL-1β release by A-498 cells, that could not be detected with our assay, or IL-1β secretion activation could be cell line context-dependent. Altogether, these data suggest that *Cg* activates NF-κB, and suppresses p38 and ERK signalling in ECs, with CgYapsins playing an important role in modulation of these signalling pathways. Furthermore, p38 MAPK activation in *Cgyps1-11Δ*-infected A-498 cells raises the possibility that *Cgyps1-11Δ* may invoke an inflammatory response in A-498 cells via p38 signalling.

To determine if increased cytokine secretion is dependent upon p38 activation, we used a specific p38 MAPK inhibitor, SB203580, and found elevated IL-6 (Fig. 4C) and IL-8 (Fig. 4D) secretion to be abrogated in *Cgyps1-11Δ*-infected A-498. Importantly, SB203580 treatment also led to a twofold increase in *Cgyps1-11Δ* viability, while exerting no effect on *wt* replication in A-498 (Fig. 4E), thereby attributing *Cgyps1-11Δ* killing to p38 activation. This killing could in part be due to increased reactive oxygen production upon hyperactive p38 signalling, as p38 activation is associated with increased ROS levels (Canovas and Nebreda, 2021). Moreover, SB203580 treatment had no effect on *wt* and *Cgyps1-11Δ* internalization (Fig. EV2C), underscoring that p38 signalling is not involved in *Cg* ingestion.

Further, BAY11-7082, which inhibits NF-κB signalling by blocking cytokine-stimulated phosphorylation of κB kinase (Pierce et al, 1997), restored *Cgyps1-11Δ* viability (Fig. 4F) and reduced IL-6 (Fig. 4G) and IL-8 (Fig. 4H) secretion in *Cgyps1-11Δ*-infected A-498. Of note, *wt* and *Cgyps1-11Δ* internalization was not affected by BAY11-7082 treatment (Fig. EV2D). Altogether, these data suggest that NF-κB activation contributes to the inflammatory response evoked upon *Cgyps1-11Δ* infection, without impacting *Cg*

ingestion. Further, since NF-κB signalling was also activated in *wt*-infected A-498 cells, *albeit* to a lesser extent than *Cgyps1-11Δ*-infected A-498 (Fig. 4B; Appendix Fig. S1C), the *wt*-induced decrease in cytokine secretion in A-498 cells may be due to downregulated ERK and p38 signalling, suppression of a yet-to-be-identified pathway or CgYapsins-mediated processing of IL-6 and IL-8 cytokines. In this context, it is noteworthy that many CgYapsins have recently been identified in the secretome of *Cg* (Rasheed et al, 2020a). Altogether, these data suggest that CgYapsins-dependent repression of p38 MAPK signalling activation could be a *Cg* defense strategy against the inflammatory response of the host epithelial cells.

## CgYapsins impair the actin filament assembly in ECs

To obtain mechanistic insights into *Cg*-mediated cell signalling suppression, we sought to identify EC proteins, that bind to CgYps1 and CgYps7, as when expressed individually, CgYps1 and CgYps7 were able to reverse the hyperadherence phenotype of the *Cgyps1-11Δ* mutant (Fig. 1B). For interactome analysis, we incubated Talon beads with *Escherichia coli*-purified CgYps1-6X-HIS and CgYps7-6X-HIS proteins, followed by addition of EC extracts. EC extract-incubated beads were used as negative control. The pulled-down proteins were identified by LC-MS/MS analysis. We identified 187 and 411 EC proteins that interacted with CgYps1 and CgYps7, respectively, with 102 common interactors (Fig. 5A; Dataset EV1). Functional analysis of common EC interactors revealed many gene ontology terms including actin cytoskeleton organization and RNA splicing, to be enriched (Fig. 5B; Dataset EV2). Therefore, to check the effect of the actin network on *Cg*-EC interaction, we first determined if *Cg* internalization is affected upon actin polymerization inhibition. For this, we measured *Cg* ingestion in cytochalasin D (actin polymerization inhibitor)-treated A-498 cells. A 2.6-fold decrease in *wt* internalization in cytochalasin D-treated cells,

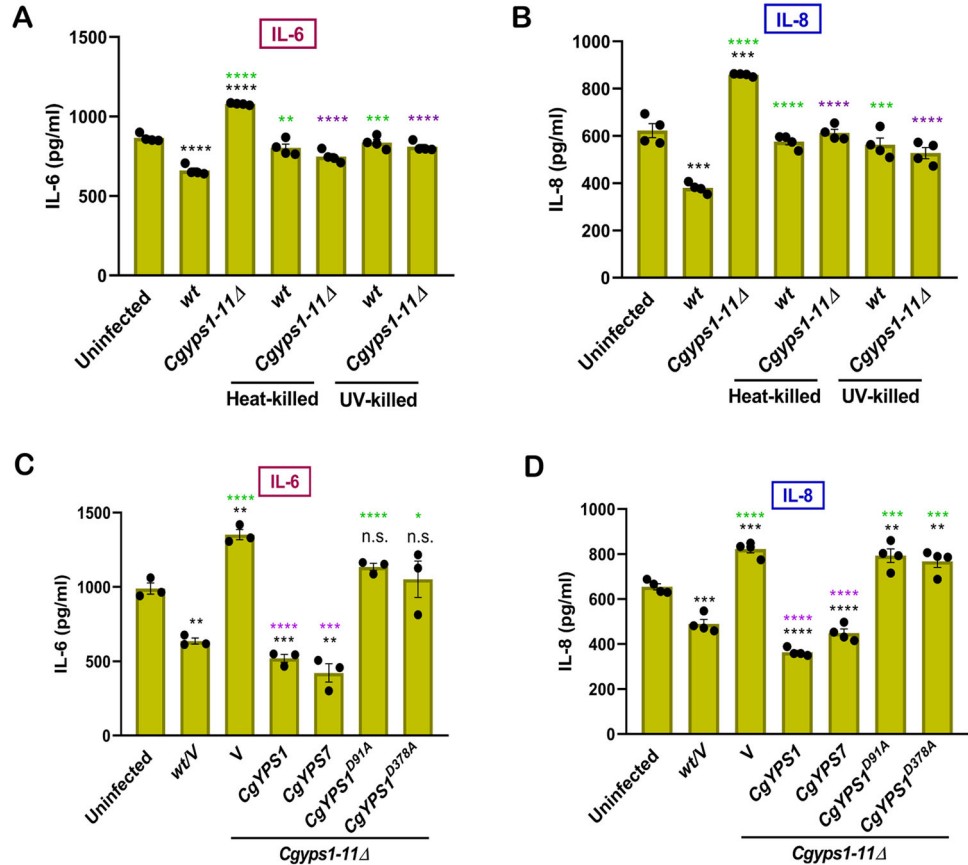

**Figure 2. CgYapsin loss leads to increased IL-6 and IL-8 secretion in A-498 cells.**

(A–D) Secreted IL-6 (A, C) and IL-8 (B, D) cytokine levels in uninfected, *wt*- and *Cgyps1-11Δ*-infected A-498 cells after 24 h incubation. Infection was carried out at 1:1 Mol (multiplicity of infection). *Cg* was killed either by heating (10 min incubation at 95 °C) or UV irradiation (four repeated exposures to 100 kJ/m² UV radiation). Purple, green and black asterisks indicate statistically-significant differences in cytokine secretion, as compared to *Cgyps1-11Δ*-infected, *wt*-infected and uninfected A-498 cells, respectively. *V*, empty vector. Strains carrying empty vector were used as control strains under indicated conditions. *n* = 4 biological replicates in (A, B, D), and *n* = 3 biological replicates in (C). Data information: In (A–D), data are presented as mean ± SEM. *P < 0.05; **P < 0.01; ***P < 0.001; ****P < 0.0001; n.s. not significant. Unpaired two-tailed Student's *t* test in (A–D). P = 0.0000001 (*Cgyps1-11Δ* vs. *wt*), P = 0.00004554 (*wt* vs. uninfected), P = 0.00000267 (*Cgyps1-11Δ* vs. uninfected), P = 0.0027 (*wt*-Heat-killed vs. *wt*-live), P = 0.0004 (*wt*-UV-killed vs. *wt*-live), P = 0.00000216 (*Cgyps1-11Δ*-Heat-killed vs. *Cgyps1-11Δ* -live), P = 0.00000169 (*Cgyps1-11Δ*-UV-killed vs. *Cgyps1-11Δ*-live) in (A). P = 0.000000013 (*Cgyps1-11Δ* vs. *wt*), P = 0.0002 (*wt* vs. uninfected), P = 0.0002 (*Cgyps1-11Δ* vs. uninfected), P = 0.0000336 (*wt*-UV-killed vs. *wt*-live), P = 0.0009 (*wt*-UV-killed vs. *wt*-live), P = 0.0000041 (*Cgyps1-11Δ*-Heat-killed vs. *Cgyps1-11Δ*-live), P = 0.0000083 (*Cgyps1-11Δ*-UV-killed vs. *Cgyps1-11Δ* live) in (B). P = 0.00005725 (*Cgyps1-11Δ/V* vs. *wt/V*), P = 0.0011 (*wt/V* vs. uninfected), P = 0.002 (*Cgyps1-11Δ/V* vs. uninfected), P = 0.0005 (*Cgyps1-11Δ/CgYPS1* vs. uninfected), P = 0.0014 (*Cgyps1-11Δ/CgYPS7* vs. uninfected), P = 0.00003838 (*Cgyps1-11Δ/CgYPS1* vs. *Cgyps1-11Δ/V*), P = 0.0002 (*Cgyps1-11Δ/CgYPS7* vs. *Cgyps1-11Δ/V*), P = 0.00009518 (*Cgyps1-11Δ/CgYPS1^{D91A}* vs. *wt/V*), P = 0.0285 (*Cgyps1-11Δ/CgYPS1^{D378A}* vs. *wt/V*) in (C). P = 0.000013 (*Cgyps1-11Δ/V* vs. *wt/V*), P = 0.0005 (*wt/V* vs. uninfected), P = 0.0002 (*Cgyps1-11Δ/V* vs. uninfected), P = 0.00000028 (*Cgyps1-11Δ/CgYPS1* vs. uninfected), P = 0.00000469 (*Cgyps1-11Δ/CgYPS7* vs. uninfected), P = 0.0056 (*Cgyps1-11Δ/CgYPS1^{D91A}* vs. uninfected) P = 0.0096 (*Cgyps1-11Δ/CgYPS1^{D378A}* vs. uninfected), P = 0.0000003 (*Cgyps1-11Δ/CgYPS1* vs. *Cgyps1-11Δ/V*), P = 0.0000047 (*Cgyps1-11Δ/CgYPS7* vs. *Cgyps1-11Δ/V*), P = 0.0001 (*Cgyps1-11Δ/CgYPS1^{D91A}* vs. *wt/V*), P = 0.0002 (*Cgyps1-11Δ/CgYPS1^{D378A}* vs. *wt/V*) in (D). Source data are available online for this figure.

compared to solvent-treated A-498 (Fig. EV2E), suggested that *Cg* internalization requires actin polymerization.

Next, from identified CgYps1 and CgYps7 interacting proteins, we selected Arpc1B (actin-related protein 2/3 complex subunit 1B), which is a subunit of the seven-subunit human Arp2/3 complex (Abella et al, 2016; Papalazarou and Machesky, 2021), for further analysis. Arp2/3 complex nucleates actin filaments, is required for actin branching and regulates cell growth and differentiation (Abella et al, 2016; Papalazarou and Machesky, 2021). The reasons for Arpc1B selection were threefold. First, Arpc1B interacted with both CgYps1 and CgYps7 (Dataset EV1). Second, mutations in Arpc1B are associated with increased susceptibility to microbial

infections (Kahr et al, 2017; Brigida et al, 2018; Tur-Gracia and Martinez-Quiles, 2021). Third, the role of Arpc1B in cellular defense against pathogenic fungi is not known.

We first verified the interaction of CgYps1 (Fig. EV3A) and CgYps7 (Fig. EV3B) with C-terminally SFB (S protein-FLAG-Streptavidin-binding peptide)-tagged Arpc1B that was ectopically expressed in A-498 cells. Since CgYapsin binding to Arpc1B could alter actin networks, we next visualized actin cytoskeletal structures using rhodamine-phalloidin. Uninfected and *Cgyps1-11Δ*-infected A-498 displayed uniformly-distributed actin network comprised of long actin fibres (Fig. 5C). Contrarily, actin filaments were diminished in *Cg* and *C. albicans* (used as a control)-infected

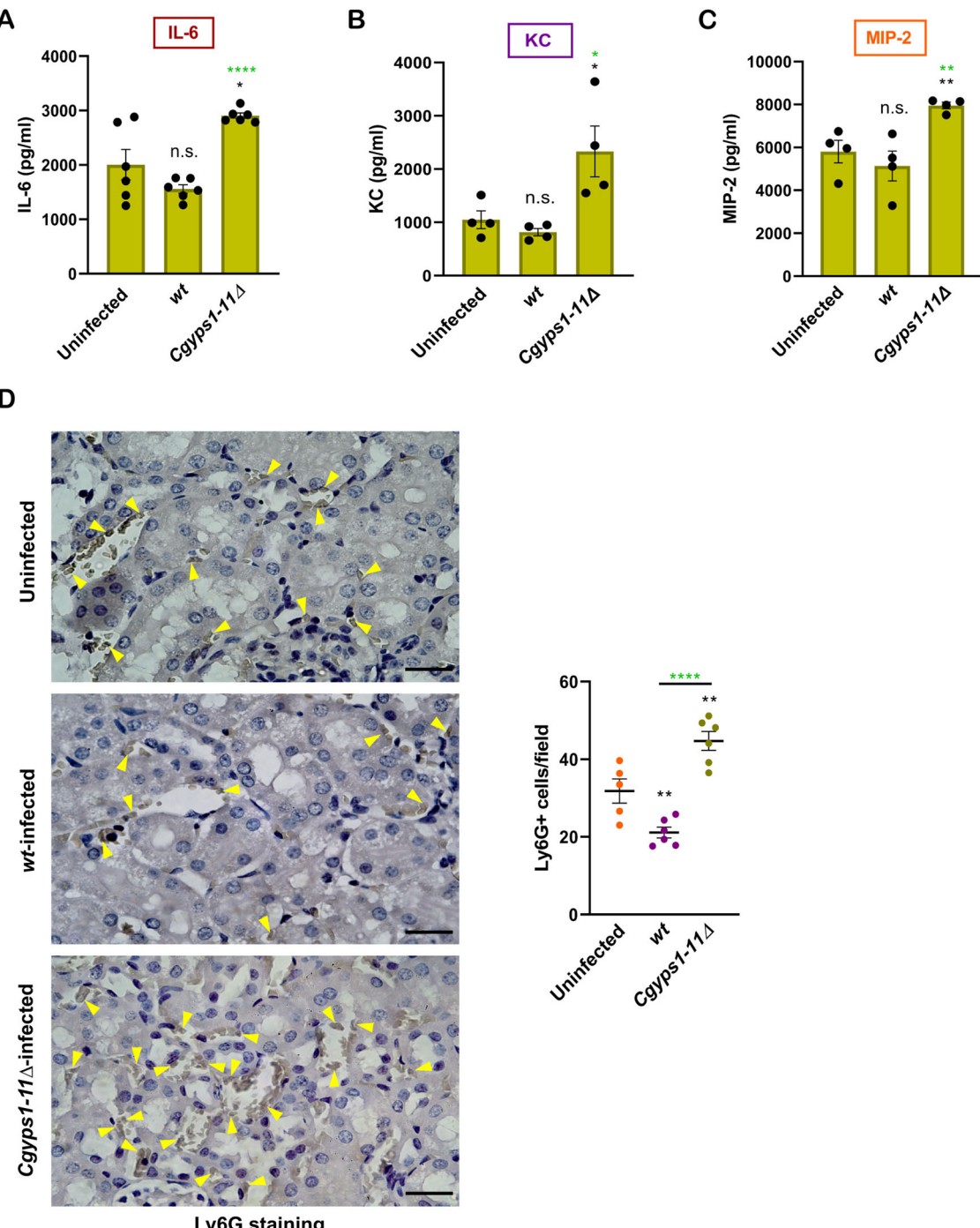

**Figure 3. CgYapsin loss results in increased infiltration of neutrophils.**

(A–C) IL-6 (A), KC (B) and MIP-2 (C) levels in kidney homogenates of *Cg*-infected BALB/c mice after 4 days of infection. $n = 6$ mice/group in (A) and $n = 4$ mice/group in (B, C). Green and black asterisks denote cytokine differences, as compared to *wt*-infected and uninfected mice, respectively. (D) Representative images of Ly6G-stained kidney sections of uninfected, *wt*-infected, and *Cgyps1-11Δ*–infected female BALB/c mice at day 1 post-infection. $n = 5$ mice for uninfected group, and $n = 6$ mice for *wt*- and *Cgyps1-11Δ*–infected groups. For quantification, a minimum of 100 Ly6G-positive (Ly6G +) cells, in 6–8 different fields of each mouse kidney section, were counted for each group. The yellow arrowheads mark representative Ly6G+ polymorphonuclear neutrophils. Data represent the average number of Ly6G positive cells per field for each group. Black asterisks denote Ly6G+ cell number differences, compared to uninfected mouse kidney sections. Images were taken at ×40 magnification. Data information: In (A–D), data are presented as mean ± SEM. $*P < 0.05$; $**P < 0.01$; $****P < 0.0001$; n.s. not significant. Unpaired two-tailed Student's $t$ test in (A–D). $P = 0.00000007$ (*Cgyps1-11Δ* vs. *wt*), $P = 0.0104$ (*Cgyps1-11Δ* vs. uninfected) in (A). $P = 0.0199$ (*Cgyps1-11Δ* vs. *wt*), $P = 0.044$ (*Cgyps1-11Δ* vs. uninfected) in (B). $P = 0.0074$ (*Cgyps1-11Δ* vs. *wt*), $P = 0.0076$ (*Cgyps1-11Δ* vs. uninfected) in (C). $P = 0.0083$ (*wt* vs. uninfected), $P = 0.0084$ (*Cgyps1-11Δ* vs. uninfected), $P = 0.0000077$ (*Cgyps1-11Δ* vs. *wt*) in (D). Scale bar = 20 μm in (D). Source data are available online for this figure.

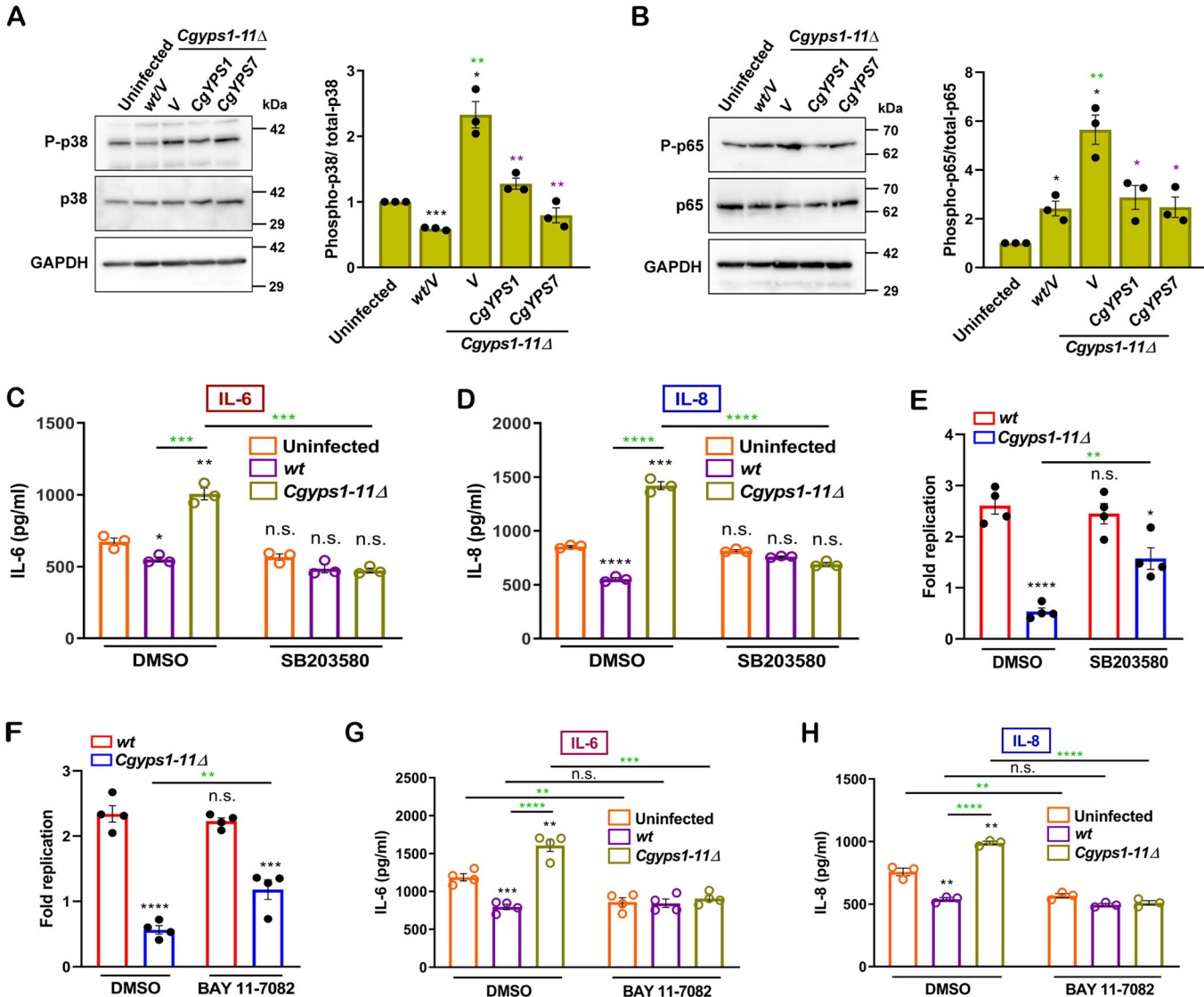

A-498, with these cells also containing actin aggregates (Fig. 5C), and increased globular actin levels (Fig. 5D). These results suggest that A-498 cells reorganize the filamentous actin assembly upon *Candida* exposure, and that, *Cg* probably engages CgYapsins to modulate the actin cytoskeleton in ECs.

Further, to investigate the effect of *Cg* infection on Arpc1B dynamics, we examined Arpc1B cellular levels. Arpc1B was expressed ubiquitously (Fig. 5E) but its levels were ~2.5-fold lower in *wt*-infected A-498, compared to uninfected A-498 (Fig. 5E). Arpc1B transcript levels remained unchanged (Fig. EV3C). Notably, Arpc1B protein levels in *Cgyps1-11Δ*-infected A-498 were similar and 2-fold higher, compared to uninfected and *wt*-infected A-498, respectively (Fig. EV3D). A-498 infected with *CgYPS1*- or *CgYPS7*-expressing *Cgyps1-11Δ* showed a reduction in Arpc1B levels, similar to *wt*-infected A-498 (Fig. EV3D). Further, Arpc1B levels in *wt*-infected A-498 were reduced within 30 min of incubation with *Cg* (Fig. 5F), thereby representing an early event during *Cg*-EC interaction. No change in Arpc1B levels even after 4 h incubation of A-498 with *Cgyps1-11Δ* (Fig. 5F) suggests that

CgYapsins are essential for Arpc1B reduction upon *Cg* infection. Consistently, CgYps1's catalytic activity was required for Arpc1B reduction in A-498 (Fig. 5G). Importantly, *wt* infection led to a decrease in Arpc1B levels in A-498, irrespective of the MoI used (Fig. EV3E), indicating that a reduction in Arpc1B levels is a bonafide outcome of *Cg*-EC interaction. Altogether, these results place Arpc1B, the interactor of CgYps1 and CgYps7, at the forefront during *Cg*-EC confrontation, with *Cg* probably stimulating Arpc1B degradation which results in dissolution of actin filaments.

## CgYapsins decrease Arpc1B-p38 interaction

*Cgyps1-11Δ*-infected A-498 cells displayed activated p38 signalling, increased IL-6 and IL-8 secretion, and no reduction in Arpc1B levels. To investigate if Arpc1B directly governs the inflammatory response in *Cgyps1-11Δ*-infected A-498, we performed seven experiments. First, we showed that Arpc1B interacts with both phosphorylated and non-phosphorylated forms of p38 (Fig. 5H). Second, Arpc1B-p38 interaction was lower and higher in *wt*-infected and *Cgyps1-11Δ*-infected A-498,

◀ **Figure 4. CgYapsin loss activates A-498 cells.**

(A, B) Representative immunoblots illustrating elevated phospho-p38 (A) and phospho-p65 (B) levels in *Cgyps1-11Δ*-infected A-498 cells. At 6 hpi, indicated epithelial cell lysates (120 µg) were resolved on 12% SDS-PAGE gel. The p38 and p65 protein bands correspond to 38 and 65 kDa, respectively. For quantification, the signal intensity in each lane was measured using the ImageJ software, and p38 (A) and p65 (B) intensity values were normalized against the corresponding GAPDH signal values. Fold-change in phosphorylated-p38 and phosphorylated-p65 levels was determined by normalizing against the GAPDH-normalized p38 (A) and p65 (B) signals, respectively. $n = 3$ biological replicates. Data represent fold-change in phosphorylated-p38 protein and phosphorylated-p65 protein levels in *Cg*-infected A-498, compared to uninfected A-498 cells (considered as 1.0) and are plotted on the right side of the blots. Purple, green and black asterisks mark statistically-significant differences, as compared to *Cgyps1-11Δ*-infected, *wt*-infected and uninfected A-498 cells, respectively. *V*, empty vector. (C, D) Secreted IL-6 (C) and IL-8 (D) levels in SB203580 (10 µM)-treated, indicated A-498 cells. Infection was carried out at 1:1 Mol. $n = 3$ biological replicates. (E) Intracellular *Cg* survival analysis in SB203580 (10 µM)-treated, indicated A-498 cells. Fold replication represents the ratio of 24 h CFUs to 4 h CFUs. $n = 4$ biological replicates. (F) Intracellular *Cg* survival analysis in BAY11-7082 (10 µM)-treated, indicated A-498 cells. Fold replication represents the ratio of 24 h CFUs to 4 h CFUs. $n = 4$ biological replicates. (G, H) Secreted IL-6 (G) and IL-8 (H) levels in BAY11-7082 (10 µM)-treated, indicated A-498 cells. Infection was carried out at 1:1 Mol. $n = 4$ biological replicates in (G), and $n = 3$ biological replicates in (H). Data information: In (A–H), data are presented as mean ± SEM. *$P < 0.05$; **$P < 0.01$; ***$P < 0.001$; ****$P < 0.0001$; n.s. not significant. Unpaired or paired two-tailed Student's *t* test in (A–H). $P = 0.001$ (*wt/V* vs. uninfected), $P = 0.0224$ (*Cgyps1-11Δ/V* vs. uninfected), $P = 0.001$ (*Cgyps1-11Δ/V* vs. *wt/V*), $P = 0.0086$ (*Cgyps1-11Δ/CgYPS1* vs. *Cgyps1-11Δ/V*), $P = 0.0027$ (*Cgyps1-11Δ/CgYPS7* vs. *Cgyps1-11Δ/V*) in (A). $P = 0.0415$ (*wt/V* vs. uninfected), $P = 0.0163$ (*Cgyps1-11Δ/V* vs. uninfected), $P = 0.0086$ (*Cgyps1-11Δ/V* vs. *wt/V*), $P = 0.0233$ (*Cgyps1-11Δ/CgYPS1* vs. *Cgyps1-11Δ/V*), $P = 0.0124$ (*Cgyps1-11Δ/CgYPS7* vs. *Cgyps1-11Δ/V*) in (B). $P = 0.0169$ (*wt* vs. uninfected), $P = 0.0024$ (*Cgyps1-11Δ* vs. uninfected), $P = 0.0005$ (*Cgyps1-11Δ* vs. *wt*), $P = 0.0003$ (*Cgyps1-11Δ*-SB203580 vs. *Cgyps1-11Δ*-DMSO) in (C). $P = 0.00004918$ (*wt* vs. uninfected), $P = 0.0001$ (*Cgyps1-11Δ* vs. uninfected), $P = 0.00002468$ (*Cgyps1-11Δ* vs. *wt*), $P = 0.00005211$ (*Cgyps1-11Δ*-SB203580 vs. *Cgyps1-11Δ*-DMSO) in (D). $P = 0.00002787$ (*Cgyps1-11Δ*-DMSO vs. *wt*-DMSO), $P = 0.0225$ (*Cgyps1-11Δ*-SB203580 vs. *wt*-SB203580), $P = 0.0034$ (*Cgyps1-11Δ*-SB203580 vs. *Cgyps1-11Δ*-DMSO) in (E). $P = 0.00001648$ (*Cgyps1-11Δ*-DMSO vs. *wt*-DMSO), $P = 0.0006$ (*Cgyps1-11Δ*-BAY 11-7082 vs. *wt*-BAY 11-7082), $P = 0.0096$ (*Cgyps1-11Δ*-BAY 11-7082 vs. *Cgyps1-11Δ*-DMSO) in (F). $P = 0.0006$ (*wt* vs. uninfected), $P = 0.0038$ (*Cgyps1-11Δ* vs. uninfected), $P = 0.00008$ (*Cgyps1-11Δ* vs. *wt*), $P = 0.0044$ (uninfected, BAY11-7082-treated vs. uninfected, DMSO-treated), $P = 0.0002$ (*Cgyps1-11Δ*-BAY11-7082 vs. *Cgyps1-11Δ*-DMSO) in (G). $P = 0.0029$ (*wt* vs. uninfected), $P = 0.0022$ (*Cgyps1-11Δ* vs. uninfected), $P = 0.0000200$ (*Cgyps1-11Δ* vs. *wt*), $P = 0.0055$ (uninfected, BAY11-7082-treated vs. uninfected, DMSO-treated), $P = 0.00002712$ (*Cgyps1-11Δ*-BAY11-7082 vs. *Cgyps1-11Δ*-DMSO) in (H). Source data are available online for this figure.

respectively (Fig. 5I,J), indicating that CgYapsins disrupt Arpc1B-p38 interaction which could partly be due to diminished Arpc1B levels in *wt*-infected A-498.

Third, p38 and Arpc1B co-localization was lower and higher in *wt*-infected and *Cgyps1-11Δ*-infected A-498, compared to uninfected A-498 (Fig. EV3F). Fourth, CK-666 (inhibitor of the Arp2/3 complex) treatment restored *Cgyps1-11Δ* viability partially in A-498 (Fig. 6A) and brought IL-6 (Fig. 6B) and IL-8 (Fig. 6C) secretion down to *wt*-infected A-498 levels, suggesting that Arp2/3 complex function inhibition aids *Cgyps1-11Δ* survival in ECs. Contrarily, CK-666 led to an increase in cytokine release in *wt*-infected A-498 (Fig. 6B,C), suggesting that CK-666 may have additional effects on other factors that affect cytokine responses upon *Cg* infection. Fifth, CK-666 had no effect on *wt* and *Cgyps1-11Δ* internalization (Fig. EV3G), thereby precluding a prominent role for the Arp2/3 complex in EC ingestion of *Cg*. Further, although CK-666 did not alter *wt* adherence, it abolished the increased adherence of *Cgyps1-11Δ* (Fig. EV3H). Sixth, siRNA-mediated knockdown of Arpc1B reduced *Cgyps1-11Δ* adherence (Fig. 6D), rescued *Cgyps1-11Δ* viability partially (Fig. 6E), and lowered IL-6 (Fig. 6F) and IL-8 (Fig. 6G) secretion in *Cgyps1-11Δ*-infected A-498, suggesting that the reduction in Arpc1B levels is pivotal to dampen the inflammatory response in A-498 cells. Of note, Arpc1B knockdown lowering the hyperadherence of *Cgyps1-11Δ* was unexpected, and could partly arise from the altered cell membrane structure upon Arpc1B knockdown. Consistent with this, Annexin V staining analysis revealed increased binding in Arpc1B-knockdown cells (Fig. EV3I), reflecting phosphatidylserine exposure probably due to a damaged cell membrane. Finally, Arpc1B-knockdown (Fig. EV4A,B), and CK-666 (Fig. EV4C) and SB203580 (Fig. EV4D) treatment impeded the nuclear localization of NF-κB. Collectively, these results suggest that Arpc1B is required for cytokine release in *Cgyps1-11Δ*-infected A-498, and that, CgYapsin-Arpc1B interaction may dictate the EC response outcome by inhibiting Arpc1B-p38 interaction, and, by-extension, of *Cg* infection.

## Arpc1B deletion impairs p38 signalling activation

*Cg*-responsive EC signalling is not well-defined. Therefore, to demonstrate unequivocally that p38 governs the cytokine response to *Cg* infection, we attempted to delete the p38-encoding MAPK14 gene in A-498 cells, but were unsuccessful. Therefore, we turned to HEK293T (immortalized human embryonic kidney cells), and deleted MAPK14 and ARPC1-B genes using two sets of paired guide RNAs simultaneously via CRISPR-Cas9 technology. We verified the lack of p38 and Arpc1B expression in independently generated p38−/− (Appendix Fig. S2A) and Arpc1B−/− (Appendix Fig. S2B) knockout cells, respectively. Similar to A-498, the *Cgyps1-11Δ* mutant was hyperadherent (Appendix Fig. S2C) and replication-defective (Appendix Fig. S2D) in HEK293T cells. Notably, while p38 (Fig. 6H) and Arpc1B (Fig. 6I) loss, respectively, led to an increased and a decreased adherence of *wt* and *Cgyps1-11Δ*, it also facilitated *Cgyps1-11Δ* proliferation in HEK293T cells (Fig. 6J,K). Further, p38 (Fig. 6L) and Arpc1B (Fig. 6M) deletion lowered down the IL-8 secretion in *Cgyps1-11Δ*-infected cells to that of uninfected HEK293T, whereas IL-6 was not detected in the culture supernatants of *Cg*-infected HEK293T cells. Of note, the more pronounced effect of p38 on *Cgyps1-11Δ* survival (Fig. 6J), compared to Arpc1B loss (Fig. 6K), suggests that p38 activation modulation is probably carried out by both Arpc1B-dependent and Arpc1B-independent mechanisms, and that, p38 may play a major role in regulating *Cg*-EC interplay. Moreover, the differential reversal of increased IL-8 secretion in *Cgyps1-11Δ*-infected ECs upon Arpc1B deletion (IL-8 levels similar to *wt*-infected ECs) and p38 deletion (IL-8 levels similar to uninfected ECs) indicate that Arpc1B and p38 MAPK-mediated signalling is not the sole pathway to control IL-8 secretion. Alternatively, Arpc1B and p38 MAPK may participate in additional pathways that control cytokine response upon *Cg* infection. Importantly, Arpc1B−/− cells exhibited both diminished F-actin fibres (Fig. 7A) and a twofold lower F/G-actin ratio (Fig. 7B), compared to Arpc1B+/+ cells, indicating that the reduced F-actin levels may contribute to the increased survival of the *Cgyps1-11Δ* mutant in

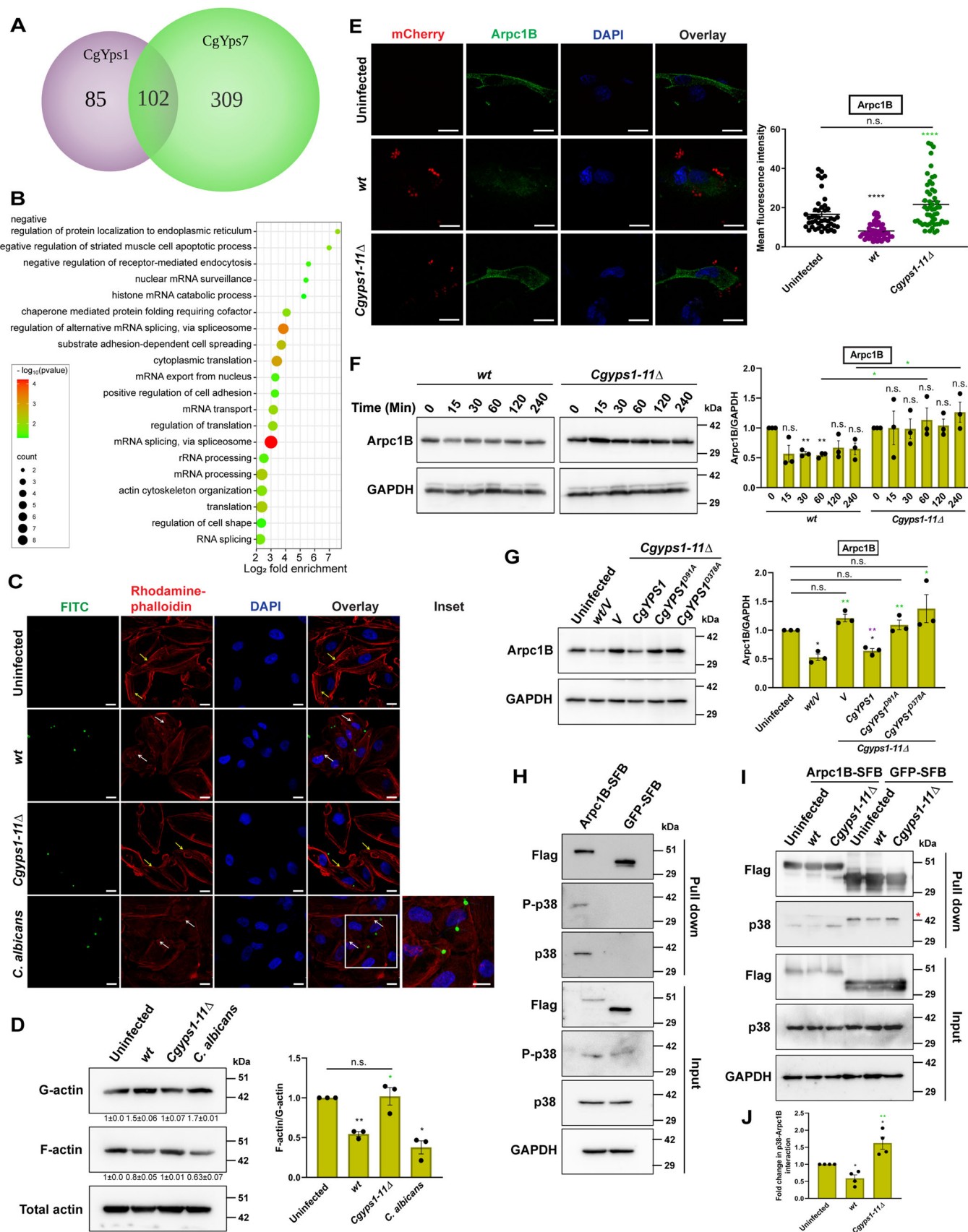

**Figure 5.  *Cg* infection reduces Arpc1B levels in A-498 cells.**

(A) Venn diagram illustrating overlap between A-498 protein interactors of CgYps1 and CgYps7. (B) Bubble plot showing enriched gene ontology (GO) terms for Biological Processes in the common host epithelial A-498 cell interactome of CgYps1 and CgYps7, as determined using the DAVID tool. (C) Representative confocal micrographs illustrating rhodamine-phalloidin-stained actin structures in indicated A-498 cells, at 6 hpi with FITC-labelled *Cg* or *C. albicans*. Representative z-stack images were obtained using the confocal microscope (Leica SP8) with 63X/1.44 NA objective lens. Yellow and white arrows mark actin fibres and aggregates, respectively. DAPI stains A-498 nuclei. Since FITC signal was poor in the hyphal form of *C. albicans*, the GFP signal intensity in images was increased to detect the hyphal form, and *C. albicans* hyphae are shown in the inset. $n = 2$ biological replicates. (D) Representative immunoblots illustrating total actin-normalized globular (G)-actin and filamentous (F)-actin levels in indicated A-498 cells at 6 hpi. Green and black asterisks mark F/G actin ratio differences, as compared to *Cg* wt-infected and uninfected (taken as 1.0) A-498 cells, respectively. $n = 3$ biological replicates. (E) Representative confocal micrographs depicting immunofluorescence analysis of Arpc1B cellular localization. A-498 cells were transfected with Arpc1B-SFB-expressing plasmid and infected with mCherry-expressing *Cg* strains for 6 h, followed by staining with anti-FLAG antibody, and imaging using the confocal microscope (Leica SP8) with 63X/1.44 NA objective lens in the z-stack mode. The fluorescent Arpc1B-SFB signal was quantified, using the ImageJ software, in a minimum of 44 cells, and shown as a scatter plot on the right side of the micrographs. Green and black asterisks indicate statistically significant differences, as compared to *wt*-infected and uninfected A-498 cells, respectively. $n = 3$ biological replicates. (F) Representative immunoblots illustrating Arpc1B levels in *wt*- and *Cgyps1-11Δ*-infected A-498 cells at indicated time points post-infection. Data represent fold-change in Arpc1B levels, as compared to 0 h time point (taken as 1.0). $n = 3$ biological replicates. (G) Representative immunoblots illustrating Arpc1B protein levels in A-498 cells that were left uninfected or infected for 6 h with indicated *Cg* strains expressing either active CgYps1 or catalytically-dead CgYps1 enzymes, CgYps1$^{D91A}$ and CgYps1$^{D378A}$. Data are plotted on the right side of the blots. Purple, green and black asterisks indicate Arpc1B level differences, as compared to *Cgyps1-11Δ/V*-infected, *wt/V*-infected and uninfected A-498 cells, respectively. *V* denotes empty plasmid. $n = 3$ biological replicates. (H) Representative immunoblots ($n = 2$ biological replicates) validating Arpc1B-SFB and p38 MAPK interaction. Lysates (1.5 mg) of Arpc1B-SFB or GFP-SFB-expressing A-498 cells were incubated with streptavidin beads for 2 h at 4 °C, and resolved on 12% SDS-PAGE. (I, J) Representative immunoblots ($n = 4$ biological replicates) illustrating reduced Arpc1B-p38 interaction in *wt*-infected A-498 at 6 hpi (I). Lysates of *Cg*-infected, Arpc1B-SFB or GFP-SFB-expressing A-498 cells were incubated with streptavidin beads, followed by sample resolution on 12% SDS-PAGE. Red asterisk denotes non-specific bands. Arpc1B-p38 interaction was determined by normalizing against the GAPDH (loading control)-normalized p38 signal, and quantified using the ImageJ software. Data represent fold-change in Arpc1B-p38 ($J$; $n = 4$ biological replicates) interaction, as compared to uninfected A-498 cells (taken as 1.0). Green and black asterisks indicate statistically significant differences, as compared to *wt*-infected and uninfected A-498 cells, respectively. Data information: In (D–G, J), data are presented as mean ± SEM. *$P < 0.05$; **$P < 0.01$; ****$P < 0.0001$; n.s., not significant. Benjamini test in (B). Unpaired or paired two-tailed Student's *t* test in (D–G, J). $P = 0.0045$ (*wt* vs. uninfected), $P = 0.0171$ (*C. albicans* vs. uninfected), $P = 0.014$ (*Cgyps1-11Δ* vs. *wt*) in (D). $P = 0.00000002$ (*wt* vs. uninfected), $P = 0.00000000002$ (*Cgyps1-11Δ* vs. *wt*) in (E). $P = 0.0051$ (*wt*-30 min vs. *wt*-0 min), $P = 0.003$ (*wt*-60-min vs. *wt*-0 min), $P = 0.0408$ (*Cgyps1-11Δ*-60 min vs. *wt*-60 min), $P = 0.0352$ (*Cgyps1-11Δ*-240 min vs. *wt*-240 min) in (F). $P = 0.0143$ (*wt/V* vs. uninfected), $P = 0.0123$ (*Cgyps1-11Δ/CgYPS1* vs. uninfected), $P = 0.0014$ (*Cgyps1-11Δ/V* vs. *wt/V*), $P = 0.0017$ (*Cgyps1-11Δ/CgYPS1* vs. *Cgyps1-11Δ/V*), $P = 0.0053$ (*Cgyps1-11Δ/CgYPS1$^{D91A}$* vs. *wt/V*), $P = 0.0279$ (*Cgyps1-11Δ/CgYPS1$^{D378A}$* vs. *wt/V*) in (G). $P = 0.0279$ (*wt* vs. uninfected), $P = 0.04$ (*Cgyps1-11Δ* vs. uninfected) $P = 0.0024$ (*Cgyps1-11Δ* vs. *wt*) in (J). Scale bar = 20 μm in (C). Scale bar = 10 μm in (E). Source data are available online for this figure.

the Arpc1B-deficient HEK293T cells. Collectively, these data highlight that p38 MAPK and Arpc1B control the EC inflammatory response to *Cg* infection.

We earlier showed that *Cg* infection led to a decrease in Arpc1B protein levels in ECs, which required the catalytic activity of CgYps1 (Fig. 5G). Therefore, we sought to employ Arpc1B$^{-/-}$ cells to determine the relevance of CgYps1-Arpc1B interaction for Arpc1B reduction. For this, we first identified, via molecular docking, two arginine residues, Arg-74 and Arg-142 in Arpc1B, that interacted with CgYps1. Notably, CgYps1 has recently been shown to cleave at an arginine residue in its yeast substrate CgPst2 (Battu et al, 2021). Next, we expressed the mutant Arpc1B proteins, that contained alanine in place of Arg-74 or Arg-142, in Arpc1B$^{-/-}$ cells. We found impaired actin filaments (Fig. 7C) and decreased F/G actin ratio (Fig. 7D), indicating the role of Arg-74 and Arg-142 residues in Arpc1B functions in maintaining the actin cytoskeletal network. Importantly, the Arg-142 residue was also found to be essential for *Cg*-induced reduction in Arpc1B levels (Fig. 7E), thereby raising the possibility of CgYps1-dependent processing of Arpc1B to occur at Arg-142, with Arg-142 residue also being pivotal to Arpc1B functions in F-actin assembly. Whether mutations in Arg-142 codon increases susceptibility to fungal infections remains to be investigated.

Next, to dissect the cross talk among *Cg* infection, Arpc1B degradation, and p38 signalling suppression, we performed three experiments. First, we showed that Arpc1B (Fig. 7F) or p38 (Fig. 7G) deletion had no effect on *wt* ingestion, consistent with the CK-666 (Fig. EV3G) and SB203580 (Fig. EV2C) inhibitor data. Second, Arpc1B deletion did not affect internalization of the *Cgyps1-11Δ* mutant (Fig. 7F), while p38 deletion reduced *Cgyps1-11Δ* ingestion by ~10% (Fig. 7G), thereby ruling out any significant contribution of

Arpc1B and p38 MAPK to *Cg* internalization. Finally, p38 deletion did not alter Arpc1B levels (Fig. 7H), while Arpc1B$^{-/-}$ cells contained 5-fold less phosphorylated-p38, compared to Arpc1B$^{+/+}$ cells (Fig. 7I). Altogether, these data place p38 downstream of Arpc1B, and suggest that Arpc1B loss impairs p38 signalling activation. Consistently, IL-8 secretion was lower in Arpc1B-deficient cells (Fig. 6G,M). These results also favour the notion that *Cg* targets Arpc1B, via CgYapsins, to restrain p38 MAPK signalling, probably to limit the proinflammatory cytokine response.

## *Cg*-infected A-498 cells hamper neutrophil migration

The proinflammatory cytokine IL-8 acts as a chemoattractant for neutrophils (Kolaczkowska and Kubes, 2013). Thus, to define the biological relevance of *Cg*-mediated suppression of IL-8 secretion, we examined neutrophil migration towards *Cg* and *Cg*-infected A-498. We observed that 2.0-fold lower and 1.4-fold higher number of neutrophils migrated towards *wt*-infected and *Cgyps1-11Δ*-infected A-498, respectively, compared to uninfected A-498 (Fig. 8A), indicating a negative effect of the *wt* infection of ECs on the neutrophil movement. Importantly, *Cg*-neutrophil co-incubation led to no appreciable neutrophil migration (Appendix Fig. S2E), suggesting that *wt* and *Cgyps1-11Δ* do not directly impact the neutrophil mobility. Further, only 55% *wt* and 30% *Cgyps1-11Δ* cells survived 2 h co-incubation with neutrophils (Fig. 8B), indicating that CgYapsins aid *Cg* evade neutrophil killing. Collectively, these data suggest that CgYapsins modulate the EC response to prevent the IL-8-mediated activation of neutrophil recruitment and the neutrophil attack on *Cg*. Consistently, higher IL-8 levels and elevated neutrophil infiltration in kidneys of *Cgyps1-11Δ*-infected mice (Fig. 3) are likely to account for the attenuated

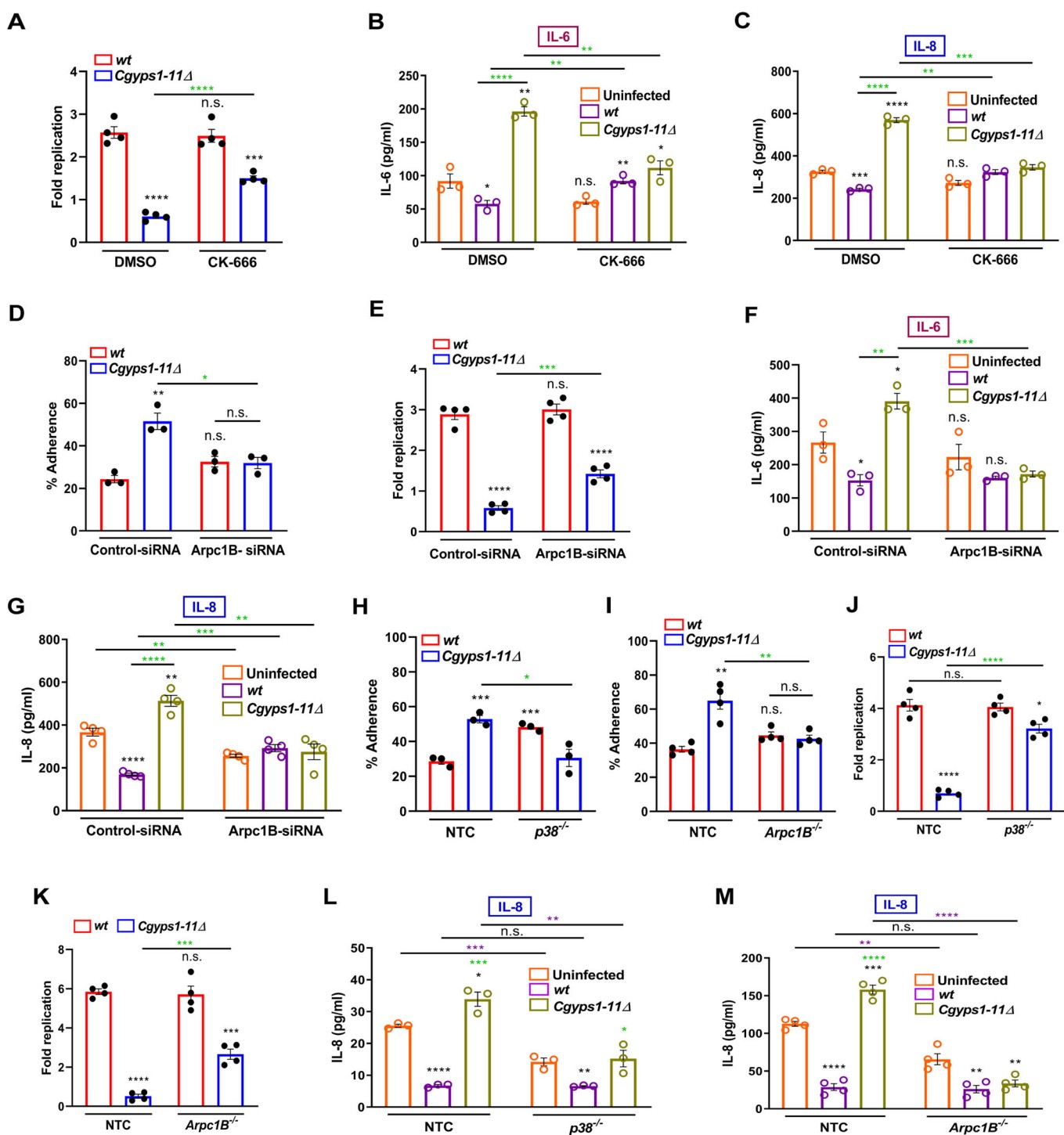

survival of the *Cgyps1-11Δ* mutant in mice (Kaur et al, 2007; Rasheed et al, 2018).

## Discussion

Despite *Cg* infections being associated with high mortality (Meyahnwi et al, 2022; Rasheed et al, 2020a), our understanding of *Cg*-host communication is limited. Epithelial cells are pivotal sentinel of the host defense against opportunistic pathogens, as these restrict pathogen dissemination by myriad strategies (Larsen et al, 2020). We show that *Cg* adheres to different EC types, with CgYapsin loss elevating *Cg* adherence to kidney (Fig. 1A), stomach (Fig. EV1B), intestinal (Fig. EV5A) and ovary epithelial cells (Kaur et al, 2007). The cell surface adhesin proteins are known to aid *Cg* adhesion, with many adhesins being upregulated during *Cg*-vaginal

◄ **Figure 6. Arpc1B or p38 loss reverses elevated cytokine secretion in *Cgyps1-11Δ*-infected A-498 cells.**

(A) Intracellular *Cg* survival analysis in CK-666 (1 µM)-treated, indicated A-498 cells. $n = 4$ biological replicates. (B, C) Secreted IL-6 (B) and IL-8 (C) levels in CK-666 (1 µM)-treated, indicated A-498 cells. Infection was carried out at 1:1 Mol. $n = 3$ biological replicates. (D) Adherence of indicated, $S^{35}$-labelled *Cg* strains to control-siRNA- or Arpc1B-siRNA-transfected, fixed A-498 cells after 2 h co-incubation. $n = 3$ biological replicates. (E) Intracellular *Cg* survival analysis in control-siRNA or Arpc1B-siRNA-transfected A-498 cells. $n = 4$ biological replicates. (F, G) Secreted IL-6 (F) and IL-8 (G) levels in control-siRNA or Arpc1B-siRNA-transfected, *Cg*-infected A-498 cells. Infection was carried out at 1:1 Mol. $n = 3$ biological replicates in (F), and $n = 4$ biological replicates in (G). (H) Adherence of indicated, $S^{35}$-labelled *Cg* strains to fixed, $p38^{-/-}$ and NTC (non-targeting control; cells transfected with the non-targeting control guide RNA)-HEK293T cells after 2 h co-incubation. Black asterisks mark adherence differences, as compared to *wt*-infected, NTC-HEK293T cells. $n = 3$ biological replicates. (I) Adherence of indicated *Cg* strains to fixed $Arpc1B^{-/-}$ and NTC-HEK293T cells, after 2 h co-incubation, as determined by CFU-based assay. Black asterisks mark adherence differences, as compared to *wt*-infected, NTC-HEK293T cells. $n = 4$ biological replicates. (J, K) Intracellular *Cg* survival analysis in $p38^{-/-}$ (J) and $Arpc1B^{-/-}$ (K) HEK293T epithelial cells. $n = 4$ biological replicates. (L, M) Secreted IL-8 levels in *Cg*-infected, $p38^{-/-}$ (L) or $Arpc1B^{-/-}$ (M) HEK293T cells. Infection was carried out at 1:1 Mol. Green and black asterisks indicate differences, as compared to *wt*-infected and uninfected, NTC-HEK293T cells, respectively. $n = 3$ biological replicates in (L), and $n = 4$ biological replicates (M). Data information: In (A–M), data are presented as mean ± SEM. *$P < 0.05$; **$P < 0.01$; ***$P < 0.001$; ****$P < 0.0001$; n.s. not significant. Unpaired two-tailed Student's *t* test in (A–M). $P = 0.00000956$ (*Cgyps1-11Δ*-DMSO vs. *wt*-DMSO), $P = 0.0009$ (*Cgyps1-11Δ*-CK-666 vs. *wt*-CK-666), $P = 0.00002031$ (*Cgyps1-11Δ*-CK-666 vs. *Cgyps1-11Δ*-DMSO) in (A). $P = 0.045$ (*wt* vs. uninfected), $P = 0.0012$ (*Cgyps1-11Δ* vs. uninfected), $P = 0.00009810$ (*Cgyps1-11Δ* vs. *wt*), $P = 0.0071$ (*wt*-CK-666-treated vs. uninfected, CK-666-treated), $P = 0.01$ (*Cgyps1-11Δ*-CK-666-treated vs. uninfected, CK-666-treated), $P = 0.0075$ (*wt*-CK-666 vs. *wt*-DMSO), $P = 0.0026$ (*Cgyps1-11Δ*-CK-666 vs. *Cgyps1-11Δ*-DMSO) in (B). $P = 0.0006$ (*wt* vs. uninfected), $P = 0.00005649$ (*Cgyps1-11Δ* vs. uninfected), $P = 0.00001361$ (*Cgyps1-11Δ* vs. *wt*), $P = 0.0029$ (*wt*-CK-666 vs. *wt*-DMSO), $P = 0.0002$ (*Cgyps1-11Δ*-CK-666 vs. *Cgyps1-11Δ*-DMSO) in (C). $P = 0.003$ (*Cgyps1-11Δ*-Control siRNA vs. *wt*-Control siRNA), $P = 0.0137$ (*Cgyps1-11Δ*-Arpc1B siRNA vs. *Cgyps1-11Δ*-Control siRNA) in (D). $P = 0.00000296$ (*Cgyps1-11Δ*-Control siRNA vs. *wt*-Control siRNA), $P = 0.0003$ (*Cgyps1-11Δ*-Arpc1B siRNA vs. *Cgyps1-11Δ*-Control siRNA), $P = 0.00006937$ (*Cgyps1-11Δ*-Arpc1B siRNA vs. *wt*-Arpc1B siRNA) in (E). $P = 0.0338$ (*wt*-Control siRNA vs. uninfected-Control siRNA), $P = 0.0348$ (*Cgyps1-11Δ*-Control siRNA vs. uninfected-Control siRNA), $P = 0.0012$ (*Cgyps1-11Δ*-Control siRNA vs. *wt*-Control siRNA), $P = 0.001$ (*Cgyps1-11Δ*-Arpc1B siRNA vs. *Cgyps1-11Δ*-Control siRNA) in (F). $P = 0.00004706$ (*wt*-Control siRNA vs. uninfected-Control siRNA), $P = 0.0036$ (*Cgyps1-11Δ*-Control siRNA vs. uninfected-Control siRNA), $P = 0.00001276$ (*Cgyps1-11Δ*-Control siRNA vs. *wt*-Control siRNA), $P = 0.0013$ (uninfected-Arpc1B siRNA vs. uninfected-Control siRNA), $P = 0.0004$ (*wt*-Arpc1B siRNA vs. *wt*-Control siRNA), $P = 0.0018$ (*Cgyps1-11Δ*-Arpc1B siRNA vs. *Cgyps1-11Δ*-Control siRNA) in (G). $P = 0.0008$ (*Cgyps1-11Δ*-NTC vs. *wt*-NTC), $P = 0.0007$ (*wt*-$p38^{-/-}$ vs. *wt*-NTC), $P = 0.014$ (*Cgyps1-11Δ*-$p38^{-/-}$ vs. *Cgyps1-11Δ*-NTC) in (H). $P = 0.0017$ (*Cgyps1-11Δ*-NTC vs. *wt*-NTC), $P = 0.0063$ (*Cgyps1-11Δ*-$Arpc1B^{-/-}$ vs. *Cgyps1-11Δ*-NTC) in (I). $P = 0.00000702$ (*Cgyps1-11Δ*-NTC vs. *wt*-NTC), $P = 0.0105$ (*Cgyps1-11Δ*-$p38^{-/-}$ vs. *wt*-$p38^{-/-}$), $P = 0.00000981$ (*Cgyps1-11Δ*-$p38^{-/-}$ vs. *Cgyps1-11Δ*-NTC) in (J). $P = 0.00000009$ (*Cgyps1-11Δ*-NTC vs. *wt*-NTC), $P = 0.0009$ (*Cgyps1-11Δ*-$Arpc1B^{-/-}$ vs. *wt*-$Arpc1B^{-/-}$), $P = 0.0003$ (*Cgyps1-11Δ*-$Arpc1B^{-/-}$ vs. *Cgyps1-11Δ*-NTC) in (K). $P = 0.000005318$ (*wt*-NTC vs. uninfected-NTC), $P = 0.022$ (*Cgyps1-11Δ*-NTC vs. uninfected-NTC), $P = 0.0003$ (*Cgyps1-11Δ*-NTC vs. *wt*-NTC), $P = 0.001$ (uninfected-$p38^{-/-}$ vs. uninfected-NTC), $P = 0.0056$ (*Cgyps1-11Δ*-$p38^{-/-}$ vs. *Cgyps1-11Δ*-NTC), $P = 0.0035$ (*wt*-$p38^{-/-}$ vs. uninfected-$p38^{-/-}$), $P = 0.0298$ (*Cgyps1-11Δ*-$p38^{-/-}$ vs. *wt*-$p38^{-/-}$) in (L). $P = 0.0000055$ (*wt*-NTC vs. uninfected-NTC), $P = 0.0005$ (*Cgyps1-11Δ*-NTC vs. uninfected-NTC), $P = 0.00000241$ (*Cgyps1-11Δ*-NTC vs. *wt*-NTC), $P = 0.0011$ (uninfected-$Arpc1B^{-/-}$ vs. uninfected-NTC), $P = 0.00000292$ (*Cgyps1-11Δ*-$Arpc1B^{-/-}$ vs. *Cgyps1-11Δ*-NTC), $P = 0.004$ (*wt*-$Arpc1B^{-/-}$ vs. uninfected-$Arpc1B^{-/-}$), $P = 0.0095$ (*Cgyps1-11Δ*-$Arpc1B^{-/-}$ vs. uninfected-$Arpc1B^{-/-}$) in (M). Source data are available online for this figure.

EC interaction (Pekmezovic et al, 2021; Rasheed et al, 2020a; Timmermans et al, 2018). Further, *Cg* infection influences cellular signalling, with the cell wall β-glucan stimulating phosphorylation of the EC-pattern recognition receptor EphA2 in oral ECs (Swidergall et al, 2017). Similarly, TLR2 and lactosylceramide receptor activated NF-κB signalling in tracheal and oral ECs, respectively, upon *Cg* infection (Li and Dongari-Bagtzoglou, 2009; Zhang et al, 2016). Herein, we report a new *Cg* target viz., Arpc1B, which acts upstream of p38 and NF-κB signalling to regulate the EC response.

Recruitment of innate immune cells to the site of microbial infection is an exquisitely-regulated process that banks upon co-ordinated communication between different host cell types (Rivera et al, 2016). *Cg* resides on host mucosae and traverses the EC barrier during bloodstream infections (Healey et al, 2017; Gouba and Drancourt, 2015; Richard and Sokol, 2019). However, the underlying mechanism remains unknown. Our work provides the first evidence on how *Cg* leverages CgYapsins to manipulate the actin network, induce its endocytosis and prevent the inflammatory reprogramming in ECs. Whether CgYapsins target Arpc1B at the EC membrane remains to be determined, however, Arpc1B downregulation within 30 min of *Cg* infection (Fig. 5F), and upon incubation with the proteolytically-active CgYps1 (Fig. EV5B), renders this possibility likely. However, it is also possible that the secreted forms of CgYapsins mediate Arcp1B degradation, soon after *Cg* contact with ECs. In this context, it is worth noting that CgYapsins have recently been detected in *Cg* secretome and biofilm matrix proteome, but the secreted proteolytic activity is yet to be reported in *Cg* (Rasheed et al, 2020b; Gonçalves et al, 2021).

Alternatively, in response to *Cg*-induced mechanical stress, Arp2/3 complex, by altering actin cortex and plasma membrane association, may aid in the formation of actin-driven membrane protrusions/extensions/invaginations, that may bring CgYapsins and Arpc1B in close proximity. Notably, Arp2/3 complex-mediated actin polymerization has been implicated in maintaining plasma membrane shape homeostasis in response to membrane deformations (Quiroga et al, 2023), and Arc5 (Arp2/3 complex subunit) has been found to be localized at the cell periphery (Kahr et al, 2017). However, further studies are warranted to test these possibilities for *Cg*-EC interplay.

Further, since EC-secreted IL-8 promoted neutrophil migration towards *Cgyps1-11Δ*-infected A-498, we speculate that *Cg* has evolved a multi-component system, with CgYapsins representing one component of the system, to suppress production of the neutrophil chemotactic factor IL-8 in ECs. In this context, it is noteworthy that the recruitment of monocytes, and not neutrophils, is the predominant host response to *Cg* infections (Jacobsen et al, 2010), with *Cg* replicating and getting killed in macrophages and neutrophils, respectively (Duggan et al, 2015; Kaur et al, 2007).

NF-κB is a crucial stress-responsive transcriptional regulator that drives many processes including inflammation and immune gene expression (Zhang et al, 2017). NF-κB, a dimer of different combinations of five subunits, p65/Rel A, c-Rel, Rel B, p50 and p52, exists in the cytoplasm in its inactive form (Zhang et al, 2017). External stimuli release it from the inhibitory proteins including IκB (inhibitor of κB), leading to its translocation to the nucleus wherein NF-κB governs transcription of κB binding site-containing genes (Zhang et al, 2017). IL-6 and IL-8 are two targets of NF-κB

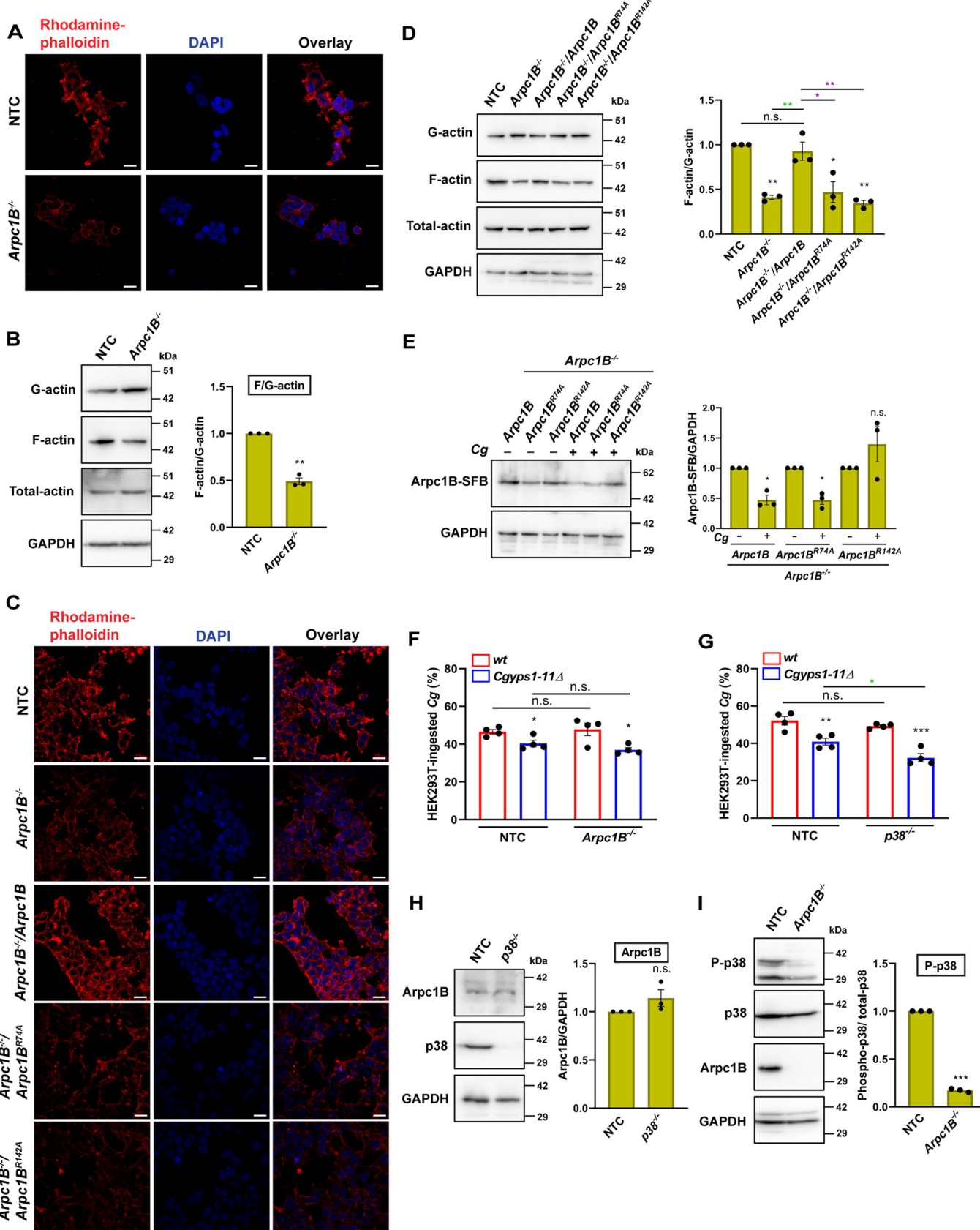

**Figure 7. Arpc1B loss inhibits F-actin assembly and p38 activation.**

(A) Representative confocal micrographs illustrating rhodamine-phalloidin-stained actin structures in $Arpc1B^{-/-}$ and NTC ($Arpc1B^{+/+}$)-HEK293T cells. Representative z-stack images were obtained using the confocal microscope (Leica SP8) with 63X/1.44 NA objective lens. (B) Representative immunoblots illustrating GAPDH-normalized globular (G)-actin and filamentous (F)-actin levels in $Arpc1B^{-/-}$ and NTC-HEK293T cells. The asterisks mark F/G actin ratio differences, as compared to NTC-HEK293T cells (taken as 1.0). $n = 3$ biological replicates. (C) Representative confocal micrographs illustrating rhodamine-phalloidin-stained actin structures in HEK293T-NTC ($Arpc1B^{+/+}$) and $Arpc1B^{-/-}$ cells expressing *wild-type* Arpc1B-SFB or mutant Arpc1B-SFB ($Arpc1B^{R74A}$ and $Arpc1B^{R142A}$) proteins. (D) Representative immunoblots illustrating Gapdh-normalized G-actin and F-actin levels in indicated HEK293T cells. Black asterisks indicate F/G actin ratio differences, as compared to NTC-HEK293T cells (taken as 1.0). $n = 3$ biological replicates. (E) Representative immunoblots illustrating Gapdh-normalized Arpc1B-SFB levels in uninfected or *wt*-infected indicated HEK293T cells at 6 hpi. Black asterisks denote Arpc1B level differences, as compared to the corresponding uninfected HEK293T cells (taken as 1.0). $n = 3$ biological replicates. (F, G) CFU-based internalization analysis of *wt* and *Cgyps1-11Δ* strains in $Arpc1B^{-/-}$ HEK293T (F) and $p38^{-/-}$ HEK293T cells (G), after 4 h co-incubation. Asterisks mark ingestion differences, as compared to *wt*-infected cells. $n = 4$ biological replicates in (F, G). (H) Representative immunoblots illustrating Arpc1B levels in $p38^{-/-}$ and NTC-HEK293T cells. $n = 3$ biological replicates. (I) Representative immunoblots illustrating diminished phospho-p38 levels in $Arpc1B^{-/-}$HEK293T cells. Data represent fold-change in phosphorylated-p38 protein levels in $Arpc1B^{-/-}$ cells, compared to NTC-HEK293T cells (considered as 1.0). $n = 3$ biological replicates. Data information: In (B, D–I), data are presented as mean ± SEM. *$P < 0.05$; **$P < 0.01$; ***$P < 0.001$; ****$P < 0.0001$; n.s. not significant. Unpaired or paired two-tailed Student's $t$ test in (B, D–I). $P = 0.0049$ ($Arpc1B^{-/-}$ vs. NTC) in (B). $P = 0.0016$ ($Arpc1B^{-/-}$ vs. NTC), $P = 0.0442$ ($Arpc1B^{-/-}/Arpc1B^{R74A}$ vs. NTC), $P = 0.002$ ($Arpc1B^{-/-}/Arpc1B^{R142A}$ vs. NTC), $P = 0.008$ ($Arpc1B^{-/-}/Arpc1B$ vs. $Arpc1B^{-/-}$), $P = 0.04$ ($Arpc1B^{-/-}/Arpc1B^{R74A}$ vs. $Arpc1B^{-/-}/Arpc1B$), $P = 0.005$ ($Arpc1B^{-/-}/Arpc1B^{R142A}$ vs. $Arpc1B^{-/-}/Arpc1B$) in (D). $P = 0.0215$ (*wt*-$Arpc1B^{-/-}/Arpc1B$ vs. uninfected-$Arpc1B^{-/-}/Arpc1B$), $P = 0.0188$ (*wt*-$Arpc1B^{-/-}/Arpc1B^{R74A}$ vs. uninfected-$Arpc1B^{-/-}/Arpc1B^{R74A}$) in (E). $P = 0.022$ (*Cgyps1-11Δ*-NTC vs. *wt*-NTC), $P = 0.019$ (*Cgyps1-11Δ*-$Arpc1B^{-/-}$ vs. *wt*-$Arpc1B^{-/-}$) in (F). $P = 0.0088$ (*Cgyps1-11Δ*-NTC vs. *wt*-NTC), $P = 0.0003$ (*Cgyps1-11Δ*-$p38^{-/-}$ vs. *wt*-$p38^{-/-}$), $P = 0.0232$ (*Cgyps1-11Δ*-$p38^{-/-}$ vs. *Cgyps1-11Δ*-NTC) in (G). $P = 0.0002$ ($Arpc1B^{-/-}$ vs. NTC) in (I). Scale bar = 20 μm in (A, C). Source data are available online for this figure.

(Zhang et al, 2017). We observed an increase in IL-6 and IL-8 secretion in *Cgyps1-11Δ*-infected and a decrease in IL-6 and IL-8 secretion in *wt*-infected A-498, compared to uninfected A-498, from 6 h onwards post-infection (Fig. EV5C). This reduced cytokine secretion, in *wt*-infected A-498, despite increased p65 nuclear localization (Appendix Fig. S1C), could probably be in part due to CgYapsins-mediated cytokine degradation in *wt*-infected A-498, as IL-8 levels were diminished upon co-incubation with the catalytically-active CgYps1 (Fig. EV5D), and with *wt* cells, but not with *Cgyps1-11Δ* cells (Fig. EV5E). These data indicate that *Cg*, via CgYapsins, probably targets two components of the inflammatory response, IL-8 and p38 MAPK (via Arpc1B), in ECs (Fig. 8C). In this regard, it is worth noting that the pathogen-mediated NF-κB and p38 activation modulation is not uncommon, as *Hemophilus influenzae* activated NF-κB via TLR-2 in ECs (Shuto et al, 2001), while the enteropathogenic *E. coli* effector NleC suppressed IL-8 secretion by inhibiting p38 and NF-κB activation in intestinal ECs (Sham et al, 2011). *Cg*-stimulated GM-CSF induction in oral ECs has also been reported to be dependent on adhesion and NF-κB activation (Li and Dongari-Bagtzoglou, 2009).

Arpc1B, a WD-repeat-containing protein, is pivotal to formation and maintenance of the actin branching-regulatory Arp2/3 complex, with Arp2/3 complex-formed mechanoresponsive branched actin networks at the membrane providing strength to the cells (Papalazarou and Machesky, 2021). Arpc1B has an isoform, Arpc1A, with both isoforms showing 67% identity (Abella et al, 2016). However, compared to Arpc1A-containing Arp2/3, Arpc1B-containing Arp2/3 complexes facilitate actin assembly better, and the formed-branched actin networks disassemble slowly (Abella et al, 2016), thereby underscoring unique Arpc1B functions. Although other Arp2/3 complex subunits were not identified as CgYps1 and CgYps7 interactors, we propose that the Arp2/3 complex per se may be crucial for the *Cg*-invoked EC response, as treatment with CK-666, which binds and stabilizes the inactive state of the Arp2/3 complex (Hetrick et al, 2013), rescued elevated cytokine secretion in *Cgyps1-11Δ*-infected A-498 (Fig. 6B,C). Furthermore, since CgYps1-interacting residues in Arpc1B, Arg-

74 and Arg-142, are conserved between Arpc1A and Arpc1B isoforms, it will be interesting to determine their requirement for Arpc1A functions in actin assembly.

Further, our findings raise a compelling question as to why CK-666 and Arpc1B loss reduces *Cgyps1-11Δ* adherence but not internalization. Since adhesins are likely to mediate *Cg* adherence to ECs through specific host receptors, it is possible that Arpc1B knockdown affects actin nucleation underneath the plasma membrane (PM), leading to perturbed cell membrane structures. This may impact the distribution/localization of adhesin receptors at the PM which could adversely affect the adherence of *Cgyps1-11Δ* cells that contain high amounts of cell surface adhesins. In this context, it is noteworthy that Arp2/3 complex-mediated branched actin networks restrain and/or regulate diffusional barriers for PM proteins and receptors, with Arp2/3 complex activity also being implicated in governing B cell responses specifically to the spatially-confined membrane-bound antigens (Mattila et al, 2016; Bolger-Munro et al, 2019). Based on our data, we propose that a single mechanism may not govern *Cg* adherence, internalization, survival and modulation of host signalling pathways, rather it is a combination of multiple regulators that work in concert to control the outcome of *Cg* infections. This notwithstanding, our findings do establish Arpc1B and p38 MAPK as key players in *Cg*-EC interplay.

How Arp2/3 complex modulates the EC response to *Cg* remains to be deciphered fully. However, bacterial and viral pathogens are known to target the Arp2/3 complex to promote actin polymerization either from outside the cell or after entering into the cytoplasm, to promote their attachment, uptake, and/or cell-cell spread (Welch and Way, 2013). Recently, the Arp2/3 complex has been implicated in formation of the actin cocoon that facilitates *Shigella* dissemination (Kühn et al, 2020). Similarly, *Cg* may manipulate Arpc1B to limit actin arrangements to facilitate its persistence as well as impede the neutrophil recruitment. Notably, the impaired Arp2/3 actin filament branching, due to Arpc1B deficiency, is associated with platelet abnormalities, recurrent invasive infections, inflammatory bowel disease, and T-cell cytoskeletal defects (Kahr et al, 2017; Brigida et al, 2018). Furthermore, Arpc1B deficiency, due to a homozygous complex

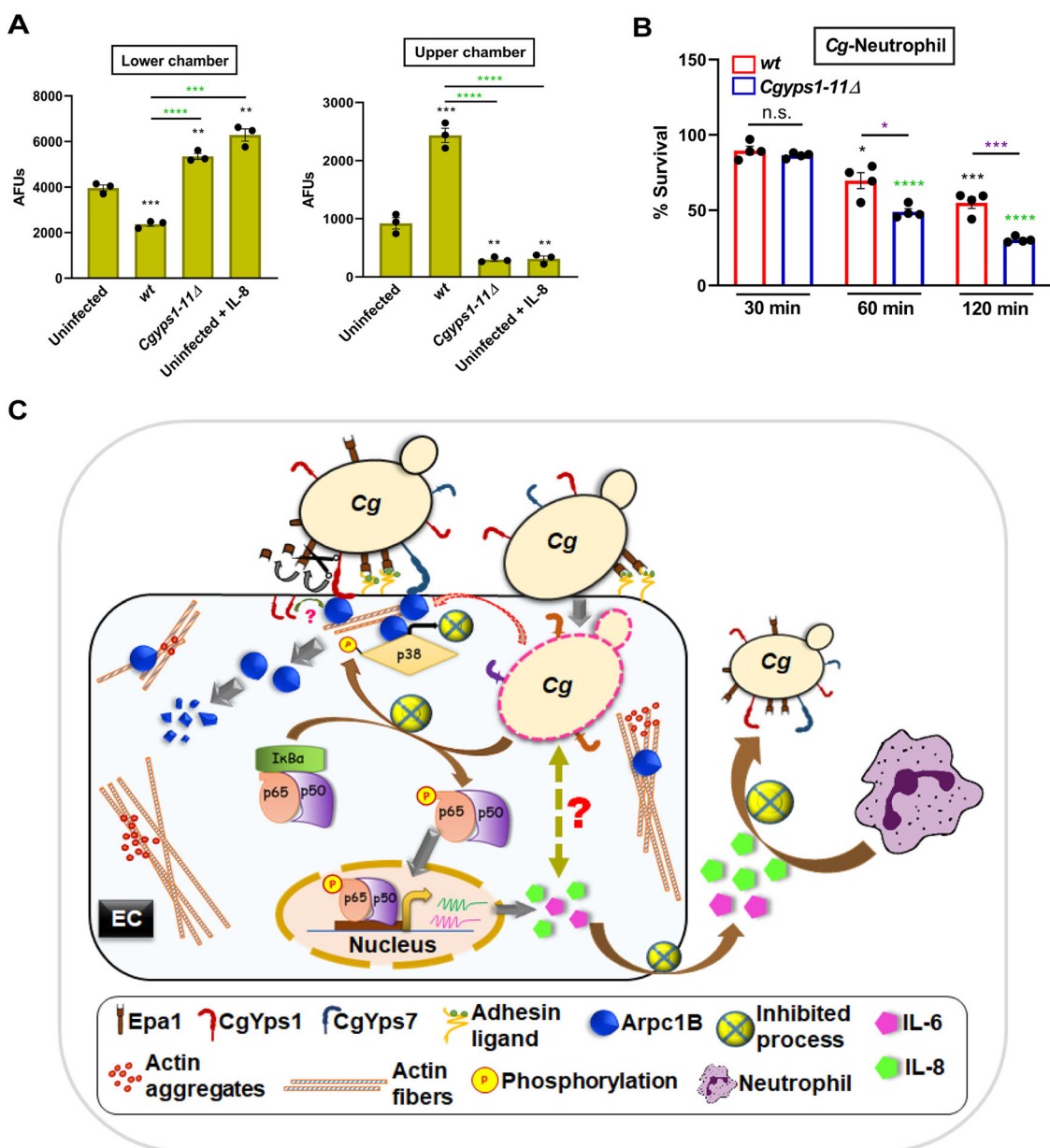

frameshift mutation in a combined immunodeficiency patient, led to F-actin polymerization defect and impeded neutrophil movement (Kuijpers et al, 2017), thereby suggesting that Arpc1B may govern neutrophil functions via many ways. However, it is also possible that the impaired neutrophil recruitment is one of the facet that Arpc1B governs, and that, Arpc1B contributions to Arp2/3 complex-mediated actin polymerization may play a more critical role in controlling *Cg* infections by regulating other cellular processes including cell mobility and endocytosis. Lastly, since Arpc1B loss is likely to result in immune dysregulation at multiple levels, *Cg* targeting of Arpc1B could be a clever strategy against the host defense.

In conclusion, we show how CgYapsins dampen the EC inflammatory response by modulating p38 signalling, via Arpc1B, to restrict the neutrophil migration.

## Methods

### Ethics statement

Mice infection experiments were conducted at Experimental Animal Facility of Centre for DNA Fingerprinting and Diagnostics (CDFD; www.cdfd.org.in), Hyderabad, India in accordance with the guidelines of the Committee for the Purpose of Control and Supervision of Experiments on Animals, Government of India. Procedures were designed to minimize animal suffering, and approved by the Institutional Animal Ethics Committee of CDFD (EAF/RK/22/2023). For neutrophil isolation experiments, informed signed consent was obtained from healthy volunteer donors, prior to the collection of blood samples in EDTA. This protocol involving human participants was approved by Institutional Bioethics Committee of CDFD (IEC 21/2015).

**Figure 8.   *Cg* infection of A-498 cells inhibits neutrophil migration.**

(A) Neutrophil migration assay. The migration of FITC-conjugated anti-CD66b-labelled human neutrophils towards *Cg*-infected A-498 cells was measured by recording the fluorescence of the cells present in the upper and the lower chamber of the transwell filter after 2 h co-culture. Uninfected, *Cg wt/Cgyps1-11Δ*-infected or IL-8 (3 ng)-treated A-498 cells were incubated for 24 h in the lower chamber, prior to co-culturing with neutrophils. Data are expressed as Arbitrary Fluorescent Units (AFUs). Black asterisks indicate differences in neutrophil movement in indicated conditions, as compared to uninfected A-498 cells. $n = 3$ biological replicates. (B) Neutrophil-mediated *Cg* killing. Data represent *wt*- and *Cgyps1-11Δ* CFUs after incubation with neutrophils at indicated time points. Black and green asterisks indicate a decrease in viable cells, as compared to 30 min CFUs of *wt* and *Cgyps1-11Δ* strains, respectively. $n = 4$ biological replicates. (C) A schematic summarizing key findings of the study. *Candida glabrata* (*Cg*) adheres to epithelial cells (EC) via Epa1 adhesin, and is internalized by ECs. Putative cell surface-associated aspartyl proteases, CgYps1 and CgYps7, interact with Arpc1B, leading to its degradation and filamentous actin disassembly. *Cg* internalization reduces p38 MAPK phosphorylation, decreases p38-Arpc1B interaction, and impedes IL-6 and IL-8 secretion. Diminished IL-8 levels inhibit neutrophil movement towards *Cg*, that protects *Cg* from neutrophil-mediated killing. Of note, CgYapsins cleave Epa1 adhesin off the *Cg* cell wall in laboratory growth conditions, degrade IL-8 in vitro and are pivotal determinants of *Cg*-EC interaction in the cell culture model, as their deletion deregulated the above-mentioned EC response, and led to elevated *Cg* killing by neutrophils. Data information: In (A, B), data are presented as mean ± SEM. *$P < 0.05$; **$P < 0.01$; ***$P < 0.001$; ****$P < 0.0001$; n.s. not significant. Unpaired two-tailed Student's *t* test in (A, B). $P = 0.0006$ (*wt* vs. uninfected), $P = 0.0019$ (*Cgyps1-11Δ* vs. uninfected), $P = 0.0015$ (uninfected, IL-8-treated vs. uninfected, untreated), $P = 0.00004960$ (*Cgyps1-11Δ* vs. *wt*), $P = 0.0002$ (*wt* vs. uninfected, IL-8-treated) in lower chamber in (A). $P = 0.0007$ (*wt* vs. uninfected), $P = 0.0034$ (*Cgyps1-11Δ* vs. uninfected), $P = 0.0049$ (uninfected, IL-8-treated vs. uninfected, untreated), $P = 0.00007254$ (*Cgyps1-11Δ* vs. *wt*), $P = 0.00009010$ (*wt* vs. uninfected, IL-8 treated) in upper chamber in (A). $P = 0.0157$ (*wt*-60 min vs. *wt*-30 min), $P = 0.0003$ (*wt*-120 min vs. *wt*-30 min), $P = 0.0000038$ (*Cgyps1-11Δ*-60 min vs. *Cgyps1-11Δ*-30 min), $P = 0.00000001$ (*Cgyps1-11Δ*-120 min vs. *Cgyps1-11Δ*-30 min), $P = 0.0109$ (*Cgyps1-11Δ*- 60 min vs. *wt*-60 min), $P = 0.0007$ (*Cgyps1-11Δ* -120 min vs. *wt*-120 min) in (B). Source data are available online for this figure.

## Mice housing

All animal experiments were performed with 6–8 weeks-old female BALB/c mice. Mice were housed and bred at Experimental Animal Facility of CDFD, Hyderabad, India. Animals were housed under specific pathogen-free conditions in individually-ventilated cages in a group of maximum of five mice under circadian rhythm (automatically-controlled 12 h light and 12 h dark cycle) at the ambient temperature of 22 °C and humidity of 40% ± 5%. Food and drinking water were supplied ad libitum.

## Strains, plasmids and growth conditions

*C. glabrata wild-type* (*wt*; *BG2* strain) and mutant (*BG2* derivatives) strains were maintained on yeast extract-peptone-dextrose (YPD) medium or minimal yeast nitrogen base containing Casamino acid (CAA) medium at 30 °C. *C. albicans* was cultured on YPD medium at 30 °C. The *Escherichia coli* DH5α strain was used for gene cloning, and grown in LB medium at 37 °C.

For complementation studies, *CgYPS1*, *CgYPS7* and *CgYPS1^{D91A}* genes were cloned in SpeI and XmaI, XbaI and XhoI, and SpeI and XmaI sites, respectively, in the pRK999 (pCU-*PDC1*) plasmid, as described previously (Askari et al, 2022; Battu et al, 2021). *CgYPS1^{D378A}* gene was cloned in SpeI and XmaI sites in the pRK999 plasmid. For mammalian gene overexpression and complementation studies, Arpc1B (1.19 kb; amplified using A-498 cDNA as template) was cloned in attP and attB sites in the SFB (S protein-FLAG-Streptavidin-binding peptide)-tagged destination vector using Gateway cloning system (Invitrogen). Arginine-74 and Arginine-142 amino acids in Arpc1B-SFB were replaced with alanine via PCR-amplification of Arpc1B-SFB sequence with mutagenic primers, followed by DpnI-digestion and rolling circle replication of self-ligated fragments in *E. coli*. The strains and plasmids, and primers, antibodies, cytokines and inhibitors, used in this study are listed in Appendix Tables S1 and S2, respectively.

## Cell culture

Human kidney epithelial A-498 (ATCC #HTB-44), gastric epithelial AGS (ATCC #CRL-1739), embryonic kidney HEK293T (ATCC #CRL-3216) and colon epithelial Caco-2 (ATCC #HTB-37) cells were

maintained in Minimal Essential Medium (MEM), F-12K Medium (Kaighn's Modification of Ham's F-12 Medium), Dulbecco's Modified Eagle Medium (DMEM) and MEM, respectively, at 37 °C in a humidified incubator with 5% $CO_2$. All media contained heat-inactivated 10% fetal bovine serum (FBS) and 1% penicillin and 1% streptomycin, unless indicated otherwise. Caco-2 cells were grown in MEM medium containing 20% FBS. All cell lines were mycoplasma free.

For siRNA (small double-stranded interfering RNAs)-mediated Arpc1B gene silencing, A-498 cells were transfected with the human Arpc1B siRNA (10 μM) or the control siRNA (10 μM) (Santa Cruz Biotechnology Inc., Dallas, Texas, USA) in MEM incomplete medium (lacks antibiotics and FBS) using lipofectamine 2000 (Invitrogen Thermo Fisher Scientific Inc., Waltham, MA, USA), and incubated at 37 °C in 5% $CO_2$ chamber. After 36–48 h, Arpc1B-siRNA knockdown cells (70–80% reduction in Arpc1B expression) were collected and used for immunoblot, immuno-fluorescence and *Cg* infection analyses.

## Knockout cell line generation

CRISPR-Cas9 system was used to generate knockout p38 and Arpc1B cell lines. Briefly, CRISPR primers/paired guide RNAs were designed using the Benchling server (www.benchling.com). For each knockout cell-line generation, two sets of paired sgRNAs (single-guide RNAs) were annealed and cloned in BbsI site in the PX458 (pSpCas9(BB)-2A-GFP) plasmid, followed by clone confirmation via sequencing. For *p38* and *Arpc1B* knockout generation, exons 7 and 8, respectively, were targeted in HEK293T. gRNA-expressing plasmids pSpCas9(BB)-2A-GFP (1 μg each) were co-introduced into ~70% confluent HEK293T cells using polyethylenimine (PEI) transfecting reagent (Polysciences, Inc., Warrington, PA, USA), following manufacturer's instructions, and incubated for 36–48 h. After GFP expression-based cell sorting using FACSAria™ III (BD Biosciences, San Jose, CA, USA), single cells were placed and grown for 48 h in a 96-well plate. Non-transfected cells were used as control for the background fluorescence signal. The knockout cell lines were confirmed by sequencing and Western blot analysis.

## Adherence measurement

A-498, Caco-2, AGS, HEK293T, Arpc1B^{−/−} and p38^{−/−} cells were seeded at a density of $2 \times 10^5$, $2 \times 10^5$, $5 \times 10^5$, $1 \times 10^6$, $1 \times 10^6$ and

$1 \times 10^6$ cells per well of a 24-well culture plate, respectively. After 12 h incubation, the medium supernatant was discarded, and cells were washed thrice with PBS. Next, epithelial cells (ECs) were fixed in 3.7% formaldehyde for 15 min, followed by two PBS washes. To determine *Candida* adherence to Arpc1B-knockdown cells, A-498 cells were transfected with control-siRNA or Arpc1B-siRNA in MEM incomplete medium, and incubated for 24–36 h at 37 °C. Thereafter, cells were fixed in 3.7% formaldehyde for 15 min. For inhibitor studies, all inhibitors were prepared in DMSO.

For adherence analysis, *Cg* strains was grown in YPD medium for 16–20 h at 30 °C, followed by culturing in YPD medium containing 100 µCi of $S^{35}$ [in vivo Pro Twinlabel mix (JONAKI, Hyderabad, India #LCS-8) or Met-35S-Label, American Radiolabeled Chemicals, Inc., St. Louis, MO, USA #ARS0110] for 16–20 h at 30 °C. The radiolabelled *Cg* cells were collected, washed with PBS, suspended in PBS, and 200 µl cell suspension was added to fixed epithelial cells to achieve a MoI (multiplicity of infection) of 1:2. After 2 h incubation at room temperature, each well was washed thrice with sterile PBS to remove non-adherent *Cg*, followed by epithelial cell lysis in 5% SDS. Lysates were transferred to tubes containing Optiphase scintillation fluid, and radioactive counts were recorded. For input samples, radioactive counts in $S^{35}$-labeled *Cg* cell suspension (200 µl) were measured. The percentage adherence was determined by dividing the output values by input values, and multiplying the number by 100.

For CFU-based adherence analysis, fixed cells were infected with *Cg* at MoI of 1:2. After 2 h incubation at room temperature, cells were washed thrice with sterile PBS to remove non-adherent *Cg*. Cells were collected in 1% SDS by scraping them from the wells, followed by plating of appropriate dilutions of cell suspensions on YPD medium. After 2 days of incubation at 30 °C, *Cg* colonies were counted. The percentage adherence was determined by dividing the 2 h *Cg* CFUs by 0 h CFUs (represent the number of cells infected to ECs). For lactose blocking experiment, *Cg* cells were treated with 10 mM lactose for 30 min at 30 °C, prior to infection. For dynasore (Dynamin inhibitor) treatment, A-498 cells were treated either with DMSO (0.05%; solvent control) or dynasore (50 µM) for 1 h, prior to formaldehyde-fixation.

### *Cg* endocytosis and intracellular replication analysis

Overnight YPD medium-grown *Cg* cells were added to ECs seeded in a 24-well plate at a MoI of 10:1, and incubated at 37 °C. At 4 h post-infection, epithelial cells were washed thrice with PBS to remove non-adherent extracellular *Cg*, and EC-*Cg* coincubation was continued for another 20 h in fresh pre-warmed MEM medium. At 4 and 24 h, ECs were washed thrice with PBS and lysed in 1 ml sterile water, followed by plating of appropriate cell lysate dilutions on YPD medium. *Cg* colonies were counted after 2 days of incubation at 30 °C. Intracellular replication was determined by dividing the 24 h CFUs by 4 h CFUs. For *Cg* ingestion analysis, the internalization percentage was calculated by dividing the CFUs recovered at 4 h by the 0 h CFUs (*Cg* cells used for EC infection), and multiplying the number by 100.

For inhibitor studies, ECs were pre-treated with DMSO, dynasore (50 µM), SB203580 (p38 MAPK inhibitor; 10 µM), BAY 11-7082 (NF-κB inhibitor; 10 µM), cytochalasin D (actin polymerization inhibitor; 5 µM) and CK-666 (Arp2/3 complex inhibitor; 1 µM) for 1 or 2 h, 1 h, 1 h, 2 h, 1 h and 1 h, respectively, prior to *Cg*

infection, and the infection was continued in the presence of the inhibitor. For lactose treatment, *Cg* cells were pre-treated with 10 mM lactose for 30 min at 30 °C, and infected to ECs in the lactose (10 mM)-containing medium. To study the Arpc1B knockdown effect on *Cg* intracellular replication, A-498 cells were transfected with control or Arpc1B-siRNA and incubated for 36–48 h, followed by *Cg* infection at a MoI of 10:1.

### Microscopy analysis

For inside-outside staining, A-498 cells were seeded in a four-chambered well glass slide at a density of $1 \times 10^5$, followed by overnight incubation at 37 °C in 5% $CO_2$ chamber. Adhered ECs were infected with mCherry-tagged *Cg wt* or *Cgyps1-11Δ* cells at a MoI of 10:1. After 2 h, 4 h and 24 h co-incubation, slides were washed thrice with sterile PBS to remove unbound *Cg*, followed by cell fixation in 3.7% formaldehyde for 15 min at room temperature. Glass slide chambers were washed twice with PBS, and cells were stained with concanavalin A-FITC (fluorescein isothiocyanate) stain (25 µg/ml) for 15–20 min at 37 °C. Of note, concanavalin A is known to bind to mannose residues in glycoproteins and glycolipids of cell membranes and cell walls, thereby staining both the mammalian cell membrane and mannans in the fungal cell wall. After PBS washes, the chambers were separated from the slide using the chamber removal device. The slide was air-dried, and visualized in the 4′, 6-diamidino-2-phenylindole (DAPI)-containing vectashield mounting medium, using the super resolution microscope (Elyra 7) or Leica SP8 confocal laser scanning microscope. The number of extracellular (cells expressing mCherry red signal, with the peripheral green signal from the concanavalin A-FITC stain) and intracellular (cells expressing mCherry red signal only) *Cg* were counted, and plotted. A minimum of 200 cells were counted for each condition.

For actin filament staining by rhodamine-phalloidin, A-498 and HEK293T cells were either infected with FITC-labelled *wt, Cgyps1-11Δ* or *C. albicans* cells at a MoI of 1:1, or left uninfected, in a four-chambered glass slide. To examine the effect of Arginine-74 and Arginine-142 residues in Arpc1B on actin filamentation, *Arpc1B^{−/−}* HEK293T cells were transfected with a plasmid that expresses either *wild-type* Arpc1B or mutant Arpc1B (Arpc1B^{R74A} and Arpc1B^{R142A}) using PEI, and incubated for 12–16 h. For FITC staining of *Candida*, overnight YPD medium-grown cells (1.0 OD_{600}) cells were incubated with 25 µg/ml FITC for 30 min at 37 °C. After 4 h of *Candida*-A-498 co-incubation, extracellular yeasts were removed with PBS washes, and the incubation was continued for another 2 h, followed by PBS washes and cell fixation in 3.7% formaldehyde for 15 min at room temperature. Next, cells were permeabilized with 0.5% Triton-X for 5–10 min at room temperature, washed with PBS, and blocked with 5% bovine serum albumin (BSA) for 1–2 h at room temperature, followed by rhodamine-phallodin (1:500 dilution prepared in 5% BSA) staining for 45 min at 37 °C in dark, and visualization in the DAPI-containing vectashield medium using the Leica SP8 confocal microscope.

For immunofluorescence, prior to *Cg* infection, A-498 cells were transfected with control-siRNA or Arpc1B-siRNA for 24–36 h, or pre-treated with DMSO, SB203580 (10 µM) or CK-666 (1 µM) for 1 h. To detect the endogenous p65, endogenous Arpc1B, endogenous p38 or ectopically expressed Arpc1B-SFB, uninfected or mCherry-tagged *Cg*-infected A-498 cells were fixed in 3.7% formaldehyde, followed by cell permeabilization in 0.5% Triton-

X-100. After blocking with 5% BSA, cells were incubated with anti-p65, anti-Arpc1B, anti-p38 and anti-Flag antibodies at 4 °C for overnight, washed and further incubated with Alexa Flour-488-conjugated anti-rabbit or anti-mouse secondary antibodies at room temperature for 1 h. Cell nuclei were counterstained with DAPI (1 µg/ml; in PBS), and cells were imaged using the Leica-SP8 confocal microscope. Co-localization of p38 and Arpc1B was analysed using the LASX software.

## Histology analysis

For histopathology studies, two biological replicates of uninfected, *wt*-infected and *Cgyps1–11Δ*-infected 6–8-week old female, BALB/c mice were sacrificed after 1 day of infection, and kidneys were harvested and fixed in formalin solution for 24 h. Collected tissues were trimmed and processed in an automatic tissue processor (Leica Tissue Processor TP 1020), followed by paraffin wax embedding using the Tissue Embedding System (Leica Histocore Arcadia Embedding Center). Embedded tissues were further trimmed with the help of an automatic microtome (Microtome Thermoscientific HM 340E), and thin vertical sections of 3 µm were cut. These sections were placed on clean, grease-free glass slides and stained with hematoxylin and eosin for 1 min at room temperature, followed by washes in distilled water, and counterstaining with 1% eosin. Slides were washed, dehydrated, and mounted with cover slips for imaging. Stained tissue section images were acquired using an inverted microscope (Nikon ECLIPSE Ti2-U) with ×40 magnification.

For immunohistochemistry analysis, 3 µm tissue sections were obtained on pre-coated glass slides, and de-paraffinized using heat at 65 °C for 20 min. Slides were placed in xylene-containing coplin jars, followed by hydration through a graded series of isopropanol (100%, 75%, 50%) and slide rinsing with water. After quenching endogenous peroxidase activity with methanol and 3% $H_2O_2$, antigen retrieval was performed using the heat induced epitope retrieval method [boiling in citrate buffer (pH 6.0) for 10 min]. After incubating tissue sections in the blocking buffer (5% Fetal Bovine Serum in PBST) for 30 min at room temperature, anti-rabbit CD45 or anti-Ly6G antibody was added. After overnight incubation at 4 °C, samples were washed with PBS and incubated with HRP-conjugated anti-rabbit IgG secondary antibody for 30 min at room temperature. After PBS washes, the substrate diaminobenzidine (DAB) was added to each tissue section and incubated for 10–15 min at room temperature. Slides were counterstained with hematoxylin, dehydrated through the graded isopropanol series and mounted. Samples were visualized using the Nikon ECLIPSE Ti2-U inverted microscope with ×40 magnification.

## Protein extraction and immunoblotting

A-498 and HEK293T were seeded at a density of $2 \times 10^6$ and $2 \times 10^7$ cells, respectively, in a 100 mm dish, and incubated for overnight at 37 °C in a 5% $CO_2$ chamber. Adhered A-498 and HEK293T cells were infected with *wt* and *Cgyps1-11Δ* cells at a MoI of 1:1. To determine the requirement of Arginine-74 and Arginine-142 residues in Arpc1B for *Cg* survival, *Arpc1B*$^{−/−}$ HEK293T cells were transfected with a plasmid that expresses either Arpc1B or Arpc1B$^{R74A}$/Arpc1B$^{R142A}$ using PEI for 12–16 h, followed by *Cg*-infection at a MoI of 1:1. At desired time points, cells were washed, scraped in ice-cold PBS and were collected after centrifugation at

1500 rpm for 5 min at 4 °C. Next, cells were lysed in NETN buffer [20 mM Tris-HCl (pH 8.0), 100 mM NaCl, 1 mM EDTA, 0.5% Nonidet P-40] containing 1× phosphatase inhibitor (Roche Diagnostics, Germany) and 1× protease inhibitor (Roche Diagnostics, Germany), and incubated on ice for 30 min. Lysates were sonicated and pelleted down at 13,000 rpm for 15 min at 4 °C. Supernatants were collected and stored at −20 °C. Before analysis, samples were incubated in SDS-loading buffer at 95 °C for 15 min, and resolved on SDS-PAGE. Proteins were transferred to the PVDF membrane, and membranes were blocked in TBST [50 mM Tris (pH 7.4), 150 mM NaCl, 2.7 mM KCl and 0.1% w/v Tween-20] containing 5% BSA or 5% skimmed milk (w/v) for 1 h at room temperature, followed by membrane probing with primary and secondary antibodies, and protein detection using Enhanced ChemiLuminescence (ECL) kit (Cytiva LifeSciences, Massachusetts, USA). Representative blot images were assembled using Adobe Illustrator CC 2018. Of note, p38, p65, Arpc1B, Arpc1B-SFB, Erk1/2, Jnk1/2, rCgYps1, rCgYps7, GFP-SFB, GAPDH and actin proteins corresponded to ~38, 65, 37, 37, 42 & 44, 46 & 54, 50, 47, 42, 37 and 45 kDa sizes, respectively.

## F-actin and G-actin analysis

A-498 and HEK293T cells were seeded at a cell density of $2 \times 10^6$ and $10 \times 10^6$ per 100 mm dish, respectively, and cultured for 24 h, followed by infection with *wt*, *Cgyps1-11Δ* or *C. albicans* cells at a MoI of 1:1. Uninfected cells were used as control. For actin analysis in *Arpc1B*$^{−/−}$ HEK293T cells expressing either Arpc1B or Arpc1B$^{R74A}$/Arpc1B$^{R142A}$, cells were transfected with appropriate plasmids using PEI for 12–16 h. At 6 h post-infection, A-498 or HEK293T cells were washed thrice with PBS, and lysed in F-actin stabilizing lysis buffer [50 mM PIPES (pH 6.9), 50 mM NaCl, 5 mM $MgCl_2$, 5 mM EGTA, 5% glycerol, 0.1% Triton X-100, 0.1% Tween-20, 0.1% NP-40, 0.1% 2-mercaptoethanol, 1 mM ATP and 1× protease inhibitor cocktail] for 1 h on ice, followed by ultracentrifugation at 100,000×*g* for 1 h. The G-actin fraction-containing supernatant was collected, and the pellet was incubated for 1 h at 4 °C, after dissolving in the chilled water containing 10 µM cytochalasin D. The pellet was centrifuged at 13,000 rpm for 30 min, and the supernatant containing F-actin was collected. Both G-actin and F-actin fractions were analyzed, using anti-actin antibody, by Western blot analysis, and the F/G-actin ratio was calculated via the ImageJ software-based immunoblot densitometric analysis.

## CgYps1 and CgYps7 protein purification

CgYps1 and CgYps7 proteins were purified from the SDS-PAGE gel using gel elution method. Briefly, log-phase cultures of the *E. coli* BL21 (DE3) strain expressing 6× histidine-tagged CgYps1 or CgYps7 protein were induced with isopropyl β-D-1-thiogalactopyranoside (IPTG; 1 mM) for 3 h at 37 °C, followed by centrifugation at 5000 rpm for 10 min. The cell pellet was washed with PBS and lysed in lysis buffer (50 mM sodium phosphate, 300 mM NaCl, 0.1% β-mercaptoethanol, 1% Triton X-100 and 1 mM PMSF; pH 7.0) for 20 min on ice, followed by sonication at 40 Hz amplitude with a 30-s pulse and 30-s pause for 15 min. Cell lysates were centrifuged at 13,000 rpm for 30 min at 4 °C and pellets were suspended in SDS-loading buffer, and resolved on 12% SDS-PAGE. Distinct gel bands corresponding to CgYps1 (~50 kDa) and CgYps7 (~47 kDa) proteins were cut and crushed in the gel elution

buffer [25 mM Tris (pH 8.8), 24 mM glycine, 1% SDS and 5% glycerol], followed by overnight incubation at 37 °C. After centrifugation at 13,000 rpm for 15 min, the supernatant was collected. The remaining gel pellet was extracted again and the supernatants are pooled together, followed by overnight precipitation with ice-chilled acetone at −20 °C. Samples were centrifuged at 13,000 rpm for 30 min at 4 °C. Pellets were collected, washed twice with acetone and were suspended in Tris-SDS buffer [10 mM Tris (pH 7.8), 0.6% SDS]. CgYps1 and CgYps7 purity was verified by SDS-PAGE and Western blotting with the anti-His antibody. CgYps1 and CgYps1$^{D91A}$ were partially purified from *Pichia pastoris*, as described previously (Battu et al, 2021).

For incubation of rCgYps1 or rCgYps1$^{D91A}$ proteins with A-498 epithelial cells, A-498 ($1 \times 10^6$) cells were seeded in a 60 mm culture dish, and incubated overnight at 37 °C, followed by rCgYps1 or rCgYps1$^{D91A}$ [100 μg; in citrate buffer (pH 6.0)] addition to A-498 cells. As control, citrate buffer (pH 6.0) as well as the supernatant of partially purified, un-induced, CgYps1-expressing *P. pastoris* strain were added to A-498 cells. After 6 h incubation, A-498 cells were collected, and lysed in NETN buffer. Cell lysates (80 μg) were resolved on 12% SDS-PAGE.

## Mass spectrometry analysis

For identification of A-498 epithelial cell proteins that interact with CgYps1 or CgYps7, the affinity-purification-mass spectrometry approach was used. Briefly, *E. coli*-purified CgYps1 and CgYps7 proteins (250 μg) were incubated overnight with 100 μl TALON beads at 4 °C on an end-to-end rotor. A-498 cells were grown in MEM in 10 cm culture dishes to 80% confluency, followed by cell collection and lysis in NETN buffer. Cell lysates (2 mg protein) were pre-cleared with NETN buffer-equilibrated TALON beads for 2 h at 4 °C and centrifuged. The supernatants were divided into two fractions. One fraction was incubated with NETN buffer pre-equilibrated beads as control, and the other fraction was incubated with TALON beads-conjugated with CgYps1/CgYps7 proteins. After overnight incubation at 4 °C, beads were centrifuged at 1500 rpm for 2 min, and washed four times with NETN buffer containing 25 mM imidazole, followed by boiling and resolving of bead-bound proteins on 12% SDS-PAGE. Gel was run for a short duration, stained with coomassie brilliant blue, and protein-containing gel bands were excised and sent to the Taplin Biological Mass Spectrometry Facility, Harvard Medical School, Boston for protein identification using LC-MS/MS. Samples from two biological replicates were analyzed. Peptide identification was carried out using the the Sequest software, and peptides, filtered to 1% false discovery rate, were mapped to the Human reference proteome database (www.uniprot.org). For identification of CgYps1 and CgYps7-specific interactors in A-498 cells, two criteria were applied. First, proteins identified in the lysate-control pulled-down samples were removed, as these represent proteins binding non-specifically to TALON beads. Second, proteins identified with ≥2 total peptides in both biological replicates of CgYps1/CgYps7-conjugated TALON bead-pulled-down samples were selected for further analysis.

## Protein interaction analysis

Arpc1B-SFB-expressing A-498 cells were cultured in MEM medium and either infected with *wt* and *Cgyps1-11Δ* at 1:1 MoI, or left uninfected for 6 h. As a control, GFP-SFB-expressing A-498

cells were grown. A-498 cells were lysed in NETN buffer, and 800 μg cell lysates were incubated with streptavidin beads for 2 h at 4 °C. For validation of CgYps1-Arpc1B or CgYps7-Arpc1B interaction, 50–100 μg of *E. coli*-purified CgYps1 or CgYps7 protein was co-incubated with Arpc1B-conjugated streptavidin beads for 12 h at 4 °C. Following washes, proteins were eluted in biotin solution (2 mg/ml), boiled in SDS-loading buffer, resolved on 12% SDS-PAGE gel and were immunoblotted. For Arpc1B-SFB and p38 interaction, 1.5 mg cell lysates were incubated with streptavidin beads, followed by immunoblotting.

To identify CgYps1-interacting amino acid residues in Arpc1B, amino acid sequences were retrieved from the protein databank (PDB; https://www.wwpdb.org/) for CgYps1 and Arpc1B proteins, and submitted to the HDOCK server in the FASTA format (http://hdock.phys.hust.edu.cn/). CgYps1 and Arpc1B were submitted as receptor and ligand, respectively. Model 1 was chosen based on the highest docking score which predicted that Arginine-74 and Arginine-142 in Arpc1B interact with Tyrosine-402 and Aspartate-445 residues in CgYps1, respectively.

## Cytokine secretion analysis

A-498 and HEK293T cells were seeded in a 24-well culture plate, and infected with *Cg* at a MoI of 1:1. Four hours post infection, cells were washed thrice with PBS, fresh medium was added, and plates were incubated for 24 h at 37 °C. Supernatants were collected, centrifuged at 1500 rpm for 10 min to eliminate the particulate matter, if any, and were analyzed for IL-6, IL-8 and GM-CSF cytokine levels using BD OptEIA ELISA kits as per the supplier's instructions. A difference of ≥20% in cytokine levels was considered as significant.

For IL-6, KC and MIP-2 measurement in mouse kidneys, 6–8-week old female BALB/c mice were infected with overnight YPD medium-grown *wt* or *Cgyps1-11Δ* ($4 \times 10^7$ cells, 100 μl cell suspension in PBS) strains via tail vein. At 4$^{th}$ day post-infection, mice were euthanized by $CO_2$ inhalation, kidneys were collected, and homogenized in 1 ml PBS. The kidney homogenates were further lysed in tissue lysis buffer [10 mM Tris-Cl (pH 8.0), 150 mM NaCl, 1% NP-40, 10% glycerol, 5 mM EDTA, and 1 X protease inhibitor cocktail] for 60 min at 4 °C, followed by sonication. Lysates were centrifuged at 13,000 rpm for 30 min at 4 °C. IL-6, KC and MIP-2 levels in the supernatants were measured using the mouse IL-6 BD OptEIA, mouse CXCL1/KC R&D DuoSet and mouse CXCL2/MIP-2 R&D DuoSet ELISA kits, respectively.

## qRT-PCR analysis

RNA from 6 h *Cg*-infected or uninfected A-498 cells was extracted using the TRIZOL method. cDNA was synthesized using 2 μg of DNAseI-digested RNA using Superscript III Reverse Transcriptase (Thermo Scientific) as per manufacturer's instructions, and used as a template to examine Arpc1B and GAPDH expression.

## Annexin V staining

A-498 cells were seeded at a density of $5 \times 10^5$ cells in a 60 mm$^2$ dish for overnight. After Arpc1B-siRNA and control-siRNA transfection for 36–48 h, cells were harvested and washed with cold PBS, followed by centrifugation and cell suspension in the Annexin-

binding buffer (10 mM HEPES, 140 mM NaCl, and 2.5 mM CaCl$_2$, pH 7.4). 100 µl cell suspension ($\sim$1 × 10$^5$ cells/ml) was incubated with Annexin V-Alexa Flour 488 conjugate (ThermoFisher, USA) for 15–20 min at room temperature, and stained cells were immediately analysed by flow cytometry. The mean fluorescence intensity (MFI) of $\sim$20,000 cells was recorded at an excitation/ emission wavelength of 488/515 nm. Data were acquired and analysed using the BD FACSDIVA v9.0 software. MFI values of unstained cells were subtracted from those of stained samples. MFI values were normalized to A-498-control siRNA-transfected cells (taken as 100), and plotted.

### Neutrophil isolation and A-498-neutrophil co-culture

Neutrophils were isolated from the blood of healthy volunteer donors using Polymorphprep™. The protocol (IEC 21/2015) was approved by the Institutional Bioethics Committee. Briefly, blood samples were collected in tubes containing EDTA as an anticoagulant, and carefully layered over Polymorphprep™ (5 ml) in a 15 ml centrifuge tube. Tubes were centrifuged at 500×$g$ for 30 min at room temperature, and polymorphonuclear leukocytes (PMNs) were gently collected from the fourth layer, followed by centrifugation and washing with 1× Hanks Balanced Salt Solution (HBSS). Next, the collected PMNs were treated with 1× RBC lysis buffer for 10 min at room temperature to remove contaminating erythrocytes, and collected by centrifugation. To check the purity of neutrophils in the PMN fraction, neutrophils were labelled with FITC-conjugated anti-CD66b antibody in the binding buffer (5% FBS + 0.25 mM EDTA in PBS) for 30–60 min at 4 °C, and the neutrophil number was quantified using a hemocytometer as well as the BD FACSAria™ cell sorter machine. The PMN fraction contained $\sim$80–90% neutrophils.

Prior to $Cg$ infection and the human neutrophil co-culture, A-498 cells were seeded at a density of 2 × 10$^5$ cells per well in a 12-well tissue culture plate. After overnight incubation at 37 °C, cells were infected with $Cg$ at a MoI of 1:1. Four hours post infection, cells were washed thrice with PBS, fresh medium was added and the plate was incubated for another 20 h at 37 °C. Next, A-498 cells were co-cultured with FITC-conjugated anti-CD66b antibody-labelled neutrophils (1 × 10$^6$) that were injected into the upper chamber of the polycarbonate cell culture insert (Nunc™; pore size of 3 µm) for 2 h at 37 °C in a 5% CO$_2$ chamber. As a positive control, labelled-neutrophils were transferred to the insert filter in a 12-well plate containing MEM medium supplemented with 3 ng human recombinant IL-8. After incubation, the non-migrating cells on the origin side (top) of the filter were dissociated by gently wiping the filter, as well as, the cells, that had migrated to the bottom chamber, were quantified by recording the fluorescence signal in a microplate reader (Excitation: 485 nm; Emission: 530 nm).

For $Cg$-neutrophil interaction analysis, $Cg$ were infected to neutrophils at a MoI of 1:1. At 30 min, 60 min and 120 min post infection, neutrophils were scraped off, incubated at 37 °C for 10 min, vortexed and appropriate dilutions were plated on YPD medium. The number of yeast colonies were counted and % $Cg$ viability was determined by dividing each time-point CFUs by 0 h CFUs, and multiplying the number by 100.

### Statistical analysis

Data, from at least three biological replicates, are reported as mean ± SEM. Statistical analyses were carried out using the GraphPad Prism software. Unpaired or paired two-tailed Student's $t$ test was used to determine the statistical significance. A $P$ value of ≤0.05 was considered as statistically significant. Asterisks were used to denote varied $P$ values viz., *$P$ ≤ 0.05; **$P$ ≤ 0.01; ***$P$ ≤ 0.001; and ****$P$ ≤ 0.0001.

## Data availability

The raw mass spectrometry proteomics data have been deposited to the ProteomeXchange Consortium via the PRIDE partner repository (Perez-Riverol et al, 2022), with the dataset identifier PXD053938 (http://www.ebi.ac.uk/pride/archive/projects/PXD053938).

The source data of this paper are collected in the following database record: biostudies:S-SCDT-10_1038-S44319-024-00270-y.

## Peer review information

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

## Acknowledgements

We are indebted to Drs. P Chandra Shekar, Maddika Subba Reddy and Rashna Bhandari for their generous gifts of pU6-2A-eGFP-2A-Puro plasmid, Gateway cloning vectors and HEK293T cell line, respectively. We thank Ross Tomaino, Taplin MS facility, Harvard Medical School, and personnel of the Sophisticated Equipment Facility of CDFD, for their help with mass spectrometry analysis, and imaging and immunohistochemistry studies, respectively. This work was supported by the DBT/Wellcome Trust India Alliance Senior Fellowships to RK [IA/S/15/1/501831 and IA/S/23/1/506745; www.indiaalliance.org/]; and grants from the Department of Biotechnology [BT/PR40336/BRB/10/1921/2020 and BT/PR42015/MED/29/1561/2021; www.dbtindia.gov.in/]; and the Science and Engineering Research Board, Department of Science and Technology [CRG/2021/000530; www.serb.gov.in/], Government of India, to RK. SP is a recipient of the research fellowship of the Department of Biotechnology, Government of India. The funders had no role in study design, data collection and analysis, or decision to publish or preparation of the manuscript.

## Author contributions

**Sandip Patra**: Conceptualization; Data curation; Formal analysis; Validation; Investigation; Visualization; Methodology; Writing—original draft; Writing—review and editing. **Rupinder Kaur**: Conceptualization; Resources; Data curation; Formal analysis; Supervision; Funding acquisition; Visualization; Writing—original draft; Project administration; Writing—review and editing.

## Disclosure and competing interests statement

The authors declare no competing interests.

# Expanded View Figures

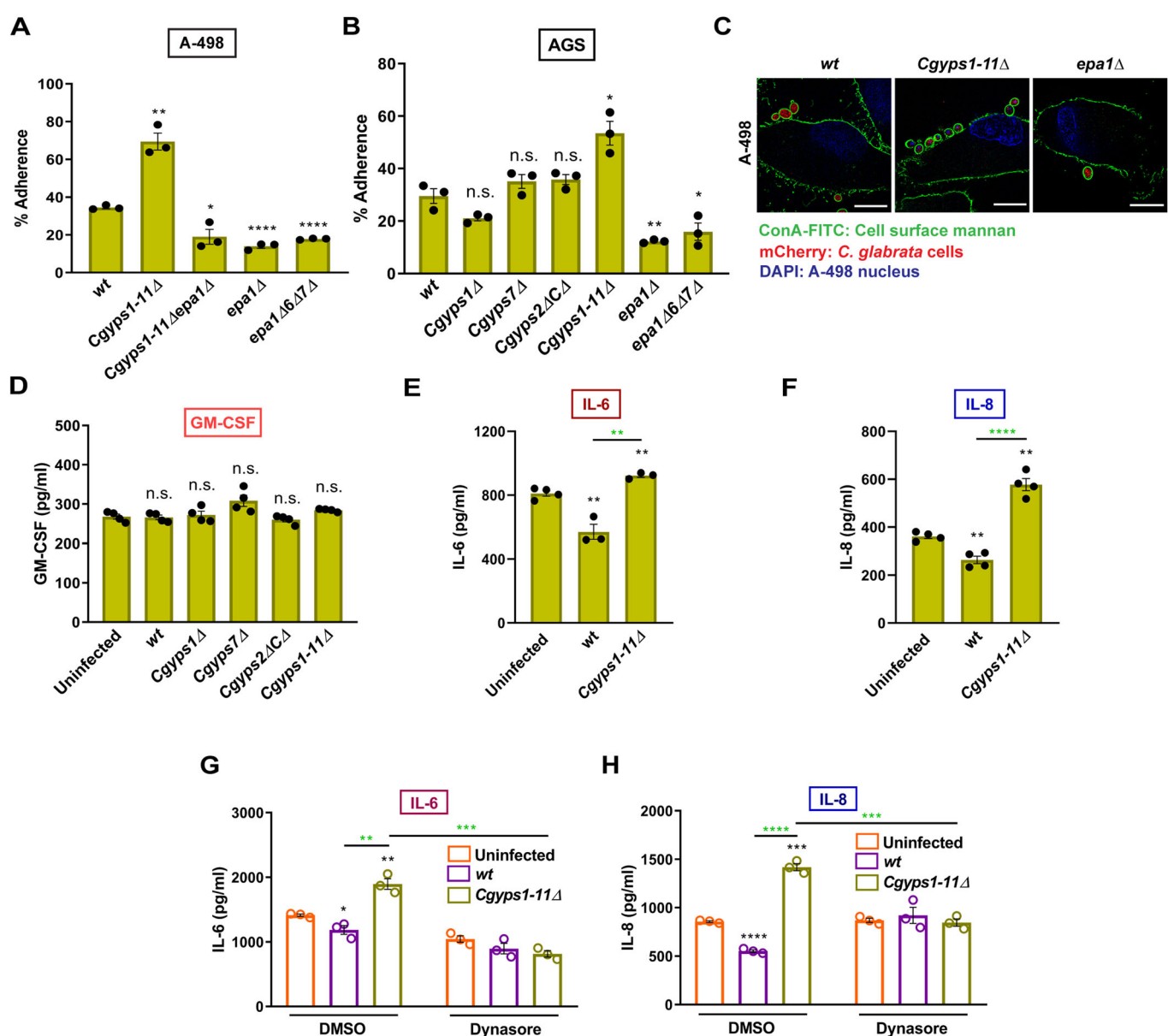

**Figure EV1. *Cgyps1-11Δ* displays increased adherence to AGS cells.**

(A) Adherence of indicated, S35-labelled *Cg* strains to fixed A-498 (human kidney epithelial) cells after 2 h co-incubation. Black asterisks denote statistically-significant adherence differences in indicated strains, compared to *wild-type (wt)*-infected A-498. *n* = 3 biological replicates. (B) Adherence of indicated, S35-labelled *Cg* strains to fixed AGS (human stomach epithelial) cells after 2 h co-incubation. Black asterisks denote statistically-significant adherence differences in indicated strain-infected, as compared to *wild-type (wt)*-infected AGS cells. *n* = 3 biological replicates. (C) Super resolution micrographs showing interaction of mCherry-expressing *wt, Cgyps1-11Δ* and *epa1Δ* strains with A-498 cells after 2 h co-culture. Infected epithelial cells were stained with concanavalin A (ConA)-FITC to differentiate between intracellular (ConA-non-stained) and extracellular (ConA-stained) *Cg*. DAPI was used to stain the host epithelial cell nuclei. Representative images were obtained from three biological experiments using Elyra 7 with 63X/1.44 NA objective lens. (D) Secreted GM-CSF (granulocyte-macrophage colony-stimulating factor) measurement in uninfected and *Cg*-infected A-498 cells after 24 h incubation. *n* = 4 biological replicates. (E, F) Secreted IL-6 (E) and IL-8 (F) levels in uninfected and *Cg*-infected A-498 cells. Infection was carried out at a MoI (multiplicity of infection) of 10:1. *n* = 3 biological replicates in (E), and *n* = 4 biological replicates in (F). (G, H) Secreted IL-6 (G) and IL-8 (H) levels in DMSO or dynasore (50 μM)-treated, *Cg*-infected, A-498 cells. Infection was carried out at 1:1 MoI. *n* = 3 biological replicates. Data information: In (A, B, D-H), data are presented as mean ± SEM. *P < 0.05; **P < 0.01; ***P < 0.001; ****P < 0.0001; n.s., not significant. Unpaired two-tailed Student's *t* test in (A, B, D-H). P = 0.0016 (*Cgyps1-11Δ* vs. *wt*), P = 0.000072 (*epa1Δ* vs. *wt*), P = 0.0178 (*Cgyps1-11Δepa1Δ* vs. *wt*), P = 0.00002 (*epa1Δ6Δ7Δ* vs. *wt*) in (A). P = 0.0106 (*Cgyps1-11Δ* vs. *wt*), P = 0.0036 (*epa1Δ* vs. *wt*), P = 0.035 (*epa1Δ6Δ7Δ* vs. *wt*) in (B). P = 0.0031 (*wt* vs. uninfected), P = 0.0042 (*Cgyps1-11Δ* vs. uninfected), P = 0.002 (*Cgyps1-11Δ* vs. *wt*) in (E). P = 0.0016 (*wt* vs. uninfected), P = 0.0002 (*Cgyps1-11Δ* vs. uninfected), P = 0.00004128 (*Cgyps1-11Δ* vs. *wt*) in (F). P = 0.0306 (*wt* vs. uninfected), P = 0.0048 (*Cgyps1-11Δ* vs. uninfected), P = 0.0027 (*Cgyps1-11Δ* vs. *wt*) P = 0.0004 (*Cgyps1-11Δ*-Dynasore vs. *Cgyps1-11Δ*-DMSO) in (G). P = 0.00004918 (*wt* vs. uninfected), P = 0.0001 (*Cgyps1-11Δ* vs. uninfected), P = 0.00002526 (*Cgyps1-11Δ* vs. *wt*), P = 0.0004 (*Cgyps1-11Δ*-Dynasore vs. *Cgyps1-11Δ*-DMSO) in (H). Scale bar = 10 μm in (C).

   

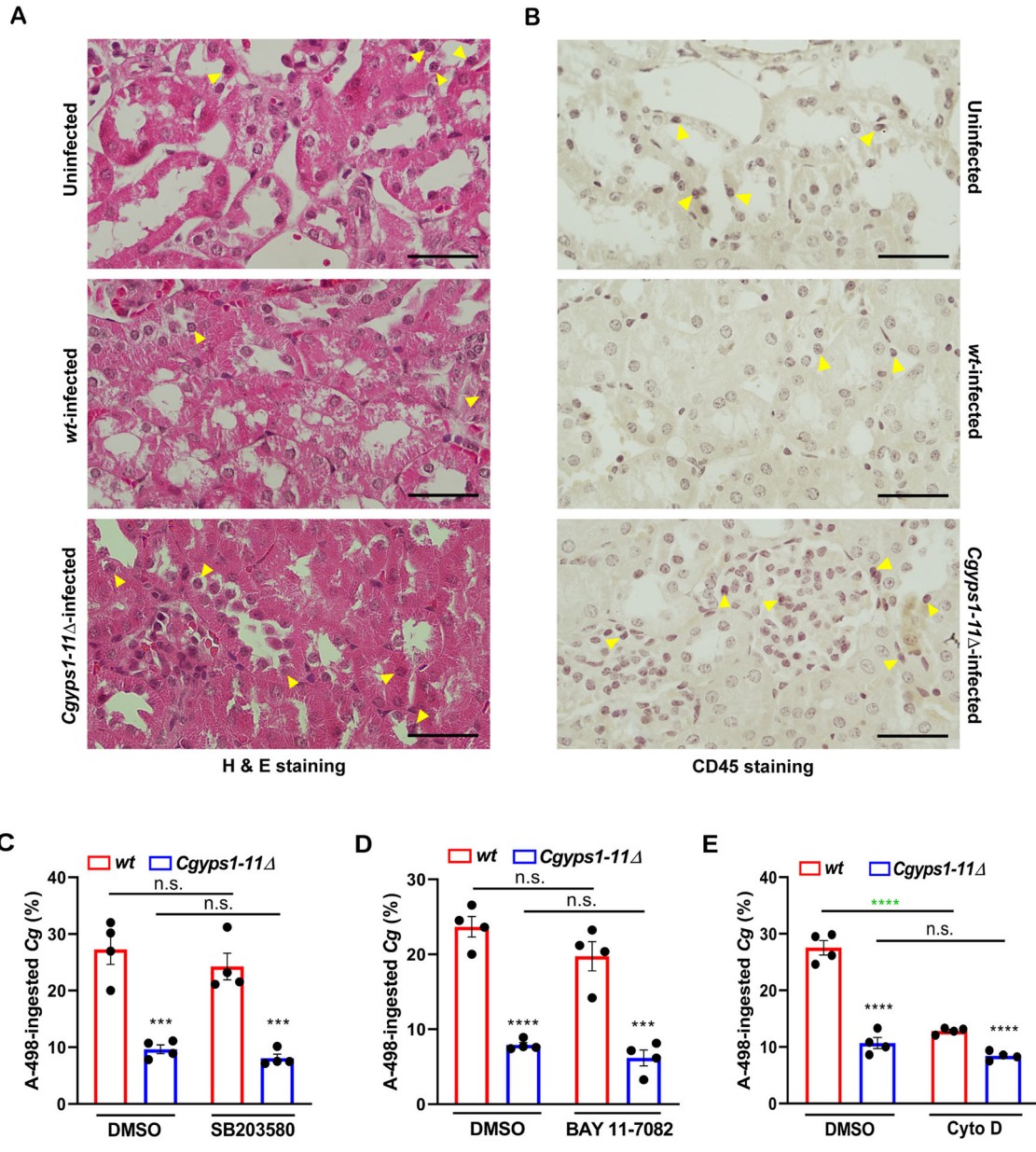

**Figure EV2.  p38 MAPK inhibition has no effect on *Cg* internalization.**

(**A, B**) Micrographs of hematoxylin-eosin (H&E)-stained (**A**) and anti-CD45 antibody-stained (**B**) kidney tissue sections (40×) of uninfected, and *wt*- or *Cgyps1-11Δ*-infected mice at day 1 post-infection. $n = 3$ mice/group. Yellow arrowheads mark representative polymorphonuclear neutrophils infiltrated into the tissue. (**C, D**) CFU-based internalization analysis of *wt* and *Cgyps1-11Δ* strains in DMSO, SB203580 (10 µM; **C**) or BAY 11-7082 (10 µM; **D**) pre-treated A-498 cells, after 4 h co-incubation. The internalization percentage was calculated by dividing *Cg* CFUs recovered at 4 h by 0 h-CFUs (*Cg* cell number used for A-498 infection), and multiplying the number by 100. Asterisks mark differences between *wt* and *Cgyps1-11Δ* ingestion. $n = 4$ biological replicates. (**E**) CFU-based internalization analysis of *wt* and *Cgyps1-11Δ* strains in DMSO or cytochalasin D (Cyto D; 5 µM) pre-treated A-498 cells, after 4 h co-incubation. Black asterisks mark differences between *wt* and *Cgyps1-11Δ* ingestion. $n = 4$ biological replicates. Data information: In (**C–E**), data are presented as mean ± SEM. *$P < 0.05$; **$P < 0.01$; ***$P < 0.001$; ****$P < 0.0001$; n.s., not significant. Unpaired two-tailed Student's *t* test in (**C–E**). $P = 0.0007$ (*Cgyps1-11Δ*-DMSO vs. *wt*-DMSO), $P = 0.0006$ (*Cgyps1-11Δ*-SB203580 vs. *wt*-SB203580) in (**C**). $P = 0.000031$ (*Cgyps1-11Δ*-DMSO vs. *wt*-DMSO), $P = 0.0009$ (*Cgyps1-11Δ*-BAY 11-7082 vs. *wt*-BAY 11-7082) in (**D**). $P = 0.0000458$ (*Cgyps1-11Δ*-DMSO vs. *wt*-DMSO), $P = 0.0000942$ (*Cgyps1-11Δ*-CytoD vs. *wt*-CytoD), $P = 0.0000295$ (*wt*-CytoD vs. *wt*-DMSO) in (**E**). Scale bar = 20 µm in (**A, B**).

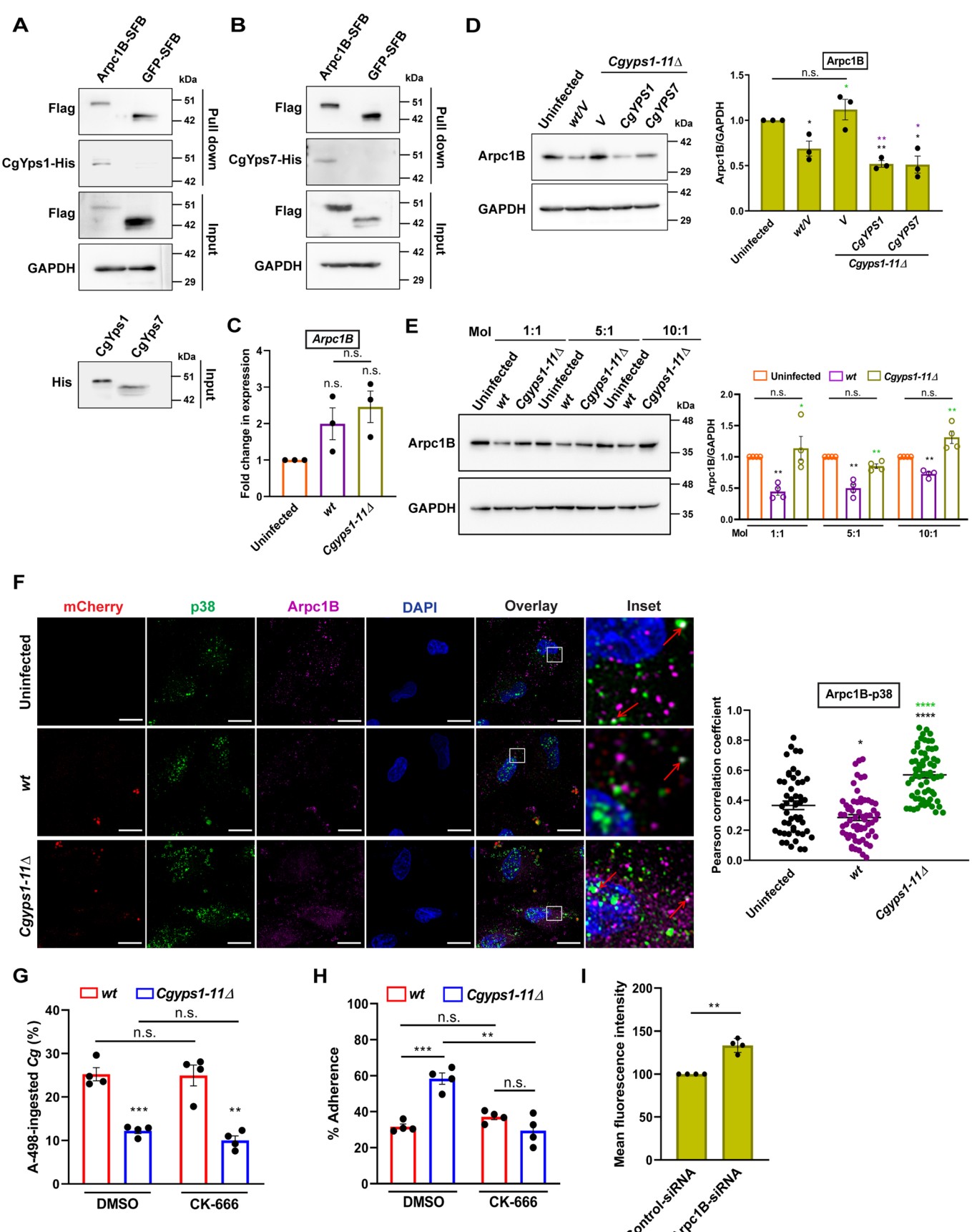

**Figure EV3. CgYps1 and CgYps7 interact with the epithelial cell protein Arpc1B.**

(A, B) Representative immunoblots ($n = 2$ biological replicates) illustrating Arpc1B interaction with CgYps1 (A) and CgYps7 (B). Lysates (800 μg) of Arpc1B-SFB- or GFP-SFB-expressing A-498 cells were incubated with streptavidin beads, followed by incubation with 50-100 μg of *E. coli*-purified, 6X-histidine-tagged CgYps1 or CgYps7 protein. Input samples (*E. coli*-purified proteins; 30 μg) for CgYps1 (A) and CgYps7 (B) are shown underneath the blots of the panel (A). (C) qRT-PCR-based Arpc1B gene expression analysis in indicated A-498 cells after 6 h of *Cg* infection. Arpc1B gene expression was normalized against GAPDH mRNA control, and represent fold-change in Arpc1B transcript levels in *Cg*-infected, compared to uninfected A-498 cells (taken as 1.0). $n = 3$ biological replicates. (D, E) Representative immunoblots illustrating Arpc1B protein levels in A-498 cells that were left uninfected or infected for 6 h at 1:1 MoI with *wt* expressing empty plasmid (*V*) or *Cgyps1-11Δ* expressing *V*, *CgYPS1* or *CgYPS7* (D) or infected with *wt* and *Cgyps1-11Δ* strains at indicated MoI (E). The signal intensity in each lane was quantified using the ImageJ software, and Arpc1B levels were normalized against the corresponding GAPDH levels. Data are plotted on the right side of the blots. Purple, green and black asterisks indicate Arpc1B level differences, as compared to *Cgyps1-11Δ/V*-infected, *wt/V*-infected and uninfected A-498 cells, respectively. $n = 3$ biological replicates in (D), and $n = 4$ biological replicates in (E). (F) Representative confocal micrographs illustrating co-localization of Arpc1B and p38 MAPK in A-498 cells. A-498 cells were left uninfected or infected with mCherry-expressing *wt* and *Cgyps1-11Δ*-strains for 6 h, followed by labelling with anti-Arpc1B and anti-p38 antibodies. Cells from two biological infection experiments were imaged using the confocal microscope (Leica SP8) with 63X/1.44 NA objective lens in z-stack mode. Arpc1B and p38 signal intensities were measured for 50 foci in a minimum of 15 cells, using the LASX software, and their co-localization was determined via the Pearson correlation coefficient (PCC). Co-localization data are plotted on the right side of the micrographs. Arpc1B and p38 co-localization is indicated by red arrows in the Inset. Green and black asterisks indicate statistically-significant differences, compared to *wt*-infected and uninfected A-498 cells, respectively. (G) CFU-based internalization analysis of *wt* and *Cgyps1-11Δ* strains in DMSO or CK-666 (1 μM) pre-treated A-498 cells, after 4 h co-incubation. Asterisks mark differences between *wt* and *Cgyps1-11Δ* ingestion. $n = 4$ biological replicates. (H) CFU-based adherence analysis of *wt* and *Cgyps1-11Δ* strains in DMSO or CK-666 (1 μM) pre-treated A-498 cells, after 2 h co-incubation. $n = 4$ biological replicates. (I) Flow cytometry-based analysis of Annexin V staining in indicated A-498 cells. $n = 4$ biological replicates. Data information: In (C–I), data are presented as mean ± SEM. *$P < 0.05$; **$P < 0.01$; ***$P < 0.001$; ****$P < 0.0001$; n.s., not significant. Unpaired or paired two-tailed Student's *t* test in (C–I). $P = 0.0482$ (*wt/V* vs. uninfected), $P = 0.0375$ (*Cgyps1-11Δ/V* vs. *wt/V*), $P = 0.0056$ (*Cgyps1-11Δ/CgYPS1* vs. uninfected), $P = 0.0349$ (*Cgyps1-11Δ/CgYPS7* vs. uninfected), $P = 0.0073$ (*Cgyps1-11Δ/CgYPS1* vs. *Cgyps1-11Δ/V*), $P = 0.0145$ (*Cgyps1-11Δ/CgYPS7* vs. *Cgyps1-11Δ/V*) in (D). $P = 0.0025$ (*wt* at 1:1 MoI vs. uninfected), $P = 0.0129$ (*Cgyps1-11Δ* vs. *wt* at 1:1 MoI), $P = 0.004$ (*wt* at 5:1 MoI vs. uninfected), $P = 0.0026$ (*Cgyps1-11Δ* vs. *wt* at 5:1 MoI), $P = 0.0023$ (*wt* at 10:1 MoI vs. uninfected), $P = 0.0013$ (*Cgyps1-11Δ* vs. *wt* at 10:1 MoI) in (E). $P = 0.0178$ (*wt* vs. uninfected), $P = 0.00000004$ (*Cgyps1-11Δ* vs. uninfected), $P = 0.00000000000000001$ (*Cgyps1-11Δ* vs. *wt*) in (F). $P = 0.0002$ (*Cgyps1-11Δ*-DMSO vs. *wt*-DMSO), $P = 0.0013$ (*Cgyps1-11Δ*-CK-666 vs. *wt*-CK-666) in (G). $P = 0.0003$ (*Cgyps1-11Δ*-DMSO vs. *wt*-DMSO), $P = 0.0015$ (*Cgyps1-11Δ*-CK-666 vs. *Cgyps1-11Δ*-DMSO) in (H). $P = 0.0038$ (*Arpc1B*-siRNA vs. Control siRNA) in (I). Scale bar = 20 μm in (F).

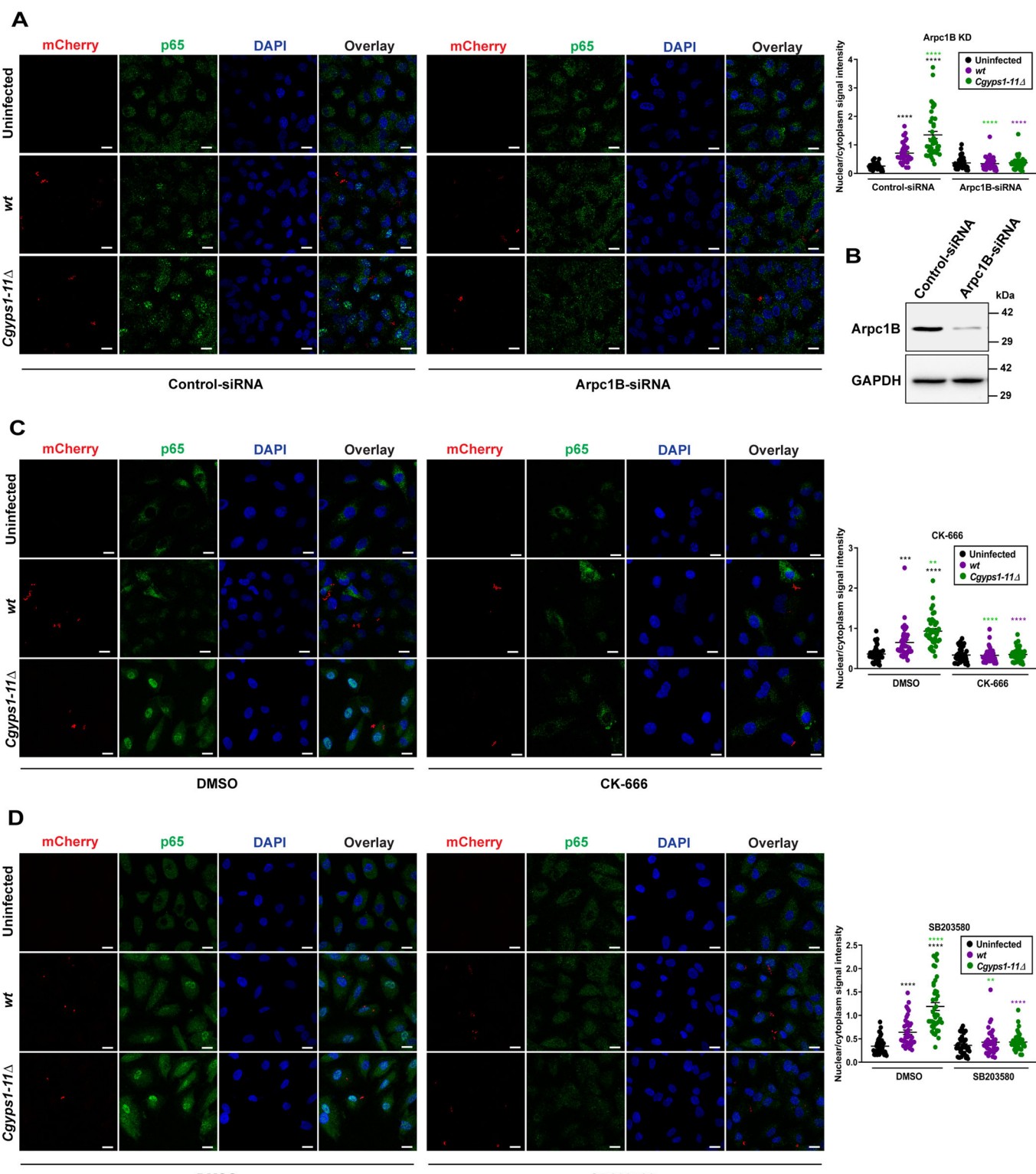

◀ **Figure EV4. Arpc1B inhibition decreases p65 nuclear localization in *Cgyps1-11Δ*-infected A-498 cells.**

(A) Confocal micrographs illustrating p65 cellular localization in control-siRNA or Arpc1B-siRNA treated A-498 cells, after 6 h of infection with mCherry-expressing *wt* or *Cgyps1-11Δ* cells. Immunofluorescence analysis of A-498 cells was performed with anti-p65 antibody, and images were captured using the confocal microscope (Leica SP8) with 63X/1.44 NA objective lens in z-stack mode. Overlay images show cytoplasmic and nuclear p65 distribution, with the anti-rabbit Alexa Flour-488 antibody in green, the mCherry-tagged *Cg* in red and the host cell nuclei in blue color. For quantification, fluorescence signal intensities in the cytoplasm and the nucleus were measured in a minimum of 40 cells in two biological experiments, using the ImageJ software. Data are plotted, as the ratio of nuclear to cytoplasm abundance, on the right side of micrographs. Green and black asterisks indicate statistically-significant differences, as compared to *wt*-infected and uninfected A-498 cells, respectively. (B) Representative immunoblots showing reduction in Arpc1B protein levels in A-498 cells that were transfected with Arpc1B-siRNA (10 μm), as compared to scrambled-siRNA (10 μm)-transfected A-498 cells. GAPDH was used as loading control. $n = 3$ biological replicates. (C, D) Confocal micrographs illustrating p65 cellular localization in DMSO or CK-666 (1 μM)-treated (C), and DMSO or SB203580 (10 μM)-treated (D) A-498 cells, after 6 h of infection with mCherry-expressing *wt* or *Cgyps1-11Δ* cells. Immunofluorescence analysis of A-498 cells was performed with anti-p65 antibody, and images were captured using the confocal microscope (Leica SP8) with 63X/1.44 NA objective lens in z-stack mode. Overlay images show cytoplasmic and nuclear p65 distribution, with the anti-rabbit Alexa Flour-488 antibody in green, the mCherry-tagged *Cg* in red and the host cell nuclei in blue color. For quantification, fluorescence signal intensities in the cytoplasm and the nucleus were measured in a minimum of 40 cells in two biological experiments, using the ImageJ software. Data are plotted, as the ratio of nuclear to cytoplasm abundance, on the right side of micrographs. Green and black asterisks indicate statistically-significant differences, as compared to *wt*-infected and uninfected A-498 cells, respectively. Data information: In (A, C, D), data are presented as mean ± SEM. \*\**P* < 0.01; \*\*\**P* < 0.001; \*\*\*\**P* < 0.0001; n.s. not significant. Unpaired two-tailed Student's *t* test in (A, C, D). $P = 0.0000000005$ (*wt*-Control siRNA vs. uninfected-Control siRNA), $P = 0.000000000001577$ (*Cgyps1-11Δ*-Control siRNA vs. uninfected-Control siRNA), $P = 0.0000234$ (*Cgyps1-11Δ*-Control siRNA vs. *wt*-Control siRNA), $P = 0.0000002043$ (*wt*-Arpc1B siRNA vs. *wt*-Control siRNA), $P = 0.0000000002$ (*Cgyps1-11Δ*-Arpc1B siRNA vs. *Cgyps1-11Δ*-Control siRNA) in (A). $P = 0.000143$ (*wt*-DMSO vs. uninfected-DMSO), $P = 0.000000000002$ (*Cgyps1-11Δ*-DMSO vs. uninfected-DMSO), $P = 0.0018$ (*Cgyps1-11Δ*-DMSO vs. *wt*-DMSO), $P = 0.00001608$ (*wt*-CK-666 vs. *wt*-DMSO), $P = 0.0000000000004$ (*Cgyps1-11Δ*-CK-666 vs. *Cgyps1-11Δ*-DMSO) in (B). $P = 0.00000104$ (*wt*-DMSO vs. uninfected-DMSO), $P = 0.000000000000023$ (*Cgyps1-11Δ*-DMSO vs. uninfected-DMSO), $P = 0.00000029$ (*Cgyps1-11Δ*-DMSO vs. *wt*-DMSO), $P = 0.0014$ (*wt*-SB203580 vs. *wt*-DMSO), $P = 0.000000000001$ (*Cgyps1-11Δ*-SB203580 vs. *Cgyps1-11Δ*-DMSO) in (D). Scale bar = 20 μm in (A, C, D).

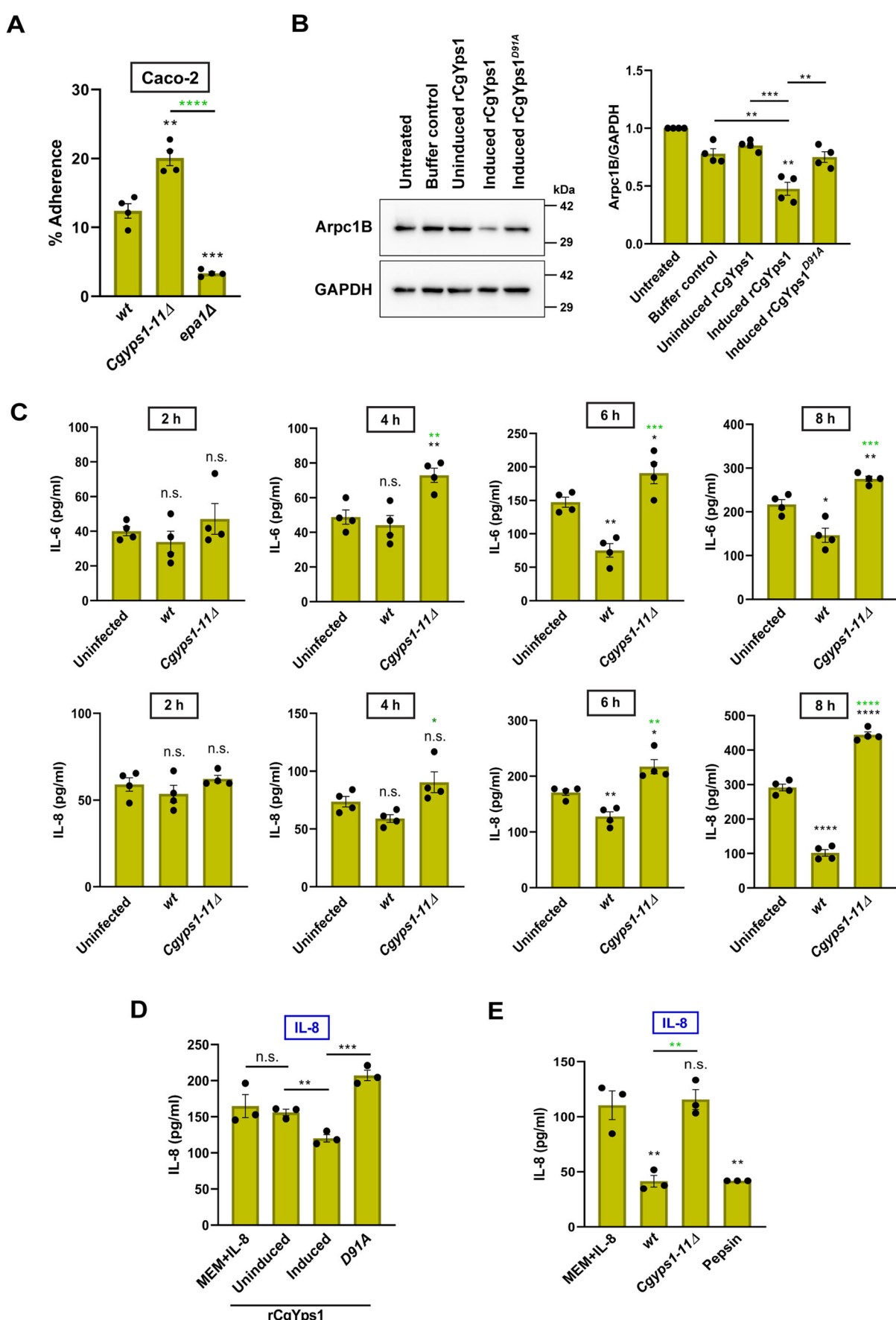

◀

**Figure EV5. rCgYps1 reduces Arpc1B levels.**

(A) Adherence of indicated *Cg* strains to fixed Caco-2 (human intestinal epithelial) cells after 2 h co-incubation, as determined by CFU-based assay. Black asterisks mark adherence differences, as compared to *wt*-coincubated A-498 cells. $n = 4$ biological replicates. (B) Representative immunoblots illustrating Arpc1B levels in A-498 cells that were either left untreated or incubated for 6 h with *P. pastoris*-partially purified rCgYps1 (100 µg), rCgYps1$^{D91A}$ (100 µg), partially-purified proteins (100 µg) from uninduced supernatants of *P. pastoris* cells or citrate buffer (pH 6.0). rCgYps1 and rCgYps1$^{D91A}$ protein expression in *P. pastoris* cells was induced by adding (2%) methanol. The signal intensity in each lane was measured using the ImageJ software, and Arpc1B levels were normalized against GAPDH levels. Data represent fold-change in Arpc1B levels in indicated conditions, as compared to untreated A-498 cells (considered as 1.0), and are plotted on the right side of the blots. $n = 4$ biological replicates. (C) Secreted IL-6 and IL-8 cytokine levels in uninfected, *wt*- and *Cgyps1-11Δ*-infected A-498 cells at indicated time points post-infection. Infection was carried at 5:1 MoI. Green and black asterisks indicate statistically-significant differences in cytokine secretion, as compared to *wt*-infected and uninfected A-498 cells, respectively. $n = 4$ biological replicates. (D) *P. pastoris*-partially-purified rCgYps1 and rCgYps1$^{D91A}$ proteins (100 µg) were incubated with the human recombinant IL-8 cytokine (200 pg; MEM medium) for 4 h at 37 °C, and IL-8 levels were measured using the human IL-8 BD OptEIA ELISA kit. IL-8 incubated in MEM medium was used as control. $n = 3$ biological replicates. (E) Human recombinant IL-8 (100 pg) was incubated with overnight YPD-grown *wt* and *Cgyps1-11Δ* cells (1.0 O.D$_{600}$; MEM medium), pepsin (100 ng) or MEM medium for 6 h at 37 °C. Samples were centrifuged to remove *Cg* cells, and IL-8 levels in supernatants were measured using the human IL-8 BD OptEIA ELISA kit. The Pepsin enzyme was used as control. Black asterisks denote statistically-significant differences, as compared to IL-8 incubated in MEM medium. $n = 3$ biological replicates. Data information: In (A–E), data are presented as mean ± SEM. *$P < 0.05$; **$P < 0.01$; ***$P < 0.001$; ****$P < 0.0001$; n.s., not significant. Unpaired or paired two-tailed Student's *t* test in (A–E). $P = 0.0002$ (*epa1Δ* vs. *wt*), $P = 0.0024$ (*Cgyps1-11Δ* vs. *wt*), $P = 0.00000603$ (*Cgyps1-11Δ* vs. *epa1Δ*) in (A). $P = 0.003$ (Induced rCgYps1 vs. untreated), $P = 0.005$ (Induced rCgYps1 vs. buffer control), $P = 0.0008$ (Induced rCgYps1 vs. uninduced rCgYps1), $P = 0.009$ (Induced rCgYps1$^{D91A}$ vs. induced rCgYps1) in (B). $P = 0.0058$ (*Cgyps1-11Δ*-4h vs. *wt*-4h), $P = 0.0063$ (*Cgyps1-11Δ*-4h vs. uninfected-4h), $P = 0.0013$ (*wt*-6h vs. uninfected-6h), $P = 0.0475$ (*Cgyps1-11Δ*-6h vs. uninfected-6h), $P = 0.0008$ (*Cgyps1-11Δ*-6h vs. *wt*-6h), $P = 0.0109$ (*wt*-8h vs. uninfected-8h), $P = 0.0037$ (*Cgyps1-11Δ*-8h vs. uninfected-8h), $P = 0.0003$ (*Cgyps1-11Δ*-8h vs. *wt*-8h) in IL-6 section of (C). $P = 0.0168$ (*Cgyps1-11Δ*-4h vs. *wt*-4h), $P = 0.0047$ (*wt*-6h vs. uninfected-6h), $P = 0.0153$ (*Cgyps1-11Δ*-6h vs. uninfected-6h), $P = 0.0011$ (*Cgyps1-11Δ*-6h vs. *wt*-6h), $P = 0.00000886$ (*wt*-8h vs. uninfected-8h), $P = 0.00002363$ (*Cgyps1-11Δ*-8h vs. uninfected-8h), $P = 0.00000019$ (*Cgyps1-11Δ*-8h vs. *wt*-8h) in IL-8 section of (C). $P = 0.0061$ (Induced rCgYps1 vs. uninduced rCgYps1), $P = 0.0006$ (Induced rCgYps1$^{D91A}$ vs. induced rCgYps1) in (D). $P = 0.0079$ (*wt* vs. MEM + IL-8), $P = 0.0061$ (Pepsin vs. MEM + IL-8), $P = 0.0021$ (*Cgyps1-11Δ* vs. *wt*) in (E).

