## [Peer Review File · EMBO Reports]

Aspartyl proteases target host actin nucleator complex protein to limit epithelial innate immunity

Sandip Patra and Rupinder Kaur

Corresponding author(s): Rupinder Kaur (rkaur@cdfd.org.in)

Review Timeline:

Submission Date:	2nd Nov 23
Editorial Decision:	7th Dec 23
Appeal Received:	8th May 24
Editorial Decision:	18th Jun 24
Revision Received:	2nd Aug 24
Editorial Decision:	13th Aug 24
Revision Received:	26th Aug 24
Accepted:	5th Sep 24

Editor: Achim Breiling / Martina Rembold

Transaction Report:

Dear Dr. Kaur,

Thank you for the submission of your research manuscript to EMBO reports. I have now received the full set of referee reports that are copied below.

I am sorry to say that the decision on your manuscript is not a positive one. As you will see, referees #1 and #3 indicate that the physiological relevance of the findings remains unclear, that part of the data are contradictory, that part of the results do not allow confident conclusions and that several conclusions are not convincingly supported by the data. Moreover, both note many experimental and technical shortcomings. Referee #2 is overall more positive, but also has concerns. As the reports are below, I will not further detail them here.

Given the comments of the referees, the amount of work required to address them, the fact that EMBO reports can only invite revision of papers that receive overall positive support from all the referees upon initial assessment, and also considering that referees hint at in their reports that these findings would be better suited for a more pathogen-oriented journal, I cannot offer to publish your manuscript.

I am sorry to have to disappoint you this time. I nevertheless hope that the referee comments will be helpful in your continued work in this area, and I thank you once more for your interest in our journal.

Yours sincerely

Referee #1:

The manuscript by Sandip Patra and Rupinder Kaur convincingly demonstrates that *Candida glabrata* (Cg) yapsins affect Arpc1B (actin nucleator Arp2/3 complex subunit) in epithelial cells, leading to actin disassembly and preventing IL-8 secretion by ECs. The authors demonstrate this by different complimentary approaches. They also show that interaction with Arpc1B has consequences for p38 activation and chemokine release, and that, thereby, certain yapsins contribute to immune evasion and fungal survival in the context of epithelial cells. These findings are novel, convincing, and of interest to the medical mycology community. However, the *in vivo* relevance remains to be demonstrated, thereby somewhat limiting my excitement about the novel findings.

There are additional shortcomings in other aspects (see below) that should be addressed.

Major points:

1) Relevance of the *in vitro* findings for systemic candidiasis in mice: The authors show that IL-6 levels in kidneys of mice infected with the Cgy^{ps1-11ko} mutant are lower compared to wt-infected tissue. However, the subsequent part of the manuscript largely addresses differences in IL-8 production. This raises two important questions: Were levels of the murine IL-8 homologues KC and MIP-2 increased in kidneys of Cgy^{ps1-11ko}-infected tissues? And if yes: Was this functionally relevant, eg was increased neutrophil recruitment observed? This is critically, as interactions in cell culture might diverge significantly from interactions *in vitro*, due to a number of reasons.

2) Connection of adhesion, internalization, and Arpc1B:

- Arpc1B levels in wt-infected A-498 cells were reduced within 15 min (significant after 30 min) of incubation with Cg (Fig. 3F) - given that Cg uptake is mediated by endocytosis, did endocytosis already occur within this time frame? There's no further decrease of Arpc1B levels over time. It seems unlikely that endocytosis is completed within 15 min, and therefore, these results would argue against internalization being required for interference host cell physiology.
- How can siRNA-mediated knockdown of Arpc1B reduce Cgy^{ps1-11ko} adherence, given that Arpc1B is located intracellularly? Is there any evidence that knockdown of Arpc1B indeed affects cell membrane structure - and that Cg adheres directly to the cell membrane rather than surface proteins or other surface molecules?

3) It should be made clear, that the different approaches used to reduce levels of Arpc1B (a strength of the manuscript) do not result in full reversal of the Cgy^{ps1-11ko} phenotype to wt:

- CK-666 treatment (Fig. 4) only partially restored Cgy^{ps1-11ko} viability in A-498, and led to an increase in IL-6 and IL-8 production in response to WT infection. This contradicts some of the previous and later conclusions around the role of Arpc1B - or it indicates that CK-666 has additional effects on other factors that affect survival and cytokine responses.
- siRNA knockdown only partially rescued Cgy^{ps1-11ko} viability (Fig. 4E)

4) It seems that activation of p38 only partially depends on Arpc1B - this should be addressed:

- Loss of p38 had a much more pronounced effect on Cgyps1-11ko survival than loss of Arpc1B (Fig. 4J, K). According to Fig. S5B, the clones selected for further experiments lacked any detectable Arpc1B expression, excluding incomplete repression of Arpc1B as the reason. Thus, it seems that pathways independent of Arpc1B likely activate p38 and p38-associated Cg inactivation within epithelial cells.
- Fig. 4L, M: Only Arpc1B ko reduced IL-8 levels following Cgyps1-11ko infection to that of wt-infected cells - p38 ko resulted in levels comparable to non-infected controls, but the reduction of IL-8 observed with the WT was not reached. Again, this might indicate that p38 and Arpc1B are not within the same pathway or that additional pathways are involved.

5) I'm not convinced of the validity of the neutrophil migration assay: Fig. 6A: When adding up the fluorescence intensity in the upper and lower chamber, the total fluorescence differs between the different treatment groups: It is much lower for Cg wt infected epithelial cells than for either uninfected or Cgyps1-11ko infected cells. Also, the uninfected signal is roughly a third lower than that of Cgyps1-11ko infection. One likely explanation is differences in neutrophil viability and/or issues with the stability of the fluorescent labelling. Thus, these results do not allow any confident conclusions on actual neutrophil migration. Furthermore, the exact experimental set up for Fig. S5E is not clear from the figure legend and information in materials and methods.

6) The part on other Candida pathogens is confusing regarding induction of p38 phosphorylation by *C. albicans*: The western blot in Fig. S6B shows a very strong band for P-p38, a weakened p38 band, yet according to the bar next to the western blot, the P-p38 to total p38 ration is lower than for uninfected cells. This contradicts published findings by other groups that have demonstrated robust activation of p38 by *C. albicans* and it's link to secretion of IL-8 that is reliably induced by this fungus (examples: <https://pubmed.ncbi.nlm.nih.gov/35380879/> ; <https://journals.aai.org/jimmunol/article/179/12/8435/79462/Candida-albicans-Triggers-Activation-of-Distinct> ; <https://journals.aai.org/jimmunol/article/171/6/3047/71449/Candida-albicans-Induces-Selectively>).

Minor points:

- Why was a human gastric (AGS) epithelial cell line used? There's no evidence that Cg translocates from the stomach - intestinal cells would appear more relevant.
- Fixation with formaldehyde alters protein structures. I therefore would consider these experiments of only limited relevance and show them as supplementary data. The data with living cells should be shown in the main manuscript.
- Conclusion: Any interference with normal actin polymerization results in severe consequences for a number of cellular functions including phagolysosome maturation, cell mobility, endocytosis, etc.. Thus, rather than "suggesting that Arpc1B may govern neutrophil functions via various ways", couldn't the results simply be the consequence of it's role as part of Arp2/3 complex-mediated actin polymerization? This warrants more in-depth discussion.

Referee #2:

In this study, the authors investigated how the human fungal pathogen *Candida glabrata* (Cg) dampens the proinflammatory response of epithelial cells (EC). Through comprehensive and systemic analyses, the author discovered that GPI-linked aspartyl proteases (Yapsins) of Cg block EC-neutrophil signaling by causing the degradation of the EC protein Arpc1B, a subunit of the actin nucleator Arp2/3 complex. Arpc1B degradation leads to actin disassembly and impedes IL-8 secretion by ECs. Furthermore, the diminished IL-8 secretion inhibits neutrophil migration and protects Cg from neutrophil-mediated killing.

This paper reports a novel and important finding, shedding new light onto the mechanisms by which a prevalent fungal pathogen evades host immunity. Although the findings will be of great interest mainly to fungal researchers, they will also have a broader impact because of the connection of an actin assembly complex with specific signaling pathways that activate proinflammatory response. The data are solid and well-controlled. Multiple experiments were conducted for each major conclusion.

There are some minor issues that the authors should address. I suggest that the authors add line numbers in their revision, which will make it much easier for reviewers to indicate where changes are needed.

Page 2

line 5. aid Cg in maintaining cell wall integrity, pH balance, vacuole homeostasis, and survival in macrophages, as well as in fly and mouse hosts.

line 7. ...the cell wall. CgYPS1-11 deletion resulted....; delete the comma after 'processing'.

Page 3

Line 3 from the bottom. The word 'rescued' is confusing, should it be 'reduced' or 'abolished' the elevated secretion...?

Page 4

The first sentence of the second paragraph is too long and hard to understand. I suggest breaking it into two sentences. We also examined the activation status of the nuclear factor- κ B....

Cytokine levels were examined at 24 hpi while the activation status of three MAPKs was examined at 6 hpi. It would be nice to add a couple of earlier time points for cytokine levels, which will readers to know the trajectory of cytokine release activation.

Middle of the page. Rephrase the sentence: Altogether, these data suggest that Cg infection of ECs activates NF- κ B and suppresses p38 and ERK signaling, respectively,

Page 5

Spell out SFB

Page 6

The result 'siRNA-mediated knockdown of Arpc1B reduced Cgyps1-11 Δ adherence' is surprising. The authors thought it was due to altered membrane structure. However, it is hard to imagine this altered membrane structure does not affect internalization. Was it also observed during CK-666 treatment?

Page 7

Line 9 from the bottom, delete the 0 in '0contained'.

Page 12, the description 'The percentage adherence was determined by dividing the 2 h Cg CFUs by 0 h CFUs' does not seem to be correct. Shouldn't it be 'dividing the 2 h CFU by the total number of cells loaded into each well'?

In the sentence 'ECs were pre-treated with DMSO...', were all inhibitors used dissolved in DMSO?

Page 14

Line 2 from the bottom, delete the word 'insolubilized'.

Referee #3:

Candida glabrata expresses multiple cell wall adhesions, among which Epa1 has been shown to be involved in adherence to epithelial cells. This organism also produces multiple aspartyl proteases and among their functions are processing and cleaving Epa1 from the cell wall and thereby modulating Cg adherence with epithelial cells. The stated purpose of this study is to examine the role of Cg yapsins in epithelial uptake and survival. However there are many panels of distracting data that make the major story difficult to follow.

Fig. 1 repeats and somewhat extends the work of Cormack group in showing that Cg yapsins and Epa1 regulate adherence to epithelial cells, but do not show that some or all yapsins are responsible for this effect directly through Epa1, since it is possible that yapsins alter other cell wall adhesions as well as Epa1. Since Epa1 is not further examined, this data is irrelevant to the main findings. It also tends to undercut the finding that if Cgyps1-11 deletion elevated adherence, why are these mutants so defective in ingestion by epithelial cells? Lactose blocking experiments do not add much understanding. The new data is that endocytosis of Cg by epithelial cells is reduced in Cgyps1-11 deletion mutants and these mutants have reduced survival within epithelial cells.

Next, intracellular signaling upon internalization of Cg is examined, although the rationale for looking at this is not well defined. One concern with these experiments is how will the authors compare signaling between epithelial cells that have taken up a certain number of wt cells but only half the number of Cgyps1-11 deletion mutants. There must be some means of accounting for this especially since NF κ B and MAPK signaling activates proportionally to yeast form fungal load. NF κ B activation needs to be normalized to fungal cell numbers endocytose or possibly at various MOIs.

Another major concern is that the cell line A498 has very high basal cytokine levels for IL-8 and IL-6 and p38 phosphorylation, and that wt cell infection actually reduced signaling levels. Where these cell lines tested for mycoplasma contamination? P38 activation feeds into IL-8 and IL-1b release. Absence of IL-1b in EC need to explained as EC secrete IL-1B in response to proteases in *Candida* spp. As the cells were washed after 4h post infection, wt cells being less adherent to EC might have washed away compared to cgYps1-11 mutant strain, so it is possible that after 24 h, wt treated EC had less cytokine release

since they had reduced fungal load. However, they suggest that p38 activation is involved in higher killing of CgYps1-11 mutant strain, although no mechanism by which this would occur is proposed.

Next, interacting EC proteins with yps1 and yps7 are identified by pulldown assays. It is not stated why these two yapsins were selected, especially since Cormack's work suggested they have no effect individually on Epa1. Arpc1B was selected as an interacting protein since was associated with increased susceptibility to microbial infections but not with pathogenic fungi (this rationale is counterintuitive). Nevertheless, They found that Arpc1B protein was slightly diminished upon wt infection, but elevated by the CgYps1-11 mutant strain (3E) although the fold reduction as stated is misleading since it is change in fluorescence not proteins as shown in 3F. Fig. 3F shows no increase in Arpc 1B compared to control however, this could be a result of the lower uptake of this strain. Again as for inflammatory signaling, these data need to be normalized for number of Cg taken up by PCs or show differences upon changes in MOI. Data showing Yps1 catalytic activity and Arpc 1B protein levels should be in main body.

Next, they hypothesize that p38 signaling is related to Arpc 1B. While this is logical for wt infection with yapsins disrupting Arpc 1B having reduced signaling, why should infection with CgYps1-11 mutant strain that does not change Arpc 1B levels have any differences? Arpc 1B interaction in live cells is only shown for Cgwt infection 3H and interactions in 3I are for ectopic expression of Arpc 1B. Incubation of ICs with an actin inhibitor CK-666 improved viability of CgYps1-11 mutant strain and reduced inflammatory signaling- but could this be due to partial endocytosis without release of cells into the EC? This would be consistent with results in Fig 4D-G. Another EC line was used to delete p38 and Arpc 1B which reversed the adherence phenotype of Cg and increased viability of CgYps1-11 mutant strain and reduced inflammatory signaling. This again suggests that the endocytosis process is altered in which case Epa1 might have a relevant role which is not explored. Further data is shown that Arpc 1B is needed for actin assembly and p38 activation. Other experiments support p38 signaling to be downstream of Arpc 1B.

Experiments with neutrophil migration and other Candida pathogens do not add much to this story and should be removed.

Overall there is an exhausting amount of data that is very difficult reading and needs to be pared down to highlight the main points of this story.

** As a service to authors, EMBO Press provides authors with the ability to transfer a manuscript that one journal cannot offer to publish to another journal, without the author having to upload the manuscript data again. To transfer your manuscript to another EMBO Press journal using this service, please click on Link Not Available

Point-by-Point response to Referees' comments on original submission

Manuscript number: EMBOR-2023-58420V1

Manuscript title: GPI-anchored aspartyl proteases target the host actin nucleator complex protein to suppress epithelial innate immunity

Referee #1:

The manuscript by Sandip Patra and Rupinder Kaur convincingly demonstrates that *Candida glabrata* (Cg) yapsins affect Arpc1B (actin nucleator Arp2/3 complex subunit) in epithelial cells, leading to actin disassembly and preventing IL-8 secretion by ECs. The authors demonstrate this by different complimentary approaches. They also show that interaction with Arpc1B has consequences for p38 activation and chemokine release, and that, thereby, certain yapsins contribute to immune evasion and fungal survival in the context of epithelial cells. These findings are novel, convincing, and of interest to the medical mycology community.

We thank the referee for appreciating our work.

However, the in vivo relevance remains to be demonstrated, thereby somewhat limiting my excitement about the novel findings.

There are additional shortcomings in other aspects (see below) that should be addressed.

Major points:

1) Relevance of the in vitro findings for systemic candidiasis in mice: The authors show that IL-6 levels in kidneys of mice infected with the *Cgyps1-11ko* mutant are lower compared to wt-infected tissue. However, the subsequent part of the manuscript largely addresses differences in IL-8 production. This raises two important questions: Were levels of the murine IL-8 homologues KC and MIP-2 increased in kidneys of *Cgyps1-11ko*-infected tissues? And if yes: Was this functionally relevant, eg was increased neutrophil recruitment observed? This is critically, as interactions in cell culture might diverge significantly from interactions in vitro, due to a number of reasons.

We are grateful to the referee for raising these excellent questions. Following referee's suggestion, we checked levels of the murine IL-8 homologues, KC and MIP-2, in kidney homogenates of uninfected, *Candida glabrata* (Cg) wild-type (*wt*)- and *Cgyps1-11Δ*-infected mice. Both KC and MIP-2 levels were higher in kidney homogenates of *Cgyps1-11Δ*-infected mice. Consistent with this, hematoxylin-eosin staining and anti-CD45 antibody staining-based histological analysis revealed that immune cell infiltration was highly increased in kidneys of the *Cgyps1-11Δ*-infected mice, compared to the uninfected and *wt*-infected mice. These new data demonstrate the in vivo relevance of in vitro findings, and are shown in Figure 3.

2) Connection of adhesion, internalization, and Arpc1B:

• Arpc1B levels in wt-infected A-498 cells were reduced within 15 min (significant after 30 min) of incubation with Cg (Fig. 3F) - given that Cg uptake is mediated by endocytosis, did endocytosis already occur within this time frame? There's no further decrease of Arpc1B levels over time. It seems unlikely that endocytosis is completed within 15 min, and therefore, these results would argue against internalization being required for interference host cell physiology. **We agree with referee's point that endocytosis is unlikely to be completed within 30 min of Cg contact with the host epithelial cells (ECs), and thus, internalization is not a prerequisite to alter host cell physiology. We now have examined Arpc1B reduction in ECs upon Cg infection in more detail. For this, we first identified, via molecular docking, two arginine residues, Arg-74 and Arg-142 in Arpc1B, that interacted with CgYps1. Notably, CgYps1 has recently been shown to cleave at an arginine residue in its yeast substrate CgPst2 [Battu *et al* (2021) *PLoS Pathog*, 17, e1009355]. Next, we expressed the mutant Arpc1B proteins, that contained alanine in place of Arg-74 or Arg-142, in Arpc1B^{-/-} cells. We found impaired actin filaments and decreased F/G actin ratio, indicating the role of Arg-74 and Arg-142 residues in Arpc1B functions in maintaining the actin cytoskeletal network. Importantly, the Arg-142 residue was also found to be essential for Cg-induced degradation of Arpc1B, thereby raising the possibility of CgYps1-dependent processing of Arpc1B to occur at Arg-142, with Arg-142 residue also being pivotal to Arpc1B functions in F-actin assembly. These data, shown in Fig. 7C-E, along with our earlier inhibitor data raise two possibilities. First, Arpc1B reduction could be mediated by the secreted forms of CgYapsins upon Cg-EC contact. In this context, it is worth noting that CgYapsins have recently been detected in Cg secretome and biofilm matrix proteome, but the secreted proteolytic activity is yet to be reported in Cg [Rasheed *et al* (2020) *J Proteome Res* 19, 49-63; Gonçalves *et al* (2021) *Biochem J* 478, 961-74]. Alternatively, in response to Cg-induced mechanical stress, Arp2/3 (Actin-related protein 2/3) complex, by altering actin cortex and plasma membrane association, may aid in the formation of actin-driven membrane protrusions/extensions/invaginations, that may bring CgYapsins and Arpc1B in close proximity. Notably, Arpc2/3 complex-mediated actin polymerization has been implicated in maintaining plasma membrane shape homeostasis in response to membrane deformations [Quiroga *et al* (2023) *Elife* 12, e72316], and Arc5 (Arp2/3 complex subunit) has been found to be localized at the cell periphery [Kahr *et al* (2017) *Nat Commun* 8, 14816]. However, further studies are warranted to test these possibilities for Cg-EC interplay. These points are now discussed in the manuscript.**

• How can siRNA-mediated knockdown of Arpc1B reduce Cgyps1-11ko adherence, given that Arpc1B is located intracellularly? Is there any evidence that knockdown of Arpc1B indeed affects cell membrane structure - and that Cg adheres directly to the cell membrane rather than surface proteins or other surface molecules?

siRNA-mediated knockdown of Arpc1B caused damage to the cell membrane, as reflected in the increased Annexin-V membrane staining (Fig. S4I). Further, although adhesins are likely to mediate Cg adherence to ECs through specific host receptors, it is possible that Arpc1B knockdown affects actin nucleation underneath the plasma membrane (PM), leading to perturbed cell membrane structures. This may impact the

distribution/localization of adhesin receptors at the PM which could adversely affect the adherence of *Cgyps1-11Δ* cells that contain high amounts of cell surface adhesins. In this context, it is noteworthy that Arp2/3 complex-mediated branched actin networks restrain and/or regulate diffusional barriers for PM proteins and receptors, with Arp2/3 complex activity also being implicated in governing B cell responses specifically to the spatially-confined membrane-bound antigens [Mattila *et al* (2016) *J Cell Biol* 212, 267-80; Bolger-Munro *et al* (2019) *Elife* 8, e44574]. These points are now added to the manuscript.

3) It should be made clear, that the different approaches used to reduce levels of Arpc1B (a strength of the manuscript) do not result in full reversal of the *Cgyps1-11ko* phenotype to wt:

- CK-666 treatment (Fig. 4) only partially restored *Cgyps1-11ko* viability in A-498, and led to an increase in IL-6 and IL-8 production in response to WT infection. This contradicts some of the previous and later conclusions around the role of Arpc1B - or it indicates that CK-666 has additional effects on other factors that affect survival and cytokine responses.

- siRNA knockdown only partially rescued *Cgyps1-11ko* viability (Fig. 4E)

We have clarified these points, by highlighting the partial reversal of different phenotypes as well as discussing the possibility of CK-666 affecting other factors that govern *Cg* survival and cytokine release.

4) It seems that activation of p38 only partially depends on Arpc1B - this should be addressed:

- Loss of p38 had a much more pronounced effect on *Cgyps1-11ko* survival than loss of Arpc1B (Fig. 4J, K). According to Fig. S5B, the clones selected for further experiments lacked any detectable Arpc1B expression, excluding incomplete repression of Arpc1B as the reason. Thus, it seems that pathways independent of Arpc1B likely activate p38 and p38-associated *Cg* inactivation within epithelial cells.

We thank the referee for raising this point. It is now indicated in the manuscript that the more pronounced effect of p38 on *Cgyps1-11Δ* survival, compared to Arpc1B loss, suggests that p38 activation modulation is probably carried out by both Arpc1B-dependent and Arpc1B-independent mechanisms, and, that p38 may play a major role in regulating *Cg*-EC interplay.

- Fig. 4L, M: Only Arpc1B ko reduced IL-8 levels following *Cgyps1-11ko* infection to that of wt-infected cells - p38 ko resulted in levels comparable to non-infected controls, but the reduction of IL-8 observed with the WT was not reached. Again, this might indicate that p38 and Arpc1B are not within the same pathway or that additional pathways are involved.

As suggested, it is now discussed that the differential reversal of increased IL-8 secretion in *Cgyps1-11Δ*-infected ECs upon Arpc1B deletion (IL-8 levels similar to wt-infected ECs) and p38 deletion (IL-8 levels similar to uninfected ECs) indicate that Arpc1B and p38 MAPK-mediated signalling is not the sole pathway to control IL-8 secretion. Alternatively, Arpc1B and p38 MAPK may participate in additional pathways that control cytokine response upon *Cg* infection.

5) I'm not convinced of the validity of the neutrophil migration assay: Fig. 6A: When adding up the fluorescence intensity in the upper and lower chamber, the total fluorescence differs between the different treatment groups: It is much lower for Cg wt infected epithelial cells than for either uninfected or Cgyps1-11ko infected cells. Also, the uninfected signal is roughly a third lower than that of Cgyps1-11ko infection. One likely explanation is differences in neutrophil viability and/or issues with the stability of the fluorescent labelling. Thus, these results do not allow any confident conclusions on actual neutrophil migration. Furthermore, the exact experimental set up for Fig. S5E is not clear from the figure legend and information in materials and methods.

We thank the referee for raising this important point. As suggested by the referee, the observed variable fluorescence intensity was likely due to technical issues. Thus, we repeated the experiment, and found similar total fluorescence values across different samples (Fig. 8A). These results corroborate our earlier finding that neutrophil migration is higher towards *Cgyps1-11Δ*-infected ECs, compared to *wt*-infected and uninfected ECs. Further, in line with referee's suggestion, the experimental set up for neutrophil migration assay for Fig. S6E (Fig. S5E in the original manuscript) has been described in the figure legend.

6) The part on other *Candida* pathogens is confusing regarding induction of p38 phosphorylation by *C. albicans*: The western blot in Fig. S6B shows a very strong band for P-p38, a weakened p38 band, yet according to the bar next to the western blot, the P-p38 to total p38 ration is lower than for uninfected cells. This contradicts published findings by other groups that have demonstrated robust activation of p38 by *C. albicans* and it's link to secretion of IL-8 that is reliably induced by this fungus (examples:<https://pubmed.ncbi.nlm.nih.gov/35380879/> <https://journals.aai.org/jimmunol/article/179/12/8435/79462/Candida-albicans-Triggers-Activation-of-Distinct> ; <https://journals.aai.org/jimmunol/article/171/6/3047/71449/Candida-albicans-Induces-Selectively>).

The P-p38 strong band in Fig. S6B Western blot was non-specific. Regarding published findings, we have repeated our Western analyses, but still observed no activation of p38 in *C. albicans*-infected A-498 cells at 1:1 MoI at 6 h post-infection. We are enclosing these data herewith (Fig. R1) for referee's perusal, with the strong P-p38 band being marked as non-specific. It is possible that p38 activation may be regulated by different pathways in different cell lines or p38 activation kinetics is cell line and/or MoI context-dependent. Of note, the above-mentioned papers have examined p38 activation in TR146 Oral Epithelial cells, HUVEC cells and HeLa cells. We also request the editor and the referee to note that based on the Referee 3' s suggestion to focus on main findings of the study, data pertaining to adherence, p38 phosphorylation and Arpc1B reduction, upon infection with other *Candida* species, have been removed from the manuscript.

Minor points:

- Why was a human gastric (AGS) epithelial cell line used? There's no evidence that Cg translocates from the stomach - intestinal cells would appear more relevant.

AGS cell line was used to show that CgYapsin loss-associated adherence is not specific to kidney epithelial cells. Following referee's suggestion, we have now checked Cg adherence to intestinal epithelial (Caco2) cells, and found increased and decreased adherence of *Cgyts1-11Δ* and *epa1Δ* cells, respectively, compared to that of *wt* cells. These new data are shown in Fig. S7A, and underscore CgYapsin functions in host adherence across different epithelial cell types.

- Fixation with formaldehyde alters protein structures. I therefore would consider these experiments of only limited relevance and show them as supplementary data. The data with living cells should be shown in the main manuscript.

As suggested, the adherence with live A-498 cells is shown in Figure 1A in the main manuscript. Please note that after showing live A-498 cell adherence data (Fig. 1A), the statement "Importantly, the Cg adherence pattern was similar for formaldehyde-fixed A-498 cells (Fig. S1A), suggesting that formaldehyde fixation largely preserves the cell surface architecture" has been added to the manuscript.

- Conclusion: Any interference with normal actin polymerization results in severe consequences for a number of cellular functions including phagolysosome maturation, cell mobility, endocytosis, etc.. Thus, rather than "suggesting that Arpc1B may govern neutrophil functions via various ways", couldn't the results simply be the consequence of it's role as part of Arp2/3 complex-mediated actin polymerization? This warrants more in-depth discussion.

We are grateful to the referee for this excellent suggestion. This section has been rephrased, and it now reads as "Furthermore, Arpc1B deficiency, due to a homozygous complex frameshift mutation in a combined immunodeficiency patient, led to F-actin polymerization defect and impeded neutrophil movement, thereby suggesting that Arpc1B may govern neutrophil functions via many ways. However, it is also possible that the impaired neutrophil recruitment is one of the facet that Arpc1B governs, and that, Arpc1B contributions to Arp2/3 complex-mediated actin polymerization may play a more critical role in controlling Cg infections by regulating other cellular processes including cell mobility and endocytosis".

Referee #2:

In this study, the authors investigated how the human fungal pathogen *Candida glabrata* (Cg) dampens the proinflammatory response of epithelial cells (EC). Through comprehensive and systemic analyses, the author discovered that GPI-linked aspartyl proteases (Yapsins) of Cg block EC-neutrophil signaling by causing the degradation of the EC protein Arpc1B, a subunit of the actin nucleator Arp2/3 complex. Arpc1B degradation leads to actin disassembly and impedes IL-8 secretion by ECs. Furthermore, the diminished IL-8 secretion inhibits neutrophil migration and protects Cg from neutrophil-mediated killing.

This paper reports a novel and important finding, shedding new light onto the mechanisms by which a prevalent fungal pathogen evades host immunity. Although the findings will be of great interest mainly to fungal researchers, they will also have a broader impact because of the connection of an actin assembly complex with specific signaling pathways that activate proinflammatory response. The data are solid and well-controlled. Multiple experiments were conducted for each major conclusion.

We thank the referee for appreciating our work.

There are some minor issues that the authors should address. I suggest that the authors add line numbers in their revision, which will make it much easier for reviewers to indicate where changes are needed.

Page 2: line 5. aid Cg in maintaining cell wall integrity, pH balance, vacuole homeostasis, and survival in macrophages, as well as in fly and mouse hosts.

Corrected, as suggested.

line 7. ...the cell wall. CgYPS1-11 deletion resulted....; delete the comma after 'processing'.

Done, as suggested.

Page 3: Line 3 from the bottom. The word 'rescued' is confusing, should it be 'reduced' or 'abolished' the elevated secretion...?

We thank the referee for this suggestion. The word 'rescued' is replaced with the word 'abolished'.

Page 4: The first sentence of the second paragraph is too long and hard to understand. I suggest breaking it into two sentences. We also examined the activation status of the nuclear factor- κ B....

We have rephrased the sentence. The text now read as “To investigate the signalling pathways that could contribute to the differential cytokine response of A-498 to *wt* and *Cgyps1-11Δ* infection, we checked the activation status of three serine-threonine MAP kinases, extracellular signal-regulated kinase (ERK), stress-activated protein kinases c-Jun N-terminal kinase (JNK) and p38 kinase. In addition, we also examined the activation status of nuclear factor- κ B (NF- κ B) signalling, with NF- κ B consisting of two subunits, p50 and p65¹⁷”.

Cytokine levels were examined at 24 hpi while the activation status of three MAPKs was examined at 6 hpi. It would be nice to add a couple of earlier time points for cytokine levels, which will readers to know the trajectory of cytokine release activation.

We thank the referee for this excellent suggestion. We have now measured IL-6 and IL-8 cytokine secretion at 2, 4, 6 and 8 h post-infection (hpi). We found no increase in IL-6 and IL-8 levels at early time points of 2 and 4 h. However, IL-6 and IL-8 levels were increased and decreased in *Cgyps1-11Δ*- and *wt*-infected ECs, respectively, from 6 h time point onwards. These new data are shown in Fig. S7C.

Middle of the page. Rephrase the sentence: Altogether, these data suggest that *Cg* infection of ECs activates NF- κ B and suppresses p38 and ERK signaling, respectively,

Done, as suggested.

Page 5: Spell out SFB

Done, as suggested.

Page 6: The result 'siRNA-mediated knockdown of Arpc1B reduced *Cgyps1-11Δ* adherence' is surprising. The authors thought it was due to altered membrane structure. However, it is hard to imagine this altered membrane structure does not affect internalization. Was it also observed during CK-666 treatment?

We agree with the referee that it seems unlikely that the altered membrane structure affects adherence but not internalization. Thus, we checked if siRNA-mediated knockdown of Arpc1B causes damage to the cell membrane. We found increased Annexin-V membrane staining (Fig. S4I), suggestive of phosphatidylserine exposure probably due to a damaged cell membrane. Further, it is also possible that Arpc1B knockdown affects actin nucleation underneath the plasma membrane (PM), leading to perturbed cell membrane structures. This may impact the distribution/localization of adhesin receptors at the PM which could adversely affect the adherence of *Cgyps1-11Δ* cells that contain high amounts of cell surface adhesins. In this context, it is noteworthy that Arp2/3 complex-mediated branched actin networks restrain and/or regulate diffusional barriers for PM proteins and receptors, with Arp2/3 complex activity also being implicated in governing B cell responses specifically to the spatially-confined membrane-bound antigens [Mattila *et al* (2016) *J Cell Biol* 212, 267-80; Bolger-Munro *et al* (2019) *Elife* 8, e44574].

Next, following referee's suggestion, we checked *Cg* adherence upon CK-666 treatment. We found that while CK-666 had no effect on *wt* adherence, it decreased *Cgyps1-11Δ* adherence. These new data, shown in Fig. S4H, are consistent with Arpc1B knockdown results.

Page 7: Line 9 from the bottom, delete the 0 in '0contained'.

Corrected, as suggested.

Page 12, the description 'The percentage adherence was determined by dividing the 2 h *Cg* CFUs by 0 h CFUs' does not seem to be correct. Shouldn't it be 'dividing the 2 h CFU by the total number of cells loaded into each well'?

We thank the referee for raising this important point. We wish to state that the 0 h *Cg* CFUs actually reflect the number of *Cg* cells that were added to ECs. This point is now clarified in the manuscript.

In the sentence 'ECs were pre-treated with DMSO...', were all inhibitors used dissolved in DMSO?

Yes, all inhibitors were dissolved in DMSO. This point is now indicated in the manuscript.

Page 14: Line 2 from the bottom, delete the word 'insolubilized'.

Deleted, as suggested.

Referee #3:

Candida glabrata expresses multiple cell wall adhesions, among which Epa1 has been shown to be involved in adherence to epithelial cells. This organism also produces multiple aspartyl proteases and among their functions are processing and cleaving Epa1 from the cell wall and thereby modulating Cg adhesion with epithelial cells. The stated purpose of this study is to examine the role of Cg yapsins in epithelial uptake and survival. However there are many panels of distracting data that make the major story difficult to follow.

Fig. 1 repeats and somewhat extends the work of Cormack group in showing that Cg yapsins and Epa1 regulate adherence to epithelial cells, but do not show that some or all yapsins are responsible for this effect directly through Epa1, since it is possible that yapsins alter other cell wall adhesions as well as Epa1. Since Epa1 is not further examined, this data is irrelevant to the main findings.

It also tends to undercut the finding that if *Cgygs1-11* deletion elevated adherence, why are these mutants so defective in ingestion by epithelial cells? Lactose blocking experiments do not add much understanding. The new data is that endocytosis of Cg by epithelial cells is reduced in *Cgygs1-11* deletion mutants and these mutants have reduced survival within epithelial cells.

We thank the referee for this suggestion, and have removed the data pertaining to Epa1 and lactose treatment to streamline the manuscript.

Next, intracellular signaling upon internalization of Cg is examined, although the rationale for looking at this is not well defined. One concern with these experiments is how will the authors compare signaling between epithelial cells that have taken up a certain number of wt cells but only half the number of *Cgygs1-11* deletion mutants. There must be some means of accounting for this especially since NFκB and MAPK signaling activates proportionally to yeast form fungal load. NFκB activation needs to be normalized to fungal cell numbers endocytose or possibly at various MOIs.

We thank the referee for raising these important points. We had examined intracellular signaling pathways, because *wt* and *Cgygs1-11Δ* infection invoked differential cytokine response in ECs. This point is now indicated in the manuscript.

Further, as suggested by the referee, we have checked cytokine secretion and Arpc1B reduction in ECs at higher MoI, and found similar results. These new data are shown in Fig. S1E, S1F and S4E.

Another major concern is that the cell line A498 has very high basal cytokine levels for IL-8 and IL-6 and p38 phosphorylation, and that wt cell infection actually reduced signaling levels. Where these cell lines tested for mycoplasma contamination?

We thank the referee for raising this important point. We had checked, and found that our cell lines were mycoplasma free. This point is now indicated in the manuscript.

P38 activation feeds into IL-8 and IL-1b release. Absence of IL-1b in EC need to explained as EC secrete IL-1B in response to proteases in *Candida* spp.

Done, as suggested. The text now reads as “Of note, the secreted aspartyl protease Sap6 in *C. albicans* has recently been reported to invoke IL-1 β secretion in human oral epithelial cells which was mediated by p38 MAPK. [Kumar *et al* (2022) *Front Immunol* 13, 912748]. However, despite a hyperphosphorylated p38, we did not observe IL-1 β secretion in *Cgyps1-11 Δ* -infected A-498. Similarly, although *Cg* is known to activate IL-1 β gene expression weakly in oral epithelial cells at 12 h post-infection [Schaller *et al* (2002) *J Invest Dermatol*, 118, 652-7], IL-1 β secretion in *wt*-infected A-498 cells was not observed. These discrepancies could either be due to a modest IL-1 β release by A-498 cells, that could not be detected with our assay, or IL-1 β secretion activation could be cell line context-dependent”.

As the cells were washed after 4h post infection, wt cells being less adherent to EC might have washed away compared to *cgYps1-11* mutant strain, so it is possible that after 24 h, wt treated EC had less cytokine release since they had reduced fungal load. However, they suggest that p38 activation is involved in higher killing of *CgYps1-11* mutant strain, although no mechanism by which this would occur is proposed.

We thank the referee for raising this point. Despite the lower number of adherent *wt* cells at 2 h, the number of ingested *wt* cells is higher after 4 and 24 h of A-498 infection (Fig. 1C). Thus, it is unlikely that the reduced fungal load accounts for the diminished cytokine response in *wt*-infected A-498. Notably, we saw diminished cytokine secretion even at the higher MoI (Fig.S1E and F). These new data suggest that the less cytokine release by A-498 cells is a bonafide outcome of *Cg*-EC interaction.

Regarding p38 activation leading to *Cgyps1-11 Δ* killing, we propose that it could in part be due to increased reactive oxygen species (ROS) production, as p38 activation is associated with increased ROS levels [Canovas and Nebreda (2021) *Nat Rev Mol Cell Biol* 22, 346-66]. This point is now added to the manuscript.

Next, interacting EC proteins with *yps1* and *yps7* are identified by pulldown assays. It is not stated why these two yapsins were selected, especially since Cormack's work suggested they have no effect individually on Ep1.

We had selected *CgYps1* and *CgYps7*, as when expressed individually, these were able to reverse the hyperadherence phenotype of the *Cgyps1-11 Δ* mutant (Fig. 1B). This point is now added to the manuscript.

Arpc1B was selected as an interacting protein since was associated with increased susceptibility to microbial infections but not with pathogenic fungi (this rationale is counterintuitive).

We apologise for not writing the sentence correctly. We wanted to convey the message that the role of Arpc1B in cellular defense against pathogenic fungi is not known. This point has now been rephrased in the manuscript.

Nevertheless, They found that Arpc1B protein was slightly diminished upon wt infection, but elevated by the CgYps1-11 mutant strain (3E) although the fold reduction as stated is misleading since it is change in fluorescence not proteins as shown in 3F. Fig. 3F shows no increase in Arpc 1B compared to control however, this could be a result of the lower uptake of this strain. Again as for inflammatory signaling, these data need to be normalized for number of Cg taken up by PCs or show differences upon changes in MOI.

We agree that Arpc1B expression, as measured by immunofluorescence, is higher in *Cgyps1-11Δ*-infected ECs, as compared to both uninfected and *wild-type*-infected A-498 cells [Fig. 5E (Fig. 3E in the original manuscript)], and Arpc1B proteins levels in *Cgyps1-11Δ*-infected cells (Fig. S4D and E), as determined by Western blot, are similar and higher, as compared to uninfected and *wild-type*-infected A-498 cells . However, since Fig. 5F (Fig. 3F in the original manuscript)] illustrates Arpc1B expression, as compared to the zero time-point (0 h) of infection for each strain, it would not be appropriate to use these data to infer Arpc1B expression differences between uninfected and *C. glabrata*-infected cells.

Further, as suggested by the referee, we have checked reduction in Arpc1B levels using three different MoIs, and found the same results, viz., infection with *wt* cells leads to a reduction in Arpc1B levels in A-498 cells, irrespective of the MoI used. These new data are shown in Fig. S4E.

Additionally, we identified, via molecular docking, CgYps1-interacting Arginine-142 residue in Arpc1B that is pivotal to both Arpc1B functions in maintaining the actin network and Arpc1B reduction upon *Cg*-EC co-culturing. These data (shown in Fig. 7C-E) reinforce the role of Arpc1B and actin reorganization in governing *Cg*-EC interaction.

Data showing Yps1 catalytic activity and Arpc 1B protein levels should be in main body.

We thank the referee for this excellent suggestion. Data illustrating the requirement of CgYps1 catalytic activity for reduction in Arpc1B levels have been moved to the main body (Fig. 5G).

Next, they hypothesize that p38 signaling in related to Arpc 1B. While this is logical for wt infection with yapsins disrupting Arpc 1B having reduced signaling, why should infection with CgYps1-11 mutant strain that does not change Arpc 1B levels have any differences?

We propose that Arpc1B reduction, upon exposure to *Cg*, is a natural response of ECs to repress p38 activation. In the absence of this reduction, p38 MAPK is activated, and thus, *Cgy ps1-11Δ* infection results in p38 hyperphosphorylation and increased cytokine secretion.

Arpc 1B interaction in live cells is only shown for *Cg*wt infection 3H and interactions in 3I are for ectopic expression of Arpc 1B. Incubation of ICs with an actin inhibitor CK-666 improved viability of *CgYps1-11* mutant strain and reduced inflammatory signaling- but could this be due to partial endocytosis without release of cells into the EC? This would be consistent with results in Fig 4D-G. Another EC line was used to delete p38 and Arpc 1B which reversed the adherence phenotype of *Cg* and increased viability of *CgYps1-11* mutant strain and reduced inflammatory signaling. This again suggests that the endocytosis process is altered in which case *Epa1* might have a relevant role which is not explored. Further data is shown that Arpc 1B is needed for actin assembly and p38 activation. Other experiments support p38 signaling to be downstream of Arpc 1B.

We thank the referee for this suggestion. Since Arpc1B (Fig. 7F) or p38 (Fig. 7G) deletion had no significant effect on *Cg* ingestion, it is unlikely that a defect in *Cg* endocytosis results in increased inflammatory signalling in *Cgy ps1-11Δ* -infected ECs. However, in light of referee's points, we have now indicated in the manuscript that "Based on our data, we propose that a single mechanism may not govern *Cg* adherence, internalization, survival and modulation of host signalling pathways, rather it is a combination of multiple regulators that work in concert to control the outcome of *Cg* infections. This notwithstanding, our findings do establish Arpc1B and p38 MAPK as key players in *Cg*-EC interplay".

Experiments with neutrophil migration and other *Candida* pathogens do not add much to this story and should be removed.

Since the first referee had suggested to show the in vivo relevance of in vitro findings, we had measured cytokine levels in kidney homogenates of *Cg*-infected mice. We found that kidney homogenates of the *Cgy ps1-11Δ*-infected mice contained higher levels of IL-6 and KC and MIP-2 (murine IL-8 homologues), compared to the *wt*-infected mice. Additionally, an increased immune cell infiltration was observed in kidneys of the *Cgy ps1-11Δ*-infected mice that could contribute to the increased mutant clearance in the mouse model of systemic candidiasis [Kaur *et al* (2007) *Proc Natl Acad Sci*, 104, 7628-33]. In view of these in vivo data (shown in Fig. 3), we believe that neutrophil migration results are important, and would like to retain these in the manuscript. We hope that the editor and the referee will agree with us.

Overall there is an exhausting amount of data that is very difficult reading and needs to be pared down to highlight the main points of this story.

We thank the referee for this suggestion. We have substantially streamlined the manuscript by removing data pertaining to *Epa1* adhesin, lactose treatment, adherence

of *Candida* species, and Arpc1B reduction and p38 activation upon infection with other *Candida* species, that were not central to the story. Accordingly, the manuscript text has also been restructured.

We thank the editor and reviewers for carefully reviewing our manuscript. We believe that incorporation of your comments has significantly improved our manuscript and made it more structured, readable and clearer.

Figure for referee with unpublished data and its description has been removed upon request by the authors.

Dear Dr. Kaur,

Thank you for the re-submission your manuscript to EMBO reports. I have now received the reports from the three referees that were asked to re-evaluate your study, which can be found at the end of this email.

As you will see, the referees now fully support the publication of your manuscript in EMBO reports. Referee #1 has remaining concerns I ask you to address in a final revised manuscript. Please also provide a final p-b-p-response addressing the remaining points of referee #1.

Moreover, I have these editorial requests.

- Please provide your final manuscript text file as .docx formatted file (including legends for main figures, EV figures and tables), but without the figures included. Figure legends should be compiled at the end of the manuscript text.

- Please upload individual production quality figure files as .eps, .tif, .jpg (one file per figure), of main figures and EV figures. Please upload these as separate, individual files upon re-submission.

- Please also move the supplementary tables to the Appendix.

- Please upload a completed author checklist, which you can download from our author guidelines (<https://www.embopress.org/page/journal/14693178/authorguide>). Please insert page numbers in the checklist to indicate where the requested information can be found in the manuscript.

- We request that primary datasets produced in this study (e.g. RNA-seq, ChIP-seq, structural and array data) are deposited in an appropriate public database. If no primary datasets have been deposited, please also state this in a dedicated section (e.g. 'No primary datasets have been generated and deposited'), see below.

The accession numbers and database should be listed in a formal "Data Availability" section (placed after Materials & Methods) that follows the model below. This is now mandatory (like the COI statement). Please note that the Data Availability Section is restricted to new primary data that are part of this study. This section is mandatory. As indicated above, if no primary datasets have been deposited, please state this in this section

Data availability

- We now request the publication of original source data with the aim of making primary data more accessible and transparent to the reader. Our source data coordinator will contact you to discuss which figure panels we would need source data for and will also provide you with helpful tips on how to upload and organize the files.

- Our journal encourages inclusion of *data citations in the reference list* to directly cite datasets that were re-used and obtained from public databases. Data citations in the article text are distinct from normal bibliographical citations and should directly link to the database records from which the data can be accessed. In the main text, data citations are formatted as follows: "Data ref: Smith et al, 2001" or "Data ref: NCBI Sequence Read Archive PRJNA342805, 2017". In the Reference list, data citations must be labeled with "[DATASET]". A data reference must provide the database name, accession number/identifiers and a resolvable link to the landing page from which the data can be accessed at the end of the reference. Further instructions are available at: <http://www.embopress.org/page/journal/14693178/authorguide#referencesformat>

- Regarding data quantification and statistics, please make sure that the number "n" for how many independent experiments were performed, their nature (biological versus technical replicates), the bars and error bars (e.g. SEM, SD) and the test used to calculate p-values is indicated in the respective figure legends (also for EV figures and all those in an Appendix). Please also check that all the p-values are explained in the legend, and that these fit to those shown in the figure. Please provide statistical testing where applicable. Please avoid the phrase 'independent experiment', but clearly state if these were biological or technical replicates. Please also indicate (e.g. with n.s.) if testing was performed, but the differences are not significant. In case n=2, please show the data as separate datapoints without error bars and statistics. See also: <http://www.embopress.org/page/journal/14693178/authorguide#statisticalanalysis>

- Please add to each legend (main, EV and Appendix figures, where applicable) a 'Data Information' section explaining the statistics used or providing information regarding replicates and scales. See:

- Please add scale bars of similar style and thickness to microscopic images, using clearly visible black or white bars (depending on the background). Please place these in the lower right corner of the images themselves. Please do not write on or near the bars in the image but define the size in the respective figure legend.

- Please note our reference format:

- We updated our journal's competing interests policy in January 2022 and request authors to consider both actual and perceived competing interests. Please review the policy <https://www.embopress.org/competing-interests> and update your competing interests if necessary. Please name this section 'Disclosure and Competing Interests Statement' and put it after the Acknowledgements section.

- We now use CRediT to specify the contributions of each author in the journal submission system. CRediT replaces the author contribution section. Please use the free text box to provide more detailed descriptions and do NOT provide your final manuscript text file with an author contributions section. See also our guide to authors: <https://www.embopress.org/page/journal/14693178/authorguide#authorshipguidelines>

- Please make sure that all the funding information is also entered into the online submission system and is complete and similar to the one in the manuscript text file (in the Acknowledgements).

- We would encourage you to use 'Structured Methods', our new Materials and Methods format. According to this format, the Materials and Methods section should include a Reagents and Tools Table (listing key reagents, experimental models, software, and relevant equipment and including their sources and relevant identifiers), uploaded as separate file, followed by a Methods and Protocols section in which we encourage the authors to describe their methods using a step-by-step protocol format with bullet points, to facilitate the adoption of the methodologies across labs. More information on how to adhere to this format as well as downloadable templates (.doc) for the Reagents and Tools Table can be found in our author guidelines (section 'Structured Methods'):

- Please provide a final title with not more than 100 characters (including spaces).

- Please provide the abstract written in present tense throughout.

- Please restrict the keywords to 5 and order the manuscript sections like this, using these names:
Title page - Abstract - Keywords - Introduction - Results - Discussion - Methods - Data availability section - Acknowledgements -
Disclosure and Competing Interests Statement - References - Figure legends - Expanded View Figure legends

In addition, I would need from you:

- a short, two-sentence summary of the manuscript
- three to four short bullet points (two lines) that highlighting the key findings of your study
- a schematic summary figure (in jpeg or tiff format with the exact width of 550 pixels and a height of not more than 400 pixels) that can be used as a visual synopsis on our website.

I look forward to seeing a revised version of your manuscript when it is ready. Please let me know if you have questions or comments regarding the revision.

Besy,

Referee #1:

By revising their manuscript, Patr and Kaur addressed most of my previous critical remarks. I think, however, that the following point has not yet been sufficiently addressed:
The authors show tissue sections stained with HE or a CD45-antibody, respectively, to show that "immune cell infiltration was highly increased in kidneys of the Cgyyps1-11Δ-infected mice, compared to the uninfected and wt-infected mice." These new data demonstrate the in vivo relevance of in vitro findings, and are shown in Figure 3. " According to the figure legend, "Yellow arrowheads mark polymorphonuclear neutrophils infiltrated into the tissue." However, I'm not convinced that the arrows do indeed capture bona fide neutrophils, or that all neutrophils are indicated. This highlights the problem with the analysis performed: Selected histology slides are shown, a quantitative analysis is lacking, and while CD45 is a suitable marker for leukocytes, quantification of neutrophils should be performed using a specific marker, not CD45. I'm not convinced by the presented data, either image analysis of a larger number of slides on a comprehensive number of animals from all groups or flow cytometry should be performed to substantiate the authors' claim.

Referee #2:

I appreciate the authors' efforts in addressing the issues I raised. I am satisfied with their revision.

Referee #3:

The investigators have addressed all my concerns in a thoughtful and responsive manner. I found the streamlined manuscript to be much more readable.

Point-by-Point response to Referees' comments on original submission

Manuscript number: EMBOR-2023-58420V2

Manuscript title: GPI-anchored aspartyl proteases target the host actin nucleator complex protein to suppress epithelial innate immunity

We request the editor and referees to please note that the raw mass spectrometry data have been submitted to the ProteomeXchange database, with the dataset identifier, PXD053938. The reviewer account information at PRIDE repository (www.ebi.ac.uk/pride/) is as follows:

Username: [redacted]

Password: [redacted]

Referee #1:

By revising their manuscript, Patr and Kaur addressed most of my previous critical remarks. I think, however, that the following point has not yet been sufficiently addressed:

The authors show tissue sections stained with HE or a CD45-antibody, respectively, to show that "immune cell infiltration was highly increased in kidneys of the *Cgyps1-11Δ*-infected mice, compared to the uninfected and wt-infected mice." These new data demonstrate the in vivo relevance of in vitro findings, and are shown in Figure 3. " According to the figure legend, "Yellow arrowheads mark polymorphonuclear neutrophils infiltrated into the tissue." However, I'm not convinced that the arrows do indeed capture bona fide neutrophils, or that all neutrophils are indicated. This highlights the problem with the analysis performed: Selected histology slides are shown, a quantitative analysis is lacking, and while CD45 is a suitable marker for leukocytes, quantification of neutrophils should be performed using a specific marker, not CD45. I'm not convinced by the presented data, either image analysis of a larger number of slides on a comprehensive number of animals from all groups or flow cytometry should be performed to substantiate the authors' claim.

We thank the referee for appreciating our efforts in revising the manuscript. Regarding data shown in Figure 3, we agree with the reviewer that all neutrophils were not marked in the Figure. We have rectified this error by stating in the Figure legend that 'Yellow arrowheads mark representative polymorphonuclear neutrophils'. Please note that the original Figure 3D and E are Expanded View Figure 2 in the revised manuscript. Additionally, to strengthen our conclusions, we have stained mouse kidney sections with anti-Ly6G antibody, with Ly6G being a neutrophil-specific marker. We found 2-fold higher number of Ly6G positive neutrophils in kidneys of *Cgyps1-11Δ* infected mice, compared to wild type-infected mice. Representative immunohistochemical staining images, along with quantification of Ly6G positive neutrophils are shown in panel D of Figure 3 and discussed on Page 4 in the revised manuscript. The Ly6G staining experiment has been conducted on kidneys of six *Candida glabrata*-infected mice, and a minimum of 100 cells, across different fields of stained kidney sections, were counted for each mouse for quantification. This

information has been added to the legend of Figure 3 in the revised manuscript (Page 25).

Referee #2:

I appreciate the authors' efforts in addressing the issues I raised. I am satisfied with their revision.

We thank the referee for appreciating our efforts in revising the manuscript.

Referee #3:

The investigators have addressed all my concerns in a thoughtful and responsive manner. I found the streamlined manuscript to be much more readable

We thank the referee for appreciating our efforts in revising the manuscript.

We thank the editor and referees for carefully reviewing our manuscript. We believe that incorporation of your comments has significantly improved our manuscript and made it more structured, readable and clearer.

Dear Dr. Kaur,

Thank you for the submission of your further revised manuscript to EMBO reports. I now went through this and your final p-b-p-response and consider the remaining points of referee #1 as adequately addressed.

Before proceeding with formal acceptance, I have these further editorial requests I ask you to address:

- Please provide a final title with not more than 100 characters (including spaces).
- Please add a direct link to the dataset to the 'Data availability section' and make sure that the dataset is public latest on the publication date of the manuscript.
- Please make sure that the number "n" for how many independent experiments were performed, their nature (biological versus technical replicates), the bars and error bars (e.g. SEM, SD) and the test used to calculate p-values is indicated in the respective figure legends (for main, EV and Appendix figures) of the final revised manuscript. Please also check that all the p-values are explained in the legend, and that these fit to those shown in the figure. Please provide statistical testing where applicable. Please avoid the phrase 'independent experiment', but clearly state if these were biological or technical replicates. Please also indicate (e.g. with n.s.) if testing was performed, but the differences are not significant. In case n=2, please show the data as separate datapoints without error bars and statistics. See also:

<http://www.embopress.org/page/journal/14693178/authorguide#statisticalanalysis>

If n<5, please show single datapoints for diagrams. It seems, 'n.s.' is missing from most diagrams. Moreover:

- Please note that the legends for figures EV 4b-d is not provided in a sequential manner (legend for figure EV 4c-d is provided before legend of figure EV 4b). This needs to be rectified.
- Please define the annotated p values ****/**/* as well as provide the exact p-values for the same in the legend of figure 1a-b, d-g; 2a-d; 3a-d; 4a-h; 5d-g, j; 6a-m; 7b, d-g, i; 8a-b; EV 1a-b, e-h; EV 2c-e; EV 3d-i; EV 4a, c-d; as appropriate.
- Please indicate the statistical test used for data analysis in the legend of figure 5b.
- Although 'n' is provided, please describe the nature of entity for 'n' in the legends of figures 1a-g; 2a-d; 4a-h; 5d-g, j; 6a-m; 7b, d-i; 8a-b; EV 1a-b, d-h; EV 2c-e; EV 3c, g-i; EV 5a-e.
- Please add to each legend (main, EV and Appendix figures, where applicable) a 'Data Information' section explaining the statistics used or providing information regarding replicates and scales. See: <https://www.embopress.org/page/journal/14693178/authorguide#figureformat>
- Please add scale bars of similar style and thickness to microscopic images, using clearly visible black or white bars (depending on the background). Please place these in the lower right corner of the images themselves. Please do not write on or near the bars in the image but define the size in the respective figure legend. Presently, most scale bars are too thin and hardly visible. Please improve these.

Best,

All editorial and formatting issues were resolved by the authors.

Dr. Rupinder Kaur
BRIC-Centre for DNA Fingerprinting and Diagnostics
Laboratory of Fungal Pathogenesis
Inner Ring Road
Uppal
Hyderabad, Telangana 500039
India

Dear Dr. Kaur,

As my colleague Achim Breiling is currently traveling, I have taken over the handling of your manuscript. I have gone through your revised manuscript and all looks fine now from the editorial side as well. I am thus very pleased to accept your manuscript for publication in the next available issue of EMBO reports. Thank you for your contribution to our journal.

Yours sincerely,
